



# Ice nucleation activities of soot particles internally mixed with sulphuric acid at cirrus cloud conditions

Kunfeng Gao[1,2,3], Chong-Wen Zhou[1], Eszter J. Barthazy Meier[4], and Zamin A. Kanji[3]

[1]School of Energy and Power Engineering, Beihang University, Beijing, China
[2]Shenyuan Honours College of Beihang University, Beihang University, Beijing, China
[3]Department of Environmental Systems Science, Institute for Atmospheric and Climate Science, ETH Zurich, 8092 Zurich, Switzerland
[4]Scientific Centre for Optical and Electron Microscopy, ETH Zurich, 8093 Zurich, Switzerland

*Correspondence to*: Zamin A. Kanji (zamin.kanji@env.ethz.ch) and Kunfeng Gao (gaokunfeng@buaa.edu.cn)

**Abstract.** Soot particles are important candidates for ice nucleating particles (INPs) in cirrus cloud formation which is known to exert a warming effect on climate. Bare soot particles, generally hydrophobic and fractal, mainly exist near emission sources. Coated or internally mixed soot particles are more abundant in the atmosphere and have a higher probability to impact cloud formation and climate. However, the ice nucleation ability of coated soot particles is not as well understood as that of freshly produced soot particles. In this study, two samples, a propane ($C_3H_8$) flame soot and a commercial carbon black were coated

with varying $wt\%$ of sulphuric acid ($H_2SO_4$). The ratio of coating material mass to the mass of bare soot particle was controlled and progressively increased from less than 5 $wt\%$ to over 100 $wt\%$. Both bare and coated soot particle ice nucleation activities were investigated with a continuous flow diffusion chamber operated at mixed-phase and cirrus cloud conditions. The mobility size and mass distribution of size selected soot particles with/without $H_2SO_4$ coating were measured by a scanning mobility particle sizer (SMPS) and a centrifugal particle mass analyser (CPMA) running in parallel. The mixing state and morphology

of soot particles were characterized by scanning electron microscopy (SEM) and transmission electron microscopy (TEM). In addition, the evidence for the presence of $H_2SO_4$ on coated soot particle surface is shown by Energy Dispersive X-ray spectroscopy (EDX). Our study demonstrates that $H_2SO_4$ coatings suppress the ice nucleation activity of soot particles to varying degrees depending on the coating thickness, but in a non-linear fashion. Thin coatings causing pore filling in the soot-aggregate inhibits pore condensation and freezing (PCF). Thick coatings promote particle ice activation via droplet

homogeneous freezing. Overall, our findings reveal that $H_2SO_4$ coatings will suppress soot particle ice nucleation abilities in the cirrus cloud regime, having implications for the fate of soot particles with respect to cloud formation in the upper troposphere.





## 1 Introduction

Black carbon (BC) particles associated with organics are called soot (Bond et al., 2013). Soot particles are of significance in both physical and chemical atmospheric processes in the atmosphere. In particular, these carbonaceous aerosol particles can engage in cloud formation process and form hydrometeors, which affects their lifecycle in the atmosphere and is the source of uncertainties to their overall climate impacts (Liu et al., 2020). A study estimated that 7.5 Mt BC was emitted into the atmosphere in the year 2000 globally with an uncertainty larger than 26.7 % (Bond et al., 2013). Lee et al. (2020) reported that aviation emission makes an important global contribution to anthropogenic climate forcing. Aviation soot particles, directly emitted by commercial aircrafts in the upper troposphere, are potential ice nucleating particles (INPs) at high altitudes where cirrus clouds usually form, and exert warming effects on climate (Liou, 1986). Cziczo and Froyd (2014) suggested that aviation soot particles can engage in contrail evolution and cirrus formation by inducing ice crystal formation heterogeneously at aircraft cruise altitudes. McGraw et al. (2020) also suggested that the largest uncertainty of soot particle climate impacts lies in the cirrus cloud regime. However, numerous studies reported that fresh soot particles are poor INPs and require high relative humidity (RH) and low temperature ($T$) conditions to form ice crystals (Möhler et al., 2005; Friedman et al., 2011; Kanji et al., 2011). In order to understand the climate impact of soot particles, it is essential to improve the knowledge about their ice nucleation activities in the atmosphere.

In the atmosphere, ice crystals can either be formed via homogeneous freezing of liquid droplets or induced by insoluble aerosol particles via heterogeneous ice nucleation (Vali et al., 2015). Homogeneous ice formation can only be triggered at $T <$ 235 K (homogeneous nucleation temperature, HNT) and relative humidity with respect to ice ($RH_i$) higher than 140 % (Koop et al., 2000). However, heterogeneous ice nucleation has a lower ice nucleation energy barrier with the aid of an INP which promotes the critical nucleation condition by providing an external surface for ice embryo formation. This is considered as an important pathway for ice crystal formation at cirrus altitudes (Cziczo et al., 2013). In general, heterogeneous ice nucleation can occur at lower $RH_i$ (< 140 %) conditions or warmer temperatures (> HNT) with the presence of INPs in the atmosphere, compared to homogeneous nucleation conditions. For example, Cziczo et al. (2013) reported that heterogeneous ice nucleation dominates the ice crystal formation process in cirrus clouds, depending on RH conditions as well as associated INP concentrations at cirrus altitudes. Before participating in an ice nucleation event, soot particles may interact with gaseous sulphur species in aircraft exhaust and can be coated by $H_2SO_4$ in the aviation plume (Chen et al., 1998). For instance, both sulphur oxides and water vapor are the products of aviation fuel combustion (Braun-Unkhoff et al., 2016), which can form supersaturated binary $H_2SO_4$-$H_2O$ solutions (Curtius, 2002) in aviation plumes. Wyslouzil et al. (1994) treated single soot particle in a gaseous $H_2SO_4$ atmosphere to investigate its hydration behaviour and found mass increase of the soot particle by 14 ± 6 % due to addition of $H_2SO_4$, equivalent to 0.1 % of the $SO_2$ in the aircraft plume ending up as a $H_2SO_4$ coating on the soot particle surface. It is conceivable that soot particles and $H_2SO_4$ can be internally mixed and form $H_2SO_4$ coated soot particles in the atmosphere, especially in high altitude aircraft corridors where aeroengines exhaust sulphur emissions. In



addition, some soot particles, generated by incomplete combustion from natural and anthropogenic sources contaminated by
sulphur material during industrial processes, can get advected to the upper troposphere by vigorous convection. Therefore,
both bare and H₂SO₄ coated soot particles should be considered in the study of cirrus cloud formation and corresponding
climate impacts. The coating material modifies both chemical and physical properties of bare soot particles (Bond et al., 2013).
The increased heterogeneity of coated soot particles is a significant source of uncertainty for evaluating their ice nucleation
abilities. For instance, soluble sulphate material is suggested to increase soot particle water adsorption and may change its
water interaction ability (Friedman et al., 2011). However, Kärcher and Lohmann (2002) reported that sulphate aerosol
particles are unlikely to exert a sensible influence on cirrus cloud formation with evidence, suggesting internally mixed soot
and sulphate particles may play a limited role in modifying background cirrus clouds in the atmosphere. Thus, the mixing sate
of soot particle with H₂SO₄ potentially regulates its ability to be a potential INP but remaining unconstrained.

Ice nucleation abilities of aviation soot particles and their surrogates have been investigated both in in-situ field measurements
and laboratory studies. The former shows that soot is present in contrail cirrus ice crystal residues suggesting that soot particles
can act as potential INPs for cirrus cloud formation upon contrail dissipation. In situ observations also provide some evidence
that the number of soot particles emitted by per unit mass of fuel burnt increases with increasing distance downstream of the
aviation plumes whereas the ice crystal numbers in the contrail decrease by an equivalent fraction (Kleine et al., 2018), which
suggests soot particles are initially embedded in contrail ice and then can be released to the upper troposphere after contrail
dilution and sublimation (dissipation) processes. In addition, soot particles with a more compacted morphology than the freshly
emitted ones are detected in aviation contrail ice crystal residues (Petzold et al., 1998) suggesting a change in shape and size
of the soot-aggregates due to contrail processing.

Laboratory studies focus on the ice nucleation mechanism of aviation soot particle surrogates to understand which particle
physiochemical property modulates their ice nucleation activities. Popovicheva (2004) suggested that hydrophobic soot
particles, containing a few water-soluble materials, only initiate ice nucleation at $T$ < HNT by deposition nucleation. Möhler
et al. (2005b) and Kanji et al. (2011) also reported that graphite soot particles can induce ice formation below homogeneous
freezing conditions at $T$ < 238 K. During deposition nucleation, it is assumed that there is no presence of a liquid phase but the
water vapor directly deposits onto the particle surface and nucleates ice crystals. Mahrt et al. (2018) suggested that pore
condensation and freezing (PCF) rather than deposition nucleation is responsible for soot particle ice nucleation activities,
given that porous soot particles are able to form ice crystals at RH$_i$ values lower than homogenous freezing conditions at $T$ <
HNT. Previous studies also suggested that soot particle ice nucleation ability depends on its morphology and size (Mahrt et
al., 2018; Nichman et al., 2019), hydrophilic and hydrophobic properties (Koehler et al., 2009; Biggs et al., 2017), chemical
composition and surface chemical characteristics (Möhler et al., 2005; Schill and Tolbert, 2012; Xue et al., 2019), surface
oxidation levels (Whale et al., 2015; Hausler et al., 2018), as well as the particle water interaction history (Marcolli, 2017;
Mahrt et al., 2020a; Mahrt et al., 2020b). Most recently, Marcolli et al. (2021) suggested that soot PCF process fundamentally



requires optimum pore structure, size and appropriate surface wettability of the soot particle. Numerous factors involved in determining the ice nucleation activity of bare soot particles can render the impacts of the aging process on ice nucleation,

where soot becomes internally mixed with other aerosols further changing the particle properties, more complicated. Therefore, systematic laboratory experiments to reveal the ice nucleation activity of certain soot samples and soot particles in different coating states are needed.

A foreign coating material changes the chemical composition and hygroscopicity of soot particles significantly (Zhang et al.,

2008; China et al., 2015; Zhang et al., 2020). Zhang and Zhang (2005) showed that $H_2SO_4$ coating onto soot particles, produced by a soot generator burning methane, hexane or kerosene, is irreversible and enhances its hygroscopicity. Möhler et al. (2005b) reported that polydisperse graphite spark soot particles coated with $H_2SO_4$ nucleate ice at an ice saturation level higher than that of bare graphite spark soot particles but lower than the homogeneous freezing RH ($RH_{hom}$) required by pure $H_2SO_4$ droplets, at a given $T$. However, the soot coating thickness effect on ice nucleation was not systematically compared in the

study because $H_2SO_4$ coated soot particles were produced with different coating conditions. Kulkarni et al. (2016) observed toluene or α-pinene coated diesel soot particles freezing homogeneously. Dalirian et al. (2018) investigated the coating effect of different organic compounds on the cloud droplet activation ability of soot particles and concluded that even a small amount of soluble coating is able to make hydrophobic soot particles become hydrophilic enough to be active could condensation nuclei (CCN). Therefore, the water interaction behaviour of coated soot particles can be changed significantly by the coating

material. Considering its solubility in water and strong hygroscopic ability, $H_2SO_4$ coating very likely enhances soot particle surface wettability, i.e. a lower soot-water contact angle (Mahrt et al., 2020b), which plays an important role in PCF (David et al., 2020; Marcolli et al., 2021). In addition, the coating process can alter soot particle morphology and influence the availability of potential pores for PCF at cirrus cloud conditions. It is reported that external material coating changes the initial soot particle size, particle internal microstructure and surface texture (Saathoff et al., 2003; Khalizov et al., 2009; Pei et al., 2018; Zhang et

al., 2020). The size growth caused by considerable addition of coating material is a possible coating effect and the coating material distribution may further modify the soot-aggregate structure. For example, oleic acid coating can significantly deform soot particle shape and increase particle mass whereas the removal of the coating material by heating seems to recover its morphology to some extent (Bambha et al., 2013). Schnitzler et al. (2017) also demonstrated that liquid material coating can change soot-aggregate structure by the tension induced by coating condensation on the particle surfaces. It is therefore possible

that, if the particle structure can be changed, the pore volume and pore size distribution of the soot-aggregate are also alterable. Considering that soot particle ice nucleation ability via PCF shows dependence on the aforementioned parameters, coating effects on soot particle ice nucleation need systematic studies with a focus on associated morphology and hygroscopicity changes.

In this study, both a propane ($C_3H_8$) combustion soot and a porous commercial carbon black are exposed to different $H_2SO_4$ supersaturation levels to generate coated soot particles. With progressively increasing $H_2SO_4$ coatings, size selected soot





particle ice nucleation activities are systematically investigated in the mixed-phase and cirrus cloud regimes and compared to bare soot particles. The mixing state of soot particle with $H_2SO_4$ is characterized with an emphasis on the morphology and a hypothesis about the coating process of soot particle with $H_2SO_4$ and the influence on cirrus ice nucleation is proposed based on the results presented herein.

## 2 Experimental methods

### 2.1 Experimental samples

In this study, two types of soot particles were investigated. First, a $C_3H_8$ flame soot is produced by a miniature combustion aerosol standard (miniCAST, model 4200, Jing Ltd., Zollikofen, Switzerland) soot generator. The Jing miniCAST forms a diffusion flame by burning $C_3H_8$ under a fuel-lean conditions. The associated soot particle is termed as mCASTblack hereafter. This soot sample is technically the same as the sample used by Mahrt et al. (2018; 2020a; 2020b), who studied the ice nucleation activities of fresh and aged mCASTblack soot particles in the continuous flow diffusion chamber HINC (Horizontal Ice Nucleation Chamber) (Lacher et al., 2017). At the outlet of the miniCAST, part of the exhaust, containing primary soot particles or small size aggregates, is diluted by filtered synthetic air. The resulting flow containing aerosol is then fed into a 0.125 $m^3$ continuous stirred tank reactor to generate larger and more homogeneous soot-aggregates. Finally, a 1 L $min^{-1}$ sample flow is sampled out of the tank and directed to a differential mobility analyser (DMA, classifier 3080, with a 3081 column and a polonium radiation source, TSI Inc.) after passing through a molecular sieve diffusion drier, in order to select monodisperse aerosol sample with mobility size of 200 or 400 nm.

The second type of soot is FW200, a commercial carbon black (Orion Engineered Carbons GmbH, OEC, Frankfurt, Main, Germany). FW200 carbon black is a product of incomplete combustion from hydrocarbon liquid fuels. There are two purposes here for choosing this commercial carbon black as one of the experimental samples. Firstly, it is used to represent porous and surface wettable soot particles in the atmosphere because of its large specific surface area value (550 $m^2$ $g^{-1}$ provided by the manufacturer; 526 $m^2$ $g^{-1}$ reported by Mahrt et al., 2018) and according to surface chemistry information provided by the manufacturer. Secondly, future studies can reproduce ice nucleation experiments with this commercial carbon black more precisely for analysis and comparisons. Mahrt et al. (2020b) used a wet dispersion method to aerosolize soot particles aged in $H_2SO_4$ (pH = 4) by a nebulizer with a magnetic stirrer. Instead of using an aqueous suspension, a dry dispersion method is utilized in this study to reduce changes in morphology caused by droplet evaporation after wet suspension. A Venturi nozzle is deployed with a glass jar containing dry soot powder mounted on a strong magnetic stirrer. Soot powder is suspended by the stirrer and then can be entrained into a high pressure $N_2$ flow by the nozzle. The soot aerosol flow generated is directed to a flow cascade, to buffer the aerosol flow and particle number concentration fluctuations. Finally, a 1 L $min^{-1}$ sample flow is directed to the DMA and 200 or 400 nm size selected soot particles can be generated.





## 2.2 Experimental instrumentation

Figure 1 shows the experimental setup schematic illustrating aerosol sample generation, sample particle size selection, soot
particle $H_2SO_4$ coating, ice nucleation experiments and particle characterization measurements. Firstly, the DMA selects 200
nm (sheath to sample flow ratio 13 : 1) or 400 nm (sheath to sample flow ratio 7 : 1) soot particles and a 1 L min$^{-1}$ monodisperse
sample flow is produced with a particle number concentration larger than 3,000 cm$^{-3}$ for the following experiments. The aerosol
sample (1 L min$^{-1}$) either goes through a home-built $H_2SO_4$ coating apparatus or passes by a dilution system with synthetic air
flow. This dilution system has equivalent volume and flow resistance to the coating apparatus, as well as the same dilution
ratio (~ 5 : 1). The diluted or $H_2SO_4$ coated soot aerosol sample (> 5 L min$^{-1}$) is split into five pathways, distributed to a high
efficiency particulate air capsule (HEPA capsule, PALL Corporation) for exhaust flow, a condensation particle counter (CPC;
Model 3772, TSI Inc.) for particle number concentration counting (1 L min$^{-1}$), an ice nucleation experiment flow (0.22 L min$^{-1}$)
to the HINC (Lacher et al., 2017), a SMPS (scanning mobility particle sizer, Classifier 3082, Column 3081, CPC 3776 low-
flow mode 0.3 L min$^{-1}$ or CPC 3772 1 L min$^{-1}$, TSI Inc.) flow for particle size distribution measurement, a CPMA (centrifugal
particle mass analyser, Cambustion Ltd., Cambridge, UK) flow (1.5 L min$^{-1}$) sucked by a CPC (Model 3787, TSI Inc.) for
particle mass distribution measurement, and a sample flow (1 L min$^{-1}$) for microscopic grids collection by the Zurich Electron
Microscope Impactor (ZEMI) (Aerni et al., 2018; Mahrt et al., 2018). The key distinction in this study is generating $H_2SO_4$
coated size-selected soot particles and studying their properties online for size, mass (density), microscopy and ice nucleation
ability.

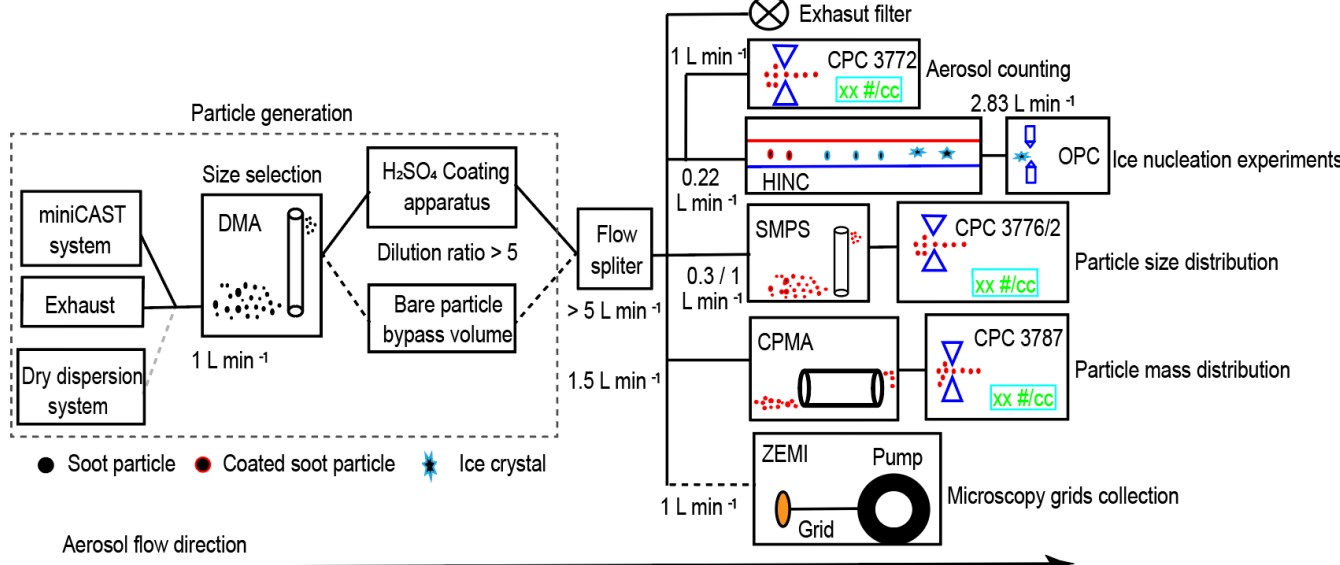


**Figure 1. Schematic of the experimental setup. miniCAST-miniature Combustion Aerosol Standard; DMA-Differential Mobility Analyser; CPC-Condensation Particle Counter; HINC-Horizontal Ice Nucleation Chamber; OPC-Optical Particle Counter; SMPS-Scanning Mobility Particle Sizer; CPMA-Centrifugal Particle Mass Analyser; ZEMI-Zurich Electron Microscope Impactor.**





### 2.2.1 Coating apparatus

The coating procedure follows the basic idea of heating to produce coating material vapour then cooling to condense or adsorb the coating material onto soot particle surfaces, finally generating $H_2SO_4$ coated soot particles, as depicted in Fig. 2. The coating mechanism is attributed to two pathways, including the direct condensation of supersaturated $H_2SO_4$ vapor and the adsorption of small $H_2SO_4$ particles formed by homogeneous nucleation (Bambha et al., 2013; Pei et al., 2018). The aerosol sample flow used to generate $H_2SO_4$ vapor is heated in an aluminium heating block before going through a flask containing

50 ml pure $H_2SO_4$ (Sigma-Aldrich, 95.0-97.0 %) mounted on another aluminium heating block. The temperature of the heating blocks is controlled by a LabView (National Instruments Corporation, Austin, Texas, US) program. While passing through the $H_2SO_4$ flask, soot particles will be mixed with $H_2SO_4$ vapor. Next, a dilution flow at the same temperature as the $H_2SO_4$/soot flow entrains the particles out of the flask. Downstream of the apparatus, there is a water-cooling system to generate $H_2SO_4$ supersaturation to nucleate or condense $H_2SO_4$ onto soot particle surfaces, thus generating $H_2SO_4$ internally mixed soot

particles. The cooling water temperature is maintained at 20 ℃ by a thermostat (LAUDA E300). By increasing the $H_2SO_4$ saturation and dilution flow temperature from 30 to 95 ℃, the $H_2SO_4$ coating $wt$ % (refer to Sect. 2.2.3 Eq. (1)) in terms of the ratio of $H_2SO_4$ coating mass to the bare soot particle mass can be increased monotonically.

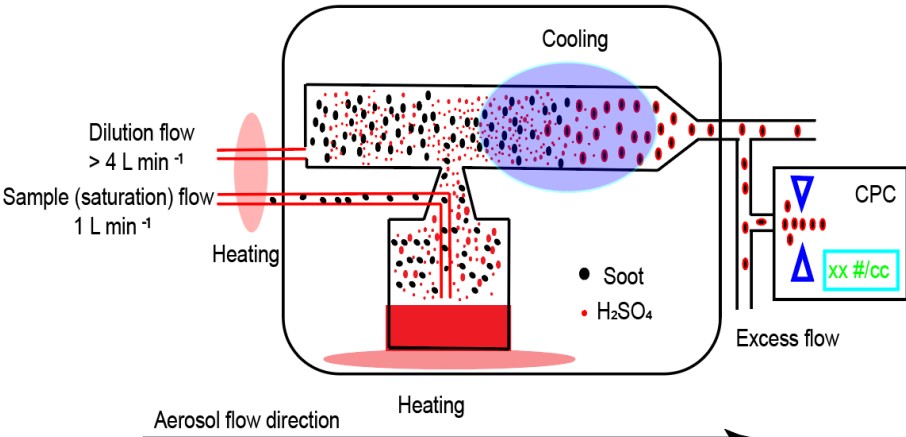

**Figure 2. The schematic of coating apparatus. Red line denotes heated tubing. Light red shaded area denotes heating block. Purple**
**shaded area denotes water cooling system.**

In order to generate internally mixed soot/$H_2SO_4$, the following operation procedure is strictly followed. First, a particle free synthetic airflow is used to flush the apparatus for several minutes at the very beginning of the experiment until the CPC records a zero particle number concentration at the outlet of the coating apparatus. Then, the heated soot aerosol sample flow is connected to the inlet of the apparatus. In order to check the particle mixing state, a SMPS-CPMA combined particle size

and mass distribution measurement is conducted at the outlet of the coating apparatus before starting the downstream experiments. The results demonstrate that only internally mixed soot particles with $H_2SO_4$ are produced from the coating apparatus, indicating no pure nucleated $H_2SO_4$. The SMPS results for 200 nm mCASTblack soot particles coating process are





presented in Appendix, Fig. A1 as an example. In the absence of soot, nucleation mode $H_2SO_4$ shows a high number

concentration and a small size mode of ~ 40 nm fitted by a log-normal distribution function. After feeding soot samples into

the coating apparatus, mixed aerosol particle size distribution mode shifts close to the size mode of bare soot particles,

meanwhile, the small size mode of $H_2SO_4$ particles becomes absent and the number concentration of $H_2SO_4$ particles decreases

dramatically (see Fig. A1). The CPMA mass scan results also show the homogeneity of the 200 nm mCASTblack soot particles

internally mixed with $H_2SO_4$, given that the log-normal mass distribution fitting for coated aerosol particles only shows a

single distinct peak (see Fig. A2) which indicates the mass mode of the aerosol is for $H_2SO_4$ coated particles only. Considering

that a 200 nm pure $H_2SO_4$ particle has a larger mass than a coated soot particle, we are confident that there is no pure 200 nm

$H_2SO_4$ particle existing in the aerosol flow.

### 2.2.2 Ice nucleation experiments

In this study, the HINC chamber (Kanji and Abbatt, 2009; Lacher et al., 2017) is utilized to investigate particle ice nucleation

abilities under varying RH conditions at a fixed temperature. HINC is a horizontal continuous flow diffusion chamber in which

aerosol particles can experience a known RH and $T$ condition for a variable time (~ 16 s in this study). Two parallel copper

plates, sandwiching a polyvinylidene fluoride (PVDF; Angst+Pfister AG, Zurich, Switzerland) frame with a thickness of 20

mm, form the core chamber space. The temperature of each copper plate is controlled by a thermostat (LAUDA, RP890) and

monitored by four thermocouples with an uncertainty of ± 0.1 K. Glass fibre paper (Pall Corporation, 66217) adhered on the

inner wall of each copper plate is wetted before the experiment and serves as a water vapor reservoir. During the experiment,

both walls are coated with a thin ice layer and remain at ice saturation condition ($RH_i$ = 100 %). By maintaining a temperature

difference ($\Delta T$) between the top and the bottom wall ($T_{top} > T_{bottom}$), a linear temperature distribution as well as a linear

distribution of water vapor pressure develops inside the chamber in the vertical dimension. However, the saturation water

vapor pressure has an exponential relation with the temperature. Hence, a nonlinear RH distribution will be generated and the

RH value at a fixed vertical position inside the chamber can be determined as well. Upstream of the chamber, there are four

ports drawing pure dry $N_2$ into the chamber as the sheath flow confining the aerosol sample flow to the centre of the chamber

at a defined RH and $T$ condition. A movable aerosol injector is used to direct the aerosol particles into the centre of the chamber

and can also be used to adjust particle residence time in the chamber. At the chamber outlet, an optical particle counter (OPC,

MetOne, GT-526S) maintains a total flow rate of 2.83 L min$^{-1}$. Before the HINC experiment, the $N_2$ sheath flow rate is adjusted

to ensure that 0.22 L min$^{-1}$ aerosol sample flow makes up part of the 2.83 L min$^{-1}$. A 1:12 aerosol sample to sheath flow ratio

is used during the whole course the study. During a HINC experiment, the CPC is used to count the total aerosol particle

number entering the chamber and the OPC is used to detect the total number of ice crystals or water droplets coming out of

the chamber in six size channels (0.3, 1.0, 2.0, 3.0, 4.0 and 5.0 μm). Thus, the ratio of ice crystal or water droplet number to

the total aerosol particle number can be derived, which is the aerosol particle activated fraction (AF) value.





For bare and coated soot samples, HINC experiments are performed from 218 to 243 K with six measurements at different $T$ and with a 5 K interval. At each $T$, at least two RH scans are conducted for each soot sample from ice saturation condition to RH conditions above water saturation. The $RH_i$ scan rate is 2 % per minute. The $RH_w$ values of the aerosol sample upstream of HINC are less than 5 % and monitored by a RH sensor. Soot particle $H_2SO_4$ coating $wt$ % for ice nucleation experiments are shown in Table 1.


**Table 1. The ratio of coated $H_2SO_4$ mass to the bare soot particle mass ($wt$ %) for 200 and 400 nm mCASTblack and FW200 soot particles. The uncertainty represents one standard deviation. \*denotes samples that are analysed with SEM technique (see Sect. 3.3.1).**

| Coating $T$ (℃) | $H_2SO_4$ coating ($wt$ %) | | | |
| --- | --- | --- | --- | --- |
| | mCASTblack | | FW200 | |
| | 200 nm | 400 nm | 200 nm | 400 nm |
| 30 | 2.7 ± 1.9 | 1.9 ± 3.5* | 3.5 ± 2.6 | 1.8 ± 2.3* |
| 35 | 5.4 ± 2.0 | 4.2 ± 3.9 | | 3.8 ± 1.5 |
| 40 | 8.5 ± 3.6 | 6.2 ± 4.3 | 9.3 ± 2.8 | 6.1 ± 1.8 |
| 45 | 10.9 ± 2.9 | | 15.0 ± 4.1 | 8.0 ± 2.0 |
| 50 | 15.6 ± 2.4 | 10.9 ± 4.4 | 18.1 ± 3.8 | 8.7 ± 1.4 |
| 55 | 19.5 ± 3.1 | 13.7 ± 3.7 | 22.3 ± 3.7 | |
| 65 | 24.2 ± 3.8 | 20.9 ± 3.5 | 28.8 ± 4.8 | 10.8 ± 2.6 |
| 75 | | 23.6 ± 4.2 | | 18.8 ± 2.7 |
| 80 | 28.1 ± 4.8 | | 40.3 ± 4.5 | |
| 85 | | 31.1 ± 4.6* | | 26.4 ± 2.9 |
| 95 | | | 135.3 ± 7.6 | 65.0 ± 4.2* |

**2.2.3 Particle characterization measurements**

**SMPS-CPMA:** The aerosol particle size distribution is measured by a SMPS system after size selection and/or coating process (see Fig.1). For 200 nm size selected soot particles with/without coating, the CPC 3772 with a flow rate of 1 L min$^{-1}$ is used and the SMPS measurement can cover a size scanning range from 12.6 to 572.5 nm, which means the double-charged particles selected by the DMA upstream can be detected. For 400 nm soot aerosol samples, the SMPS utilizes a low-flow mode CPC 3776 with a 0.3 L min$^{-1}$ flow rate and can scan up to a size value of 914 nm, which also covers the 680 nm particles carrying two charges but treated as a 400 nm particle by the upstream DMA. The amount of double-charged particles is approximately

16 and 29 % for 200 and 400 nm mCASTblack bare soot particles, respectively (see Fig. A3), and double-charged 200 and 400 nm FW200 bare soot particles amount to 21 % and 26 %, respectively (see Fig. A4). The CPMA is operated in mass scanning mode at a constant speed. A water CPC 3787 running in high-flow mode (1.5 L min$^{-1}$) is used to sample the aerosol through the instrument. The mass scanning range covers the mass value of double-charged particles (see Fig. A2). Single

particle mobility size is derived from a lognormal fitting of the aerosol size distribution measured by SMPS, assuming soot particles to be spherical. The mass value of a single particle is derived from CPMA mass distribution scans with a similar mathematical data processing method. Based on single soot particle mobility size and mass results, $H_2SO_4$ coating $wt$ %,




equivalent coating monolayers (assuming uniform coating) and particle effective density are calculated. The $H_2SO_4$ coating $wt\%$ is the ratio of soot particle coated $H_2SO_4$ mass to the mass of initial bare soot particle and defined as:

$$wt = \frac{x-y}{y} \times 100 \qquad (1)$$

where $x$ is the mass of $H_2SO_4$ coated by the soot particle and $y$ is bare soot particle mass. The number of equivalent coating monolayers ($ML$) is calculated according to Wyslouzil et al. (1994) and given by:

$$ML = \frac{wt \cdot N_A}{100 M_W \cdot S \cdot N_M} \qquad (2)$$

where $wt$ is the coating mass percentage value, $N_A$ is the Avogadro constant, $M_W$ is the molecular weight of $H_2SO_4$, $N_M$ is the number of $H_2SO_4$ molecules per cm$^2$ corresponding to one monolayer coverage ($4.5 \times 10^{14}$ cm$^{-2}$) (Wyslouzil et al., 1994), $S$ is the specific area of the soot particle (mCASTblack 120 m$^2$ g$^{-1}$, FW200 526 m$^2$ g$^{-1}$) (Mahrt et al., 2018). According to McMurry et al. (2002), the particle effective density ($\rho_{eff}$) expression is given as:

$$\rho_{eff} = \frac{6m}{\pi \cdot d^3} \qquad (3)$$

where $m$ is the single particle mass and $d$ is the particle mobility size. To control the $H_2SO_4$ coating $wt\%$, a number of factors need to be considered, including initial soot aerosol particle number concentration, saturation and dilution flow rate and temperatures, the temperature of pure $H_2SO_4$ in the flask and subsequent cooling condition (see Fig. 2), the setup volume, as well as the intrinsic soot particle properties. Here, temperatures, flow rates and the whole volume of the apparatus is well controlled. However, it is impossible to adjust the initial soot sample particle number concentration to be identical for every experiment. As a result, even though the $H_2SO_4$ coating thickness increases with increasing $H_2SO_4$ saturation temperatures, for per soot sample coating, the result shows some variance. In this study, the standard deviation value for each coating $wt\%$ is provided in Table 1 and in each panel of Fig. 3.

The $H_2SO_4$ coating $wt\%$, particle mobility size, particle $\rho_{eff}$ and the equivalent $ML$ value for each soot sample are presented in Fig. 3. Note that the equivalent $ML$ values highly depend on the soot particle specific surface area given by BET (Brunauer-Emmett-Teller) analysis (Brunauer et al., 1938), which can be influenced by different pre-treatment levels and also different gas probes used in the physisorption measurement (Lowell et al., 2004). For example, Ouf et al. (2019) reported that soot BET specific surface area can be severely influenced by the sample outgassing level and can vary in a significant range. Furthermore, it is suggested that $N_2$ BET and Argon BET results for the same soot sample show variance (Lowell et al., 2004). Nevertheless, the results in Fig. 3 still can provide relatively comparable information to evaluate soot particle mixing states with different $H_2SO_4$ coating thicknesses. Overall, 200 nm size selected soot particle mobility size can increase by ~ 5 % with increasing $H_2SO_4$ coating $wt\%$ but starts to decrease dramatically when the $H_2SO_4$ coating $wt\%$ is larger than 20 %, which means first size growth and then shrinkage (collapse). Whereas 400 nm size selected soot particle does not show apparent size growth and





starts to collapse when the coating mass percentage reaches ~ 20 %. With increasing coating $wt$ %, the equivalent $H_2SO_4$ coverage $ML$ and coated particle $\rho_{eff}$ increase monotonically as expected.

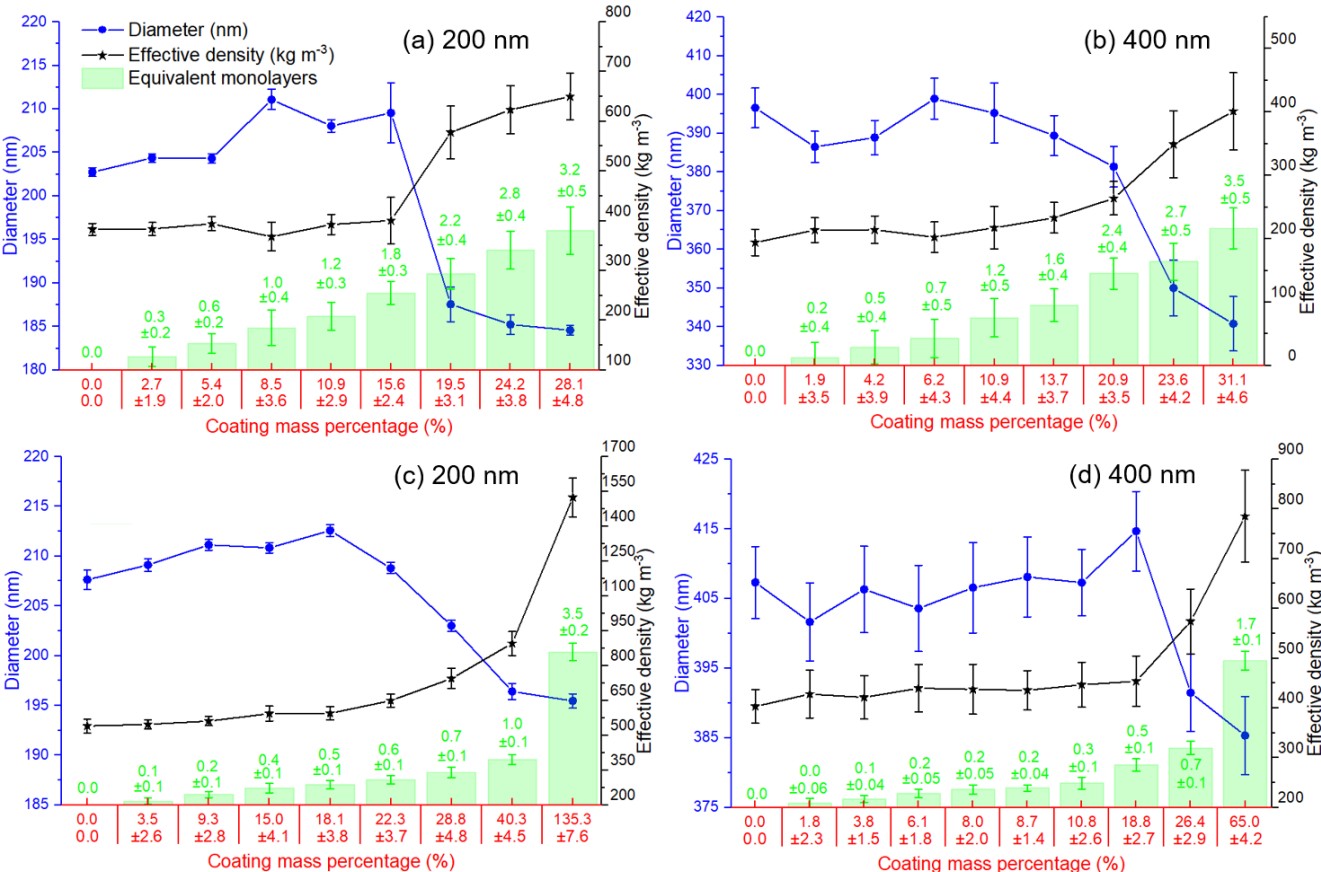

295

**Figure 3. The mobility size, effective density and the number of equivalent $H_2SO_4$ coating monolayers of mCASTblack 200 nm (a) and 400 nm (b), and FW200 200 nm (c) and 400 nm (d).**

**Scanning Electron Microscopy (SEM)** (Zeiss Leo 1530, Carl Zeiss AG, Oberkochen, Germany) is used to observe the $H_2SO_4$
300   coating state on soot particles. Both coated and uncoated FW200 and mCASTblack 400 nm soot-aggregates are collected on
400 mesh Cu girds with a formvar/carbon support film (TED, PELLA, Inc.). All grids are collected using the Zurich Electron
Microscope Impactor (ZEMI), a home-built and semi-automated rotating drum impactor (Aerni et al., 2018; Mahrt et al., 2018)
running in parallel to HINC ice nucleation experiments. During the grid sampling, ZEMI pulls a 1 L min$^{-1}$ aerosol sample flow
as shown in Fig. 1. The total sampling time is between 5 and 10 minutes depending on the particle concentration. In addition
to bare particles, coated soot-aggregates with the lowest and the highest $H_2SO_4$ coating $wt$ % (see Table 1) in the ice nucleation
experiments are collected for SEM analysis. For convenience of discussion in Sect. 3, these two coated conditions will be
termed as thin coating and thick coating, respectively. Images of interest are obtained at magnification values 20k and 200k.





The small magnification value is used to have an overview of the particles. Soot particle morphology change and $H_2SO_4$ coating state are analysed from high resolution images.


**Transmission Electron Microscopy (TEM) and High Resolution TEM (HR-TEM)** (JEOL-1400+ TEM, JEOL Ltd., Tokyo, Japan, operated at 120 kV; Hitachi HT7700 EXALENS, Hitachi Ltc., Chiyoda, Japan, operated at 100kV; TFS Talos F200X, operated at 200 kV, and TFS F30, operated at 300 kV, both Thermo Fisher Scientific Inc., Waltham MA, USA) are utilized to detect morphology changes induced by $H_2SO_4$ coating and physical evidence on $H_2SO_4$ coating. These grids were collected

separately from the SEM grids. To match with corresponding ice nucleation experiments, the coating $wt\%$ (see Table 2) of soot-aggregates for TEM analysis was controlled to be comparable to those coating $wt\%$ shown in Table 1 for 200 nm soot samples. Again, the lower $H_2SO_4$ coating $wt\%$ will be termed as thin coating and the higher one will be termed as thick coating for further SEM results discussion. Both bare and coated soot-aggregates are collected on Quantifoil Cu 200 mesh R2/2 grids (Quantifoil Micro Tools GmbH, Großlöbichau, Germany). Open holes of the R2/2 pattern provide a view on soot-

aggregates without a carbon film background.

**Energy Dispersive X-ray spectroscopy (EDX) (**TFS Talos F200X equipped with a Super-X EDS system) is used for chemical evidence on the presence of $H_2SO_4$ on soot particle surfaces. These grids used for EDX are the same as those for the TEM measurements. Both bare and coated soot-aggregates are analysed and results are presented both by 2D element distribution

maps and conventional element spectra. Carbon (C), oxygen (O) and sulphur (S) element mass content for soot particles with different $H_2SO_4$ coating $wt\%$ are also calculated and normalized to compare their chemical composition change.

**Table 2. The ratio of coated $H_2SO_4$ mass to the bare soot particle mass ($wt\%$) for 200 nm mCASTblack and FW200 soot-aggregates collected for TEM analysis. The uncertainty represents one standard deviation.**

| T (℃) | $H_2SO_4$ coating ($wt\%$) | |
| --- | --- | --- |
| | mCASTblack (200 nm) | FW200 (200 nm) |
| 30 | 2.9 ± 2.8 | 2.3 ± 2.2 |
| 80 | 30.2 ± 3.3 | |
| 95 | | 139.3 ± 10.8 |

**3 Results and discussion**

Soot particle ice nucleation abilities are presented in terms of AF curves as a function of $RH_w$ and $RH_i$ in Figs. 4 to 7. The 1 μm OPC channel data is used to plot AF curves, referring to the ratio of the total number of particles exiting the HINC chamber larger than 1 μm to the total number of particles entering the chamber as sampled by the CPC. The particle property results obtained by characterization measurements in Sect. 2 will be used to understand the corresponding soot particle ice nucleation

activities. Figs. 4 and 5 present the AF plots for coated 200 and 400 nm mCASTblack particles and uncoated counterparts as



a function of RH values at $T$ from 218 to 243 K. Ice nucleation activities of bare and coated 200 and 400 nm FW200 particles are shown in Figs. 6 and 7, respectively. Both bare mCASTblack and FW200 soot particle can only form water droplets above water saturation conditions at 243 and 238 K ($T$ > HNT), regardless of particle size. At these two temperatures, OPC signals are only observed at small channels but signals are absent at the largest channel, i.e. 5 µm channel (see Figs. B5 to B8). This

is because water droplet growth by vapor diffusion is less efficient than that of ice crystal growth at these conditions ($RH_w$ < 105%) in HINC (Lohmann et al., 2016). Thus, water droplets do not grow up to 5 µm and are thus not detected in the 5 µm OPC channel, however ice crystals can grow up to this size value (Lacher et al., 2017; Mahrt et al., 2018). Therefore, we conclude no ice nucleation onto bare soot particles at 243 and 238 K. This is in agreement with Kanji et al. (2020) who suggested that commercial black carbon particles require temperatures lower than HNT to freeze. For mCASTblack, ice

nucleation results were the same for both 200 and 400 nm particles showing no significant activation at RH < $RH_{hom}$ at cirrus relevant temperatures ($T$ < HNT). FW200 particles exhibit significant ice nucleation at $T$ < HNT unlike the mCASTblack, with 400 nm FW200 particles being more active INPs than 200 nm samples. Unsurprisingly, excellent agreement is achieved for the results of bare mCASTblack and FW200 soot between this study and Mahrt et al. (2018), who also studied these two types of soot INP samples using the same chamber. At $T$ < HNT, AF curves for bare mCASTblack soot particles stay in the

uncertainty range of homogeneous freezing. As shown in Figs. 4 and 5, only < $10^{-3}$ of the soot sample can form ice crystals at RH values slightly lower than the homogeneous RH values, even at 218 K. This is consistent with the results from Mahrt et al. (2018; 2020a; 2020b). As shown in Figs. 6 and 7, bare FW200 soot particles are active INPs and can form ice crystals at RH < $RH_{hom}$ when $T$ < HNT, also in line with the results presented by Mahrt et al. (2018). The authors suggested that these soot particles with small mesopores can be effective INPs and form ice crystals via PCF at humidity conditions lower than

homogeneous freezing conditions in the cirrus cloud regime (Mahrt et al., 2018). The paucity of mesopores relevant to PCF and its low surface wettability make mCASTblack soot a poor INP at $T$ < HNT, compared to FW200 soot. Here, PCF relevant mesopores refer to pores with right size and shape which are not only small enough to induce liquid water capillary condensation at $RH_w$ < 100% due to the inverse Kelvin effect but also large enough to allow ice growing out of the pore upon homogeneous freezing of supercooled pore water (Marcolli, 2014; Marcolli, 2020). The active ice nucleation ability of FW200

can be attributed to its low soot-water contact angle and abundant mesopore structures.

In addition to bare soot particles, more than eight different $H_2SO_4$ coating $wt$ % are performed from less than one equivalent molecule monolayer to several equivalent molecule monolayers coverage. For ease of discussing the ice nucleation activity comparison between bare and coated particles, three typical coating states, namely thin, medium and thick coating, are selected

as representative cases. The results for all coating $wt$ % corresponding to Table 1 are given in the Appendix B, Figs. B1 to B4. Thin coating stands for a $H_2SO_4$ coating $wt$ % less than 3.5 % of its initial mass value. Medium coating represents a $H_2SO_4$ coating of $wt$ % ∼ 10 % whereas thick coating represents a $H_2SO_4$ coating of $wt$ % larger than 25 %.





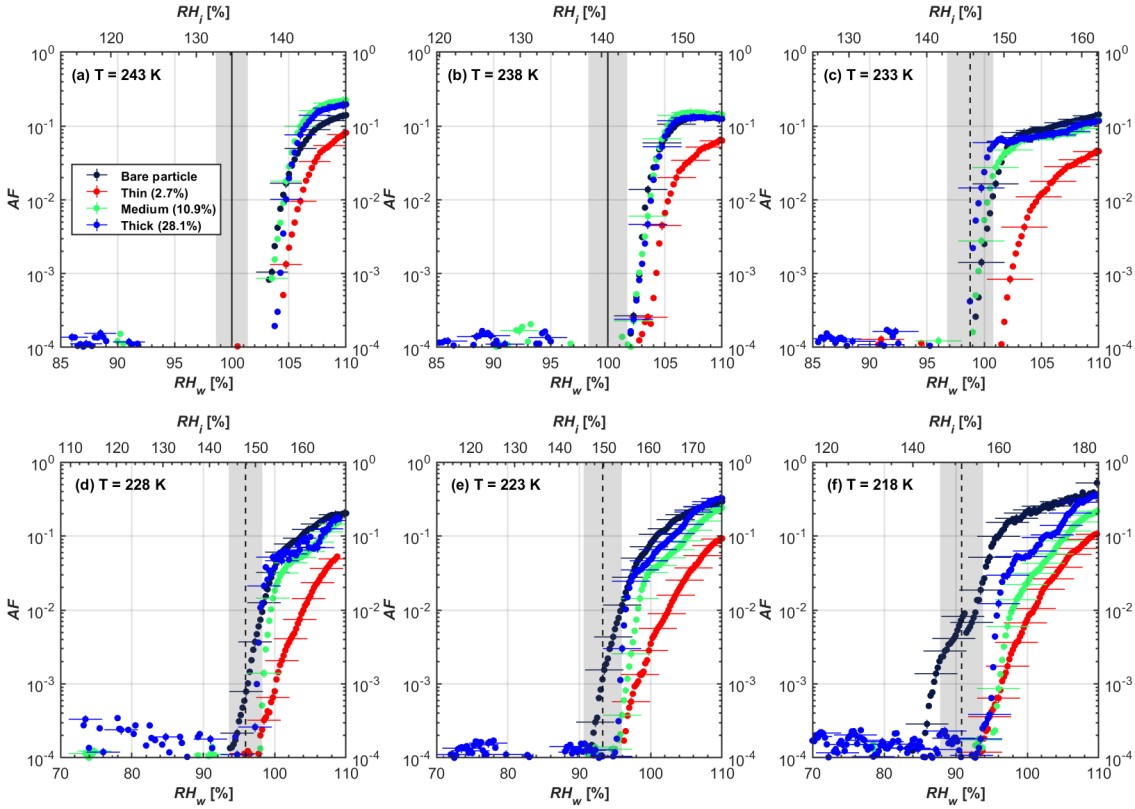

**Figure 4. AF as a function of RH at the given temperatures for 200 nm mCASTblack soot particles. Black solid lines represent water saturation conditions according to Murphy and Koop (2005). Black dashed lines denote the expected RH values for solution droplet homogeneous freezing at $T$ < HNT (Koop et al., 2000). The grey shaded area shows the possible RH variation and uncertainty in HINC for the calculated water saturation and homogeneous freezing conditions. The percentages represent the $H_2SO_4$ coating *wt %*.**





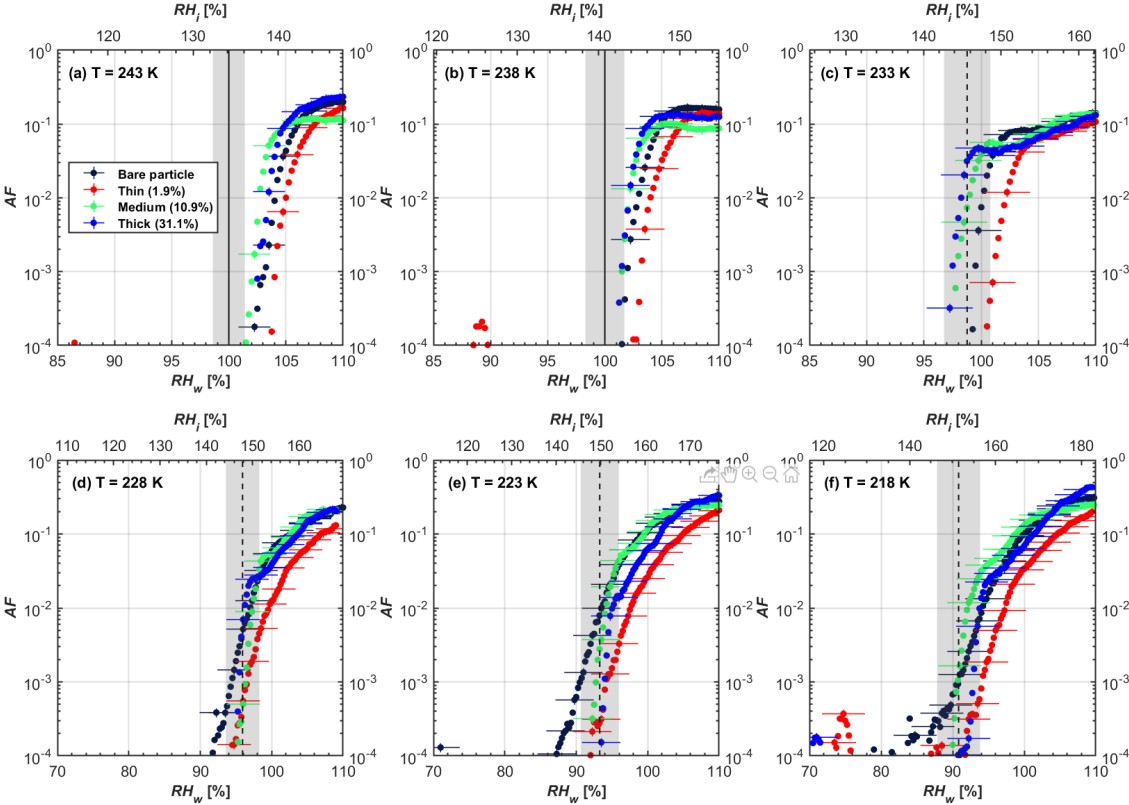

Figure 5. AF as a function of RH at the given temperatures for 400 nm mCASTblack soot particles. Black solid lines represent water saturation conditions according to Murphy and Koop (2005). Black dashed lines denote the expected RH values for solution droplet homogeneous freezing at $T$ < HNT (Koop et al., 2000). The grey shaded area shows the possible RH variation and uncertainty in HINC for the calculated water saturation and homogeneous freezing conditions. The percentages represent the $H_2SO_4$ coating $wt$ %.



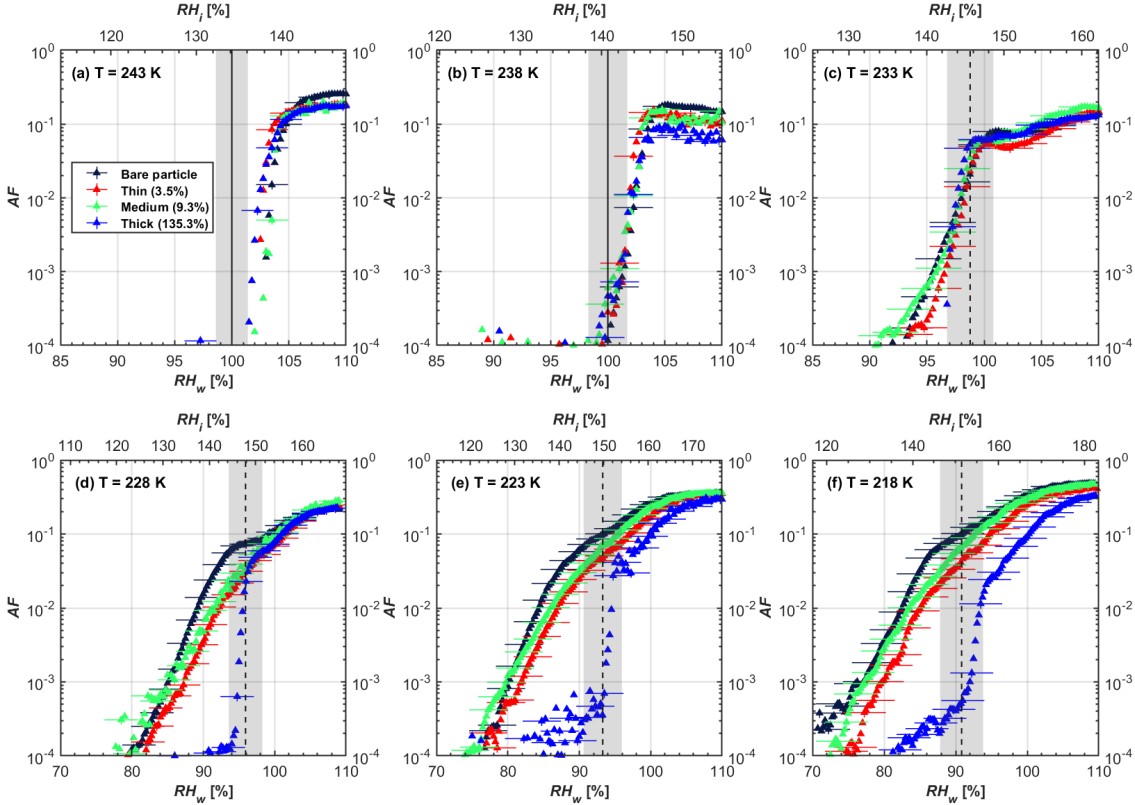

**Figure 6. AF as a function of RH at the given temperatures for 200 nm FW200 soot particles. Black solid lines represent water saturation conditions according to Murphy and Koop (2005). Black dashed lines denote the expected RH values for solution droplet homogeneous freezing at $T$ < HNT (Koop et al., 2000). The grey shaded area shows the possible RH variation and uncertainty in**

**HINC for the calculated water saturation and homogeneous freezing conditions. The percentages represent the $H_2SO_4$ coating $wt$ %.**



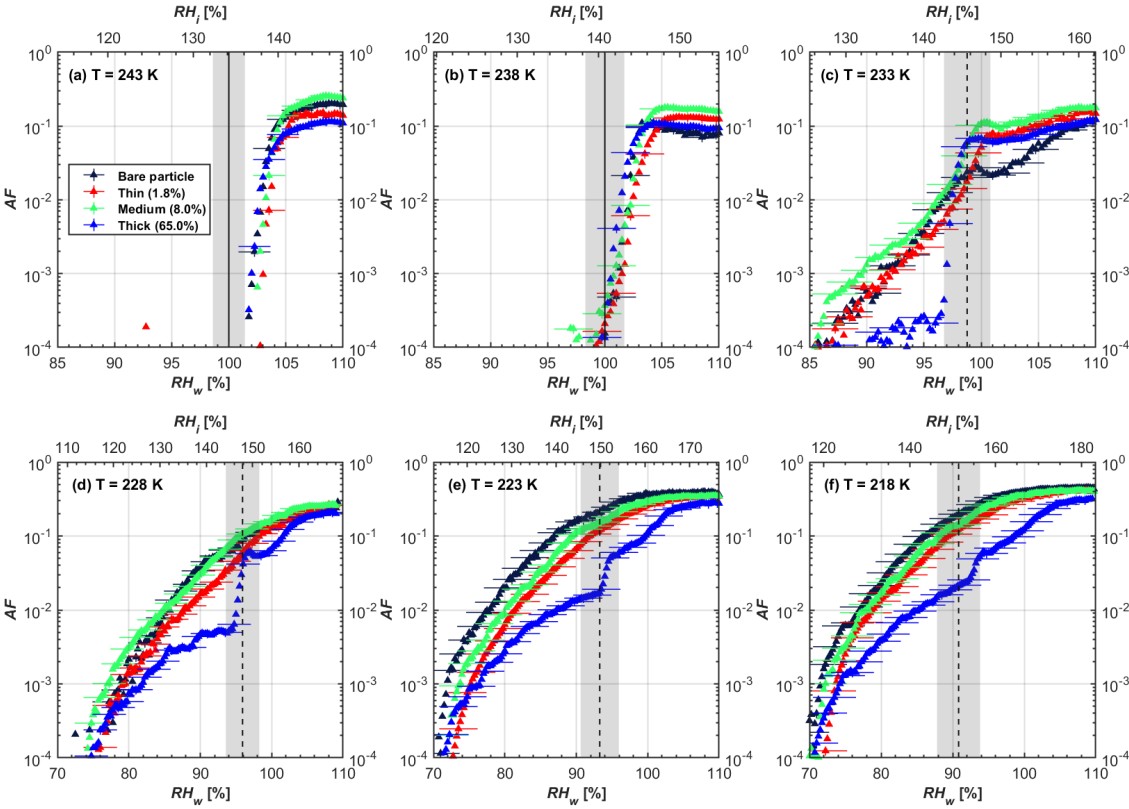

**Figure 7. AF as a function of RH at the given temperatures for 400 nm FW200 soot particles. Black solid lines represent water saturation conditions according to Murphy and Koop (2005). Black dashed lines denote the expected RH values for solution droplet**
**homogeneous freezing at $T <$ HNT (Koop et al., 2000). The grey shaded area shows the possible RH variation and uncertainty in HINC for the calculated water saturation and homogeneous freezing conditions. The percentages represent the $H_2SO_4$ coating $wt$ %.**

### 3.1 H₂SO₄ coated mCASTblack soot ice nucleation results

At $T >$ HNT (243 and 238 K), both coated and uncoated particles require water saturation conditions to form water droplets.
With similar evidence to bare particles, we conclude no ice nucleation at $T >$ HNT for coated soot of all coating masses. Soot particles coated with $H_2SO_4$ are expected to be more hygroscopic and act as CCN if a uniform coating is assumed. However, the AF curves for both 200 and 400 nm coated mCASTblack soot stay within error bars with those of bare particles at $T >$ HNT, as shown in Figs. 4 and 5. There is some indication that coated particles are better CCN at $T >$ HNT. However, we cannot claim that the hygroscopicity is significantly different as seen the AF curves overlapping with those of bare soot
particles, suggesting that the coatings may be non-uniform but form clusters on soot-aggregates. Non-uniform coating patterns can be seen from SEM images in Fig. 8. Muller et al. (1996) and Persiantseva et al. (2004) suggested that soot particle





micropores play an important role in the soot-water interaction activities if there is no soluble material available on its surfaces. Popovicheva et al. (2008a; 2008b) demonstrated that capillary condensation induced by mesopores in soot-aggregates can occur when $RH_w$ values are larger than 80 %. Therefore, it is feasible that a small amount of $H_2SO_4$ coating on mCASTblack

particle impacts its soot-water interaction activities if non-uniformly distributed coating material leads to pore filling which reduces mesopore availability. However, if further $H_2SO_4$ coating provides soluble material on soot surfaces for water adsorption, it compensates the unavailability of pores caused by thin coating pore filling and enhances soot update ability because of its high hygroscopicity. Overall, there is no significant change for mCASTblack soot CCN activity after $H_2SO_4$ coating using the method presented in this work.


At $T$ < HNT (233-218 K), all coated particles only form ice via homogeneous freezing and there is an apparent freezing depression particularly at 223 and 218 K for thinly coated soot, as can be seen in Figs. 4 and 5 by the delayed freezing onset of coated particles compared to that of bare particles at each $T$. This implies that $H_2SO_4$ coating inhibits the PCF mechanism at these low $T$. From Fig. 4, thinly coated 200 nm mCASTblack soot particles show the largest suppression in the onset of ice

formation among all coating masses. This shows coherence with the results at warm temperatures ($T$ > HNT) and can be explained by the $H_2SO_4$ pore filling effect. Homogeneous freezing of supercooled water in pores is an important step in a PCF activation process (Marcolli et al., 2021; Marcolli, 2014). According to Koop et al. (2000), the homogeneous freezing rate depends on the liquid water activity. Due to $H_2SO_4$ filled in pores for thin coating cases, inverse Kelvin effect induced water uptake will form high concentration $H_2SO_4$ solution with a low water activity, which leads to depressed homogeneous freezing

as it requires higher $RH_w$ conditions than $RH_{hom}$ for freezing. This inhibits the bulk freezing of the soot particle with low $H_2SO_4$ coating $wt$ % until the particle takes up sufficient water to dilute enough for bulk droplet freezing. With more coating (in medium and thick cases), hygroscopic $H_2SO_4$ adsorption on soot surfaces competes with $H_2SO_4$ pore filling and promotes the water uptake ability of soot particles to form a bulk water droplet more readily. This can explain why the soot particle with thicker coating tends to freeze homogeneously like a bulk solution droplet at higher RH conditions. More details about the

interaction of soot particle with $H_2SO_4$ will be discussed in Sect. 3.3. The similar ice nucleation activity of 400 nm coated mCASTblack soot particles can be seen from Fig. 5.

Results for coated mCASTblack soot in this study are comparable to the literature. For instance, ice nucleation results at low $T$ are in good agreement with previous $H_2SO_4$ coated miniCAST $C_3H_8$ soot results from Crawford et al. (2011), who treated

polydisperse miniCAST soot aerosol in saturated $H_2SO_4$ vapor and studied its ice nucleation behaviour at $T$ from 220 to 230 K. Similarly, the authors concluded that the addition of $H_2SO_4$ inhibits soot particle ice activation and shifts the freezing onset RH towards the homogeneous freezing conditions of pure $H_2SO_4$. Möhler et al. (2005b) also reported a suppression in the ice nucleation ability of polydisperse graphite spark generated soot samples (~25-300 nm) coated with $H_2SO_4$ (~40-350 nm) which freeze towards homogeneous freezing conditions. However, Mahrt et al. (2020b) reported a significant ice nucleation

enhancement for 400 nm mCASTblack soot particles aged in a bulk $H_2SO_4$ aqueous solution (pH = 4) and coated by dry



residues of $H_2SO_4$. The authors used a different aging method to treat the soot sample and the $H_2SO_4$ ageing effect and coating material distribution are different from the coating conditions in this study. The 400 nm particles in Mahrt et al. (2020b) were generated by atomizing a bulk aqueous suspension followed by drying and size selection. If a solution droplet spanning a size range from 400 nm to 2 μm is assumed before drying, the $H_2SO_4$ coating $wt$ % is estimated to be in the range from 0.0026 %

to 0.32 %, much less than the coating $wt$ % in this study. Furthermore, the $H_2SO_4$ coating generated in this way can be distributed more uniformly over the soot particle surface, compared to a nonuniform $H_2SO_4$ coating in this study (see Sect. 3.3). On the other hand, soot particles experienced a water interaction process when they were treated with $H_2SO_4$ aqueous solution, which means water interaction induced morphology changes and particle compaction could be responsible for this ice nucleation promotion as suggested by soot pure water ageing case in the same study by Mahrt et al. (2020b). In addition,

DeMott et al. (1999) investigated the freezing RH conditions for multiple-layer $H_2SO_4$ coated polydisperse lamp black soot particle (Degussa Corporation, Frankfurt, Germany) with a size distribution mode of 240 nm under cirrus cloud conditions, using a continuous flow diffusion chamber. The authors reported that these coated soot particles require $RH_i$ values 152 and 156 % respectively to reach 1 % AF level at 228 and 223 K (Demott et al., 1999), which is close to the $RH_i$ values 152 and 154 % at the same $T$ for 200 nm mCASTblack soot particles with thick $H_2SO_4$ coatings presented in this study. It is reported

that soot particles with aggregate compaction after experiencing a cloud process can nucleate ice via PCF at RH < $RH_{hom}$ conditions when $T$ < HNT (Mahrt et al., 2020a). However, even though the thick $H_2SO_4$ coating of mCASTblack soot particle in this study results in a significant size shrinkage (see Fig. 3a and b), i.e. some extent aggregate compaction (see Figs. 8c and 9c), its ice nucleation ability is not promoted compared to the uncoated particles. This suggests that the availability of pores (i.e. pore volume) without a freezing depression will be a prerequisite for PCF ice nucleation despite of lowering the soot-

water contact angle by $H_2SO_4$ coating.

### 3.2 $H_2SO_4$ coated FW200 soot ice nucleation results

In general, $H_2SO_4$ coating has limited effects on FW200 soot particle droplet activation ability at $T$ > HNT as shown in Figs. 6 and 7. There is no significant deviation between the AF curves of soot particles (both for 200 and 400 nm) with and without $H_2SO_4$ coating at 243 and 238 K. It is safe to conclude that the $H_2SO_4$ coating generated in this study does not enhance FW200

soot particle water interaction and CCN activation ability at these two $T$. The AF curves of bare and coated FW200 soot particles in Figs. 6 and 7 are consistent with the study conducted by Koehler et al. (2009) who reported that the CCN activation of hydrophilic aviation kerosene soot particles requires higher RH conditions than the Kelvin RH limit (Henson, 2007), which is required by wettable particles to show CCN activation. We attribute our results to the absence of a uniform $H_2SO_4$ coating similar to that discussed above. With such a non-uniform coating, the hygroscopicity of coated $H_2SO_4$ does not exhibit its

water interaction enhancement for its host soot particle.

From the AF curves in Figs. 6 and 7 at $T$ < HNT, thinly and moderately coated FW200 soot particles form ice crystals at RH < $RH_{hom}$ showing that PCF is still well preserved. This means pores or voids with right size for PCF are not all blocked or are



still available when soot particle is coated by $H_2SO_4$, also implying that the adsorbed $H_2SO_4$ by the soot particle does not distribute evenly or uniformly. In general, thin and moderate $H_2SO_4$ coatings cannot depress FW200 soot ice nucleation ability for both 200 and 400 nm coated FW200 soot particles. At $T < $ HNT, the 200 nm soot particles with thick $H_2SO_4$ coverage show an obvious ice activation depression and an apparent freezing mechanism shift from PCF to homogeneous freezing near the homogeneous freezing RH thresholds, as shown in Fig. 6. Thickly coated 400 nm FW200 soot particles in Fig.7 show a similar ice nucleation onset compared to bare soot particles but with a slight suppression in total AF observed, suggesting the ice

nucleation mesoporous volume for PCF is still available. Approaching $RH_{hom}$, there is a clear shift in the slope of the AF curves at $T \leq 233$ K in Fig.7 suggesting a mechanism change from PCF to bulk homogeneous freezing, which aligns with the predictions of Koop et al. (2000). Overall, 200 nm soot particle ice nucleation abilities are more affected by the thick $H_2SO_4$ coating compared to 400 nm particles. Given the approximate spherical structure of a FW200 soot-aggregate, a larger size and surface area will require a higher $H_2SO_4$ coating $wt\%$ to achieve the same equivalent coating coverage as that of a smaller

aggregate. Soot particles with a small mobility size contain less pore volume and can be more easily covered by $H_2SO_4$ coatings, thus their PCF activation is readily inhibited and they freeze homogeneously like a $H_2SO_4$ droplet with a soot particle core. It is safe to extrapolate that further $H_2SO_4$ coating for 200 nm FW200 soot particles will totally depress its PCF freezing tail (at 218 K) and the coated particle will freeze only homogeneously according to Koop et al. (2000). Comparably, 400 nm FW200 soot particles contain more abundant pore structures which are not inhibited to PCF even upon excessive $H_2SO_4$ coating. For

the same $H_2SO_4$ coating $wt\%$ more PCF active pores are available in 400 nm FW200 soot particles than in 200 nm particles. This again implies that the distribution of $H_2SO_4$ onto soot particle surfaces is not uniform or the $H_2SO_4$ coating states are different over soot particle local structures.

### 3.2.1 Summary of coated soot ice nucleation

To sum up, we propose three possible scenarios to explain the $H_2SO_4$ coating effects on mCASTblack and FW200 soot particle IN activities. Firstly, $H_2SO_4$ may fill in and block some mesopores of the soot-aggregate making it PCF inactive at low $T$ because the subsequent pore water capillary condensation into $H_2SO_4$ filled pores will form concentrated $H_2SO_4$ solutions with a low homogeneous nucleation rate (Koop et al., 2000). This can be attributed to thin or medium $H_2SO_4$ coating $wt\%$ scenario for mCASTblack. Meanwhile, if the soot particle is porous and contains sufficient pore structures, some mesopores can remain

unfilled and are still able to induce PCF such as for thin and medium coatings of 200 nm FW200 and even for thick coating of 400 nm FW200. Secondly, most of the mesopores are affected by $H_2SO_4$. At low RH conditions, mesopores filled with $H_2SO_4$ solution require a higher saturation condition to freeze, in order to recover to high enough water activity and to compensate the homogeneous nucleation rate depression. At higher RH conditions near homogeneous freezing condition, the depressed PCF mechanism is not active and so these coated particles freeze homogeneously. The second scenario can help with the

understanding of thick $H_2SO_4$ coating effects on mCASTblack particles and 200nm FW200 particles. Finally, excessive $H_2SO_4$ coating tends to block all pores and shuts down the PCF mechanism. In this case, $H_2SO_4$ might form a shell on the particle





surface or encapsulate the particle so the coated particle behaves as a $H_2SO_4$ droplet and only freezes via homogeneous freezing. We suggest the last scenario is relevant to soot particles with high $H_2SO_4$ coating $wt$ % (> 100 %). Detailed descriptions about these three coating states will be discussed in Sect. 3.3.2.

### 3.3 Typical internally mixed states of soot particles and sulphuric acid

Coupling single particle size and mass measurement results with microscopy of the particle mixing state characterization results, as well as with the ice nucleation activities of $H_2SO_4$ coated particles with different coating $wt$ %, a hypothesis depicting the internal mixing states of $H_2SO_4$ and soot particles is proposed. In this section, particle microscopy characterization results will be presented first, and next, a three-step process for soot particle $H_2SO_4$ coating will be explained to further understand particle ice nucleation activities shown in Sects. 3.1 and 3.2.

### 3.3.1 SEM, TEM and EDX results to characterize particle morphology

SEM results both for bare and coated soot particles are shown in Fig. 8. These SEM images are taken at 200k magnification for 400 nm mCASTblack and FW200 soot particles with $H_2SO_4$ coating $wt$ % similar to those coated soot particles in Figs. 5 and 7 and also termed as bare, thin and thick coatings. Images for the same soot samples at 20k magnification are provided in Fig. C1. As seen in Fig. 8a, bare 400 nm mCASTblack soot particle is fractal and looks like a long primary particle chain with lacy voids. Thinly coated 400 nm soot-aggregate (as shown in Fig. 8b) is close to the bare. Some primary particles form a chunk with thick $H_2SO_4$ coating and the shape of the primary particles is more ambiguous than in the bare aggregate, which can be seen in Fig. 8c (thick coatings). In addition, thickly coated 400 nm mCASTblack soot-aggregates tend to fold by showing a slight curvature and appears less fractal, compared to bare and thinly coated aggregates.

Figure 8d shows that a FW200 bare 400 nm soot-aggregate is less fractal than the same size bare mCASTblack soot-aggregate in Fig. 8a. This is in agreement with the particle fractal dimension results presented in Mahrt et al. (2018), where the authors reported a higher fractal dimension value for FW200 soot particle (2.35) compared with mCASTblack (1.86). There are no distinguishable morphological feature differences between Fig. 8d and e, which means thin $H_2SO_4$ coating does not result in a significant soot surface topography change. We believe that a small amount of $H_2SO_4$ first fills into pores among primary particles and is unable to modify the particle surface significantly. However, heavier mass coatings will have a part of soot-aggregate embedded into the $H_2SO_4$ material, as clearly visible in Fig. 8f. These images are comparable to the SEM images for ambient soot particles in a field study (Bhandari et al., 2019; see Fig. S4b). Besides, a ring of spray in Fig. 8f is demonstrated to be small $H_2SO_4$ droplets resulted from the impaction of soot-aggregates on the Cu grid, as seen EDX results (discussed below and see Fig. 11). Similar results about $H_2SO_4$ coating introduced morphology change for FW200 soot can also be seen from TEM images shown in Fig. C2f.





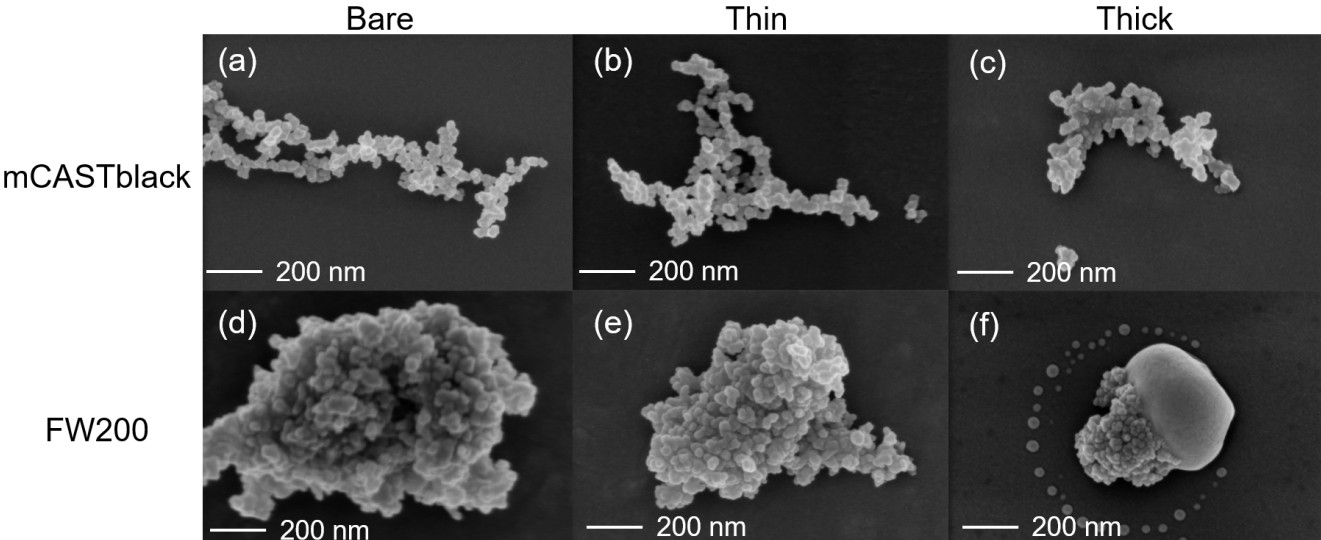

**Figure 8. SEM images (Zeiss Leo 1530, Signal=InLens, EHT = 3 kV) for 400 nm size selected bare and coated mCASTblack and FW200 soot particles. Scale bars are indicated in each image. (a) bare mCASTblack, (b) mCASTblack with a thin coating (coating $wt$ = 1.9 %), (c) mCASTblack with a thick coating (coating $wt$ = 31.1 %), (d) bare FW200, (e) FW200 with a thin coating (coating $wt$ = 1.8 %), (f) FW200 with a thick coating (coating $wt$ = 65.0 %).**

TEM images of 200 nm bare, thinly and thickly coated both for mCASTblack and FW200 soot particles are shown in Fig. 9. The coating $wt$ % is presented in Table 2 and similar to those of the ice nucleation experiments, as aforementioned in Sect. 2.2.3. There is no apparent morphology difference between bare and thinly coated aggregates for mCASTblack soot (as shown in Fig. 9a and b), which is consistent with the SEM results. However, the aggregate size in Fig. 9a and b is much larger than 200 nm. This could be an artifact during grid sample collection, which results in soot-aggregate piling or agglomeration when particles deposit onto the grid. Small aggregates can also coagulate while transporting in the aerosol flow (Kulkarni et al., 2011). In addition, soot-aggregates are of heterogeneous shapes and even size selected monodisperse soot aerosol spread a larger size distribution range than the selected size. Besides, a part of double-charged particles, which have a larger mobility size, also exist as already specified in Sect. 2.2.3 (see Fig. A3). Thus, these larger aggregates with higher inertia are more effectively collected by the ZEMI impactor. Nevertheless, the aggregate in Fig. 9a or b is already much larger than the size of a double-charged 200 nm aggregate (~ 320 nm). In order to support that the large aggregate sizes are an artifact of grid sampling, the optical size of suspended soot-aggregates for TEM grid collection are also tested by the OPC and the results are presented in Appendix C, showing that the suspended aggregates are binned into the expected optical size channels and demonstrating their size selection is acceptable. The thickly coated mCASTblack aggregate in Fig. 9c shows a more compacted 2D projection shape, which suggests the thick coating increases primary particle connectivity resulting in some extent aggregate compaction. Similar to mCASTblack, thin $H_2SO_4$ coating for FW200 soot particle does not lead to distinguishable features, comparing Fig. 9d with Fig. 9e. Clearly, thick coating for FW200 soot particle results in a shrunken aggregate projection with the edge





boundary being smoother than the bare and thinly coated ones. This shrinkage is also explained by the decrease in mobility

size of soot particles with thick coatings shown in Fig. 3c. In addition, thick $H_2SO_4$ coating seems to show a darker visualization

effect for FW200 soot-aggregates compared to bare particles. In addition, TEM images provide evidence of $H_2SO_4$ presence

on soot particle surfaces, which is consistent with SEM images. In Fig. 9f, the watermark indicated by a red square between

the particle and the edge of the carbon support film is believed to be a $H_2SO_4$ footprint. During the electron microscopy it

could be observed how the structure, which looked initially like a balloon, slowly evaporated and finally left this shrivelled

skin. Evidence on the presence of $H_2SO_4$ on soot primary particle surfaces and $H_2SO_4$ pore filling effects from HR-TEM

images is presented in the Appendix, Figs. C4 and C5.

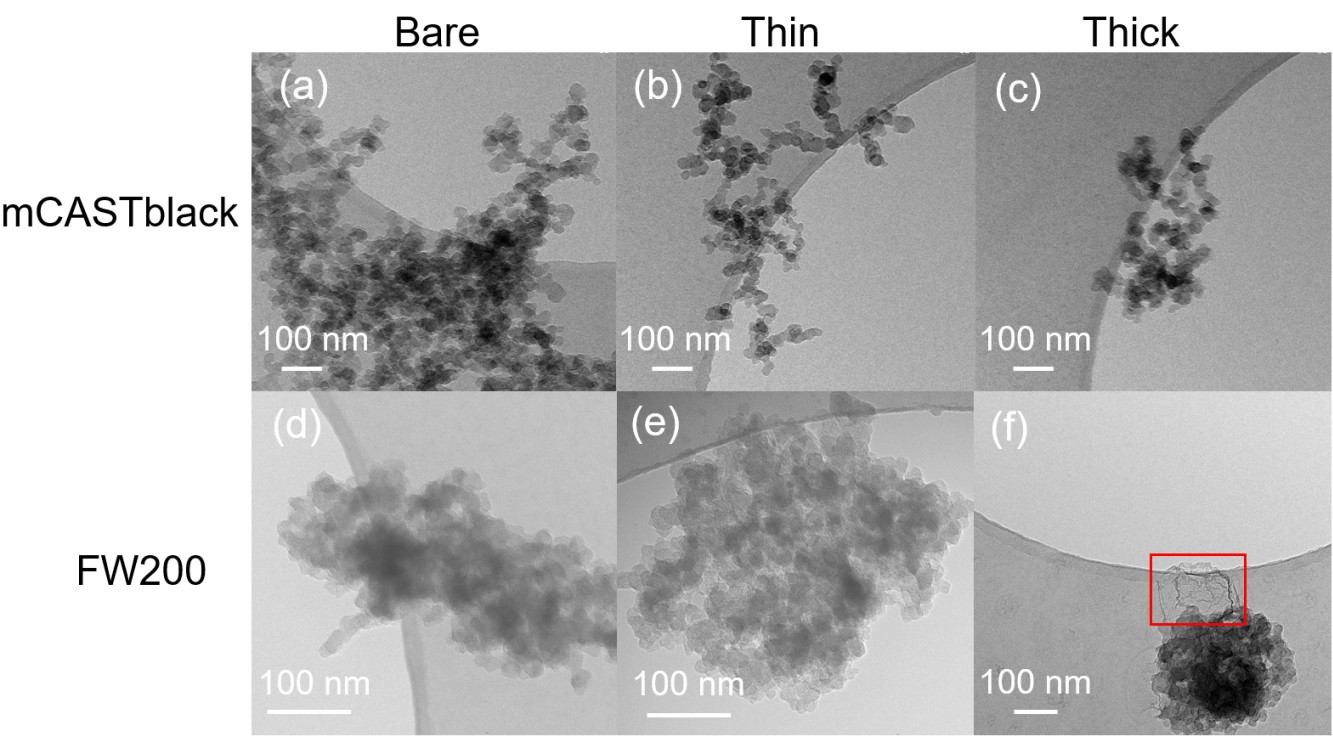

**Figure 9. TEM images for 200 nm size selected bare and coated mCASTblack and FW200 soot particles. Scale bars are indicated in**

**each image. (a) bare mCASTblack, (b) mCASTblack with a thin coating (coating _wt_ = 2.9 %), (c) mCASTblack with a thick coating**

**(coating _wt_ = 30.2 %), (d) bare FW200, (e) FW200 with a thin coating (coating _wt_ = 2.3 %), (f) FW200 with a thick coating (coating**

**_wt_ = 139.3 %). Microscopes used: (a)-(c) and (f) TFS F30, (d) and (e) Hitachi HT7700.**

EDX measurements demonstrate $H_2SO_4$ coating shows a non-uniform distribution on soot-aggregate surfaces. Grids with 200

nm mCASTblack and FW200 soot-aggregates are sampled. To indicate soot-aggregate surface $H_2SO_4$ distribution, composite

images in HAADF (High Angle Annular Dark Field) of 2D EDX elemental maps for S (red dots in Figs. 10a and 11a) are

used. With a focus on the particle surface chemical composition, C, O and S element contents for areas of interest (AOI) within





a soot-aggregate projection are quantified to obtain the spectra to avoid the influence of grid material, as indicated by the red square in Figs. 10a and 11a. The mass percentage of each element is normalized by the total result of these three elements and
attached in Figs. 10b and 11b. Background signal corrected element spectrum results are provided in Appendix C.

Comparing S maps for mCASTblack in Fig. 10a, the presence of $H_2SO_4$ on the soot is indicated by an increasing signal level with increasing coating thickness, however the coating material distribution is not uniform on the soot-aggregate surfaces. A radar chart in Fig. 10b shows quantitative chemical composition changes for $H_2SO_4$ coated mCASTblack soot. With $H_2SO_4$
coating, the S content increases as expected whereas the O content decreases, which is unexpected because $H_2SO_4$ contains a significant proportion of oxygen. This may suggest that there is a reaction between mCASTblack soot and $H_2SO_4$ depleting some O content in the process, considering $H_2SO_4$ as a strong oxidant with respect to mCASTblack soot (Mahrt et al., 2020b). From the table in Fig. 10b, even though thick coating has a $H_2SO_4$ coating mass percentage by more than 30 %, only a tiny amount of S is detected by the EDX measurement over AOI. It is feasible that the soot local surface (i.e. AOI) detected is with
a lower coating $wt$ % than the global soot particle, considering non-uniform $H_2SO_4$ coating distribution on the soot surfaces. Additionally, a part of $H_2SO_4$ coating material is lost during the measurement because of the heating effect from high energy electron beams, which also partly explains the lower $H_2SO_4$ content detected by EDX mapping. In Fig. 11a for FW200 soot, intensive signals on coated particle surfaces also suggest random distribution of coating material and that there are preferred sites for the interaction of soot and $H_2SO_4$. Watermarks attached with and around thickly coated soot-aggregate are shown to
be $H_2SO_4$ droplets. This finding coincides with the SEM result in Fig. 8f and TEM result in Fig. 9f. The normalized C, O and S mass ratio result clearly suggests the increment of $H_2SO_4$ coating material in terms of O and S and the decreasing share of soot material in terms of C (Fig. 11b). Contrary to coated mCASTblack soot particles, the normalized proportion of O in $H_2SO_4$ coated FW200 soot particles increases with higher $H_2SO_4$ coating $wt$ %, suggesting that $H_2SO_4$ coating induced surface reaction (or oxidation) for FW200 soot does not occur (or at least not to the same extent) as that for mCASTblack soot. This
can be attributed to the fact that FW200 soot is already surface oxidized during its production according to the information provided by the manufacturer.







| Black soot | C (%) | O (%) | S (%) |
|------------|-------|-------|-------|
| Bare | 89.5 | 10.5 | 0.01 |
| Thin | 91.3 | 8.7 | 0.02 |
| Thick | 92.1 | 7.8 | 0.10 |

**Figure 10. Panel (a) 2D EDX maps for sulphur (S) on 200 nm mCASTblack bare, thinly coated (coating $wt$ = 2.9 %) and thickly coated (coating $wt$ = 30.2 %) aggregates. Sulphur signal (red dots) is overlain on HAADF (high-angle annular dark-field) images. Scale bars are indicated in each image. Panel (b) the normalized carbon (C), oxygen (O) and S mass percentage of aera of interest (AOI, indicated as red square in panel (a)) for mCASTblack soot with different coating thicknesses.**







**Figure 11.** Panel (a) 2D EDX maps for sulphur (S) on 200 nm FW200 bare, thinly coated (coating *wt* = 2.3 %) and thickly coated (coating *wt* = 139.3 %) aggregates. Sulphur signal (red dots) is overlain on HAADF (high-angle annular dark-field) images. Scale bars are indicated in each image. Panel (b) the normalized carbon (C), oxygen (O) and S mass percentage of aera of interest (AOI, indicated as red square in panel (a)) for FW200 soot with different coating thicknesses.

Overall, SEM images show that thick $H_2SO_4$ coatings can change soot-aggregate surface topography. The coating material either can be distributed over the soot surfaces unevenly and change the connectivity of adjacent primary particles or will pile up over primary particle clusters to form chunks and finally form a $H_2SO_4$ shell. TEM images show thin $H_2SO_4$ coatings do not change the morphology of soot-aggregates whereas thick $H_2SO_4$ coatings can modify their morphology significantly by inducing aggregate structure compaction. EDX 2D maps provide robust evidence on the presence of $H_2SO_4$ over soot-aggregates and further demonstrate that the $H_2SO_4$ distribution is not uniform. Comparing the thick coating results for 200 nm mCASTblack soot (Fig. 9c) to those of FW200 soot (Fig. 9f), coating effects on mCASTblack soot-aggregate morphology modification are different even though the coating equivalent *ML* values are similar ($3.4 \pm 0.4$ *ML* and $3.6 \pm 0.3$ *ML*, respectively, according to Table 2). Moreover, the SEM image with a higher 3.5 equivalent *ML* coating for 400 nm





mCASTblack aggregate (Fig. 8c) compared to that of a 400 nm FW200 aggregate with 1.7 equivalent *ML* coating (Fig. 8f)

does not show more compaction of mCASTblack particles. This different coating effect between mCASTblack and FW200 soot results from the differences in intrinsic soot morphology properties. Nonetheless, we expect that mCASTblack soot-aggregates can also be embedded into $H_2SO_4$ and appear to be as compacted as thickly coated FW200 if $H_2SO_4$ coating *wt%* is achieved, e.g. > 100 %. Therefore, the equivalent *ML* value is not a reliable proxy to describe the distribution of $H_2SO_4$ coating material on soot-aggregate surfaces.

**3.3.2 A three-step process for soot particle $H_2SO_4$ coating**

Based on the aforementioned soot particle ice nucleation activity and property characterization results, we hypothesize that the soot coating process presented here has three stages (as illustrated in Fig. 12). A soot-aggregate comprising of numerous primary particles has a complex intra-aggregate structure (Vander Wal et al., 2007; Bond et al., 2013), a sketch of which is shown in Fig. 12a. We first start with a small amount of $H_2SO_4$ coating (Fig. 12b) where $H_2SO_4$ molecules fill into pores

among primary particles or voids in the soot-aggregate because of the capillary force but not all pores will be filled completely. Meanwhile, $H_2SO_4$ molecules can form clusters (humps or small islands) on the surfaces. At this point, the particle size is not expected to change as supported by the single particle mass and size results shown in Fig. 3. The particle effective density increases following a smooth slope, showing the gain of $H_2SO_4$ as the density of bare soot particles is smaller than that of pure $H_2SO_4$. In general, the overall coating effect of the first stage is pore filling and if any, some aggregate shrinking.


Further increase in coating mass results in a more complete pore filling and the spread of $H_2SO_4$ on aggregate surfaces. As depicted in Fig. 12c, at this stage of the coating process, particle size growth starts to be detectable. At this stage, the particle effective density may increase by a different manner depending on soot types. In Fig. 3, mCASTblack soot particle effective density does not change whereas FW200 soot effective density continues to increase. This is because mCASTblack soot is

more fractal or lacy (Figs. 8 and 9) and $H_2SO_4$ may form molecule clusters or branches sticking on the particle surfaces. Hence, the size growth effect overwhelms the mass increase effect. On the contrary, FW200 soot is more compact and has abundant pore structures, thus $H_2SO_4$ can be accommodated inside the pore volume. Thus, the addition of $H_2SO_4$ for FW200 soot shows less size growth effect than for mCASTblack soot. Therefore, the main effect of the second stage in $H_2SO_4$ coating process is further pore filling with coating material spreading to the surfaces. Very likely, $H_2SO_4$ shows preference to the soot particle

surfaces and will firstly stick to some special sites on the surface before forming molecule clusters. For example, Garland et al. (2008) used SEM and atomic force microscopy (AFM) to investigate how the oleic acid deposits on soot particle surfaces and suggested that oleic acid islands with a height of several nano-meters will be accumulated on the surface rather than form a uniform oleic acid coverage layer.

Finally, once pores are filled and the growth of $H_2SO_4$ cluster on the surface continues, the growing $H_2SO_4$ clusters will interact with each other or collapse onto the particle surfaces, or neighbour islands start to merge into big patches. This process induces





collapse also of the particle structure causing particle compaction, as shown in Fig. 12d. Consequently, the third mixing state results in the particle shrinkage and finally forming $H_2SO_4$ encapsulated particles with a soot core if the coating is thick enough. This argument is supported by the dramatic size shrinkage of thickly coated soot particles (see Figs. 8f and 9f) in SEM and

TEM images of particles embedded or partially embedded in $H_2SO_4$ droplets. Such a mechanism is also supported by some work in the literature. For example, fractal soot agglomerates can be compacted by the condensation of ozonolysis products of α-pinene secondary organic aerosol (SOA) with the bare soot particle serving as a nucleation centre (Saathoff et al., 2003). Khalizov et al. (2009) also pointed out that small soot-aggregates gain size growth but larger aggregates show a dramatic size shrinkage due to structure compaction after being exposed to gaseous $H_2SO_4$ for more than 10 s. Most recently, Pei et al. (2018)

investigated the morphological changes of soot particles after $H_2SO_4$ or limonene ozonolysis SOA coating, and suggested that the coating material will change soot particle nanostructure by a two-step process. Firstly, external coating material will fill into the voids or pores among soot-aggregates. Next, the particle size will grow with further vapor condensation when the voids are filled completely. In brief, our three-stage hypothesis has robust support from measurements in this study and is also in good agreement with previous findings in the literature.

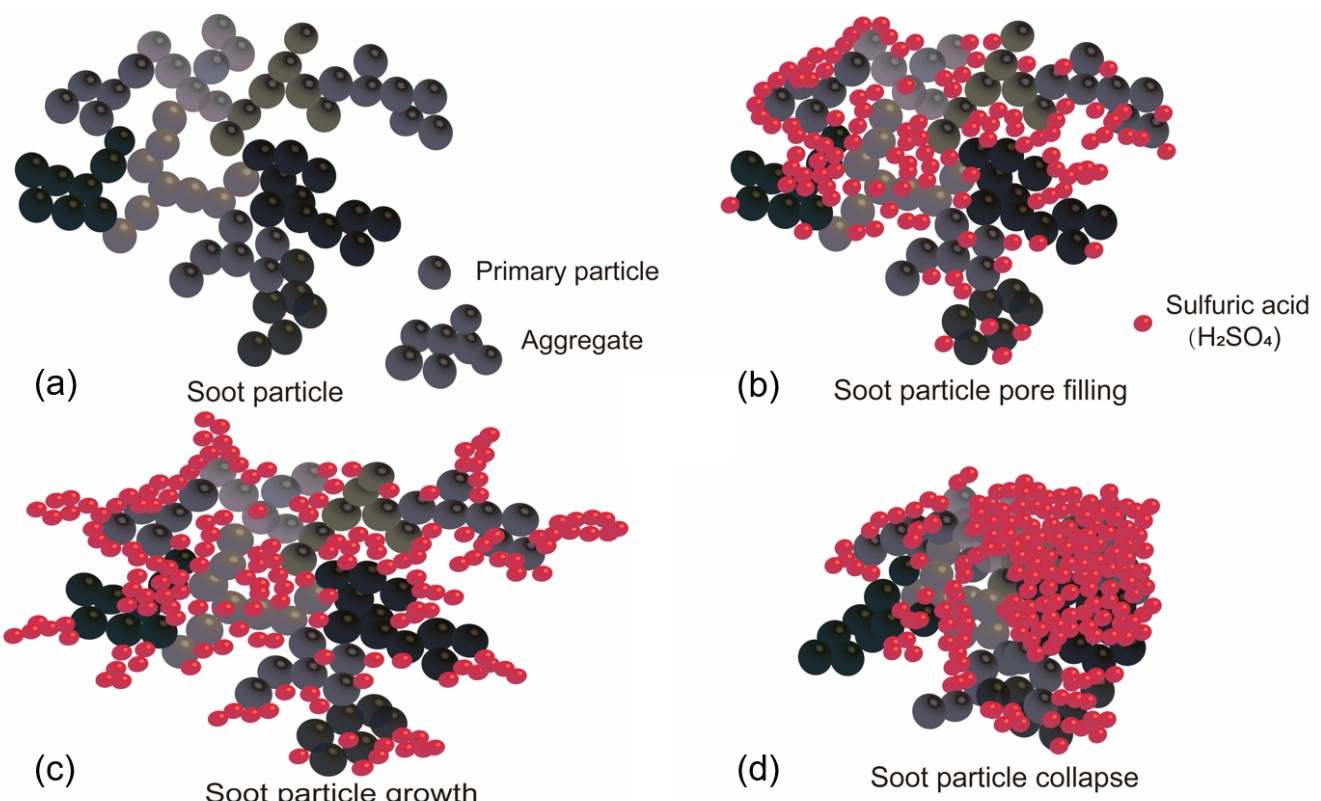


**Figure 12. Illustration of the proposed $H_2SO_4$ coating processes for soot particles (not to scale). Bare soot particle (a), soot particle pore filling and surface coating induced by progressive increase in $H_2SO_4$ coating mass (b, c, d).**





## 4 Atmospheric implications

Soot particles in the troposphere can affect cirrus cloud formation. $H_2SO_4$ coatings have clear suppression effect on soot particle ice nucleation based on the extent of the coating and the soot particle properties, with compact and porous particles being able to resist suppression in ice nucleation more than fractal particles. The difference in this suppression effect is because of the higher availability of mesopores in compact soot particles. Given the long lifetime (from ~ 5 days global mean up to 3 weeks) of soot particles in the atmosphere (Liu, 2005; Shen et al., 2014; Lund et al., 2018; Liu et al., 2020), the processes soot particles undergo with respect to ageing and cloud processing can be complex (Kanji et al., 2017). Due to the ubiquitous nature of $H_2SO_4$ in the troposphere (Pye et al., 2020), acid coatings are likely atmospheric ageing processes for soot particles. Hence, it would be important to consider the sulphate-soot mixing state when evaluating the impact of soot on cirrus cloud formation. As demonstrated in this study, soot particles internally mixed with $H_2SO_4$ exhibit a suppressed ice nucleation ability depending on their morphological properties. Thick $H_2SO_4$ coatings ($wt > 100$ %) effectively make the soot ice nucleation contribution to cirrus clouds formation comparable to homogeneous freezing of solution droplets. Therefore, the soot-sulphate mixing state impact on ice nucleation should be considered in predicting the impact of soot particles on cirrus formation, as well as when evaluating the radiative forcing and climate impacts of ice clouds in upper troposphere (Lohmann et al., 2020). In other words, using soot proxies without sulphate coatings may only be relevant for regions where soot is quickly incorporated into the cloud upon emission. Field studies detected soot-aggregates with 0.1-1 μm sizes in ice crystal residuals sampled from cirrus clouds (Petzold et al., 1998; Twohy and Poellot, 2005; Cziczo and Froyd, 2014) and the presence of 100-800 nm BC particles in the interstitial aerosol samples increases with increasing altitude from 8 to 11 km in the cirrus cloud regime (Petzold et al., 1998). Thus, we believe that the results presented in this study are of atmospherically relevant sizes (200 and 400 nm).

Considering that mCASTblack soot represents fractal and hydrophobic soot (Marhaba et al., 2019) and as an appropriate proxy for aviation soot particles (Ess and Vasilatou, 2018), $H_2SO_4$ coated mCASTblack soot particles in this study can be comparable to aviation soot particles coated by $H_2SO_4$ in the atmosphere. At $T <$ HNT, $H_2SO_4$ coated aviation soot particles may not inhibit or compete with droplet homogeneous freezing. This is because bare and $H_2SO_4$ coated 200 and 400 nm mCASTblack particles freeze at conditions for homogeneous freezing of solution droplets. Given the minor ability of ice formation for mCASTblack soot via PCF even at 218 K addressed in this study, uncoated aviation soot analogous to fractal and hydrophobic mCASTblack may not make significant contribution to cirrus cloud formation via PCF below homogeneous freezing conditions. Moreover, the PCF activation of mCASTblack soot particles with a low $H_2SO_4$ coating $wt$ % will be further depressed and require RH > $RH_{hom}$ conditions to form ice crystals. This implies that aviation soot particle proxies even with low $H_2SO_4$ coating mass (similar to our mCASTblack soot low $H_2SO_4$ coating $wt$ %) could only form ice crystals above $RH_{hom}$ conditions at $T <$ HNT.

Initially active soot particles (similar to our FW200 soot sample), if not extensively mixed with $H_2SO_4$ in the atmosphere, only demonstrate a slight suppression in ice nucleation ability at cirrus relevant temperatures ($T <$ 233 K). Thus, these coated





particles still nucleate ice via PCF forming ice crystals at RH < RH$_{hom}$ conditions, with the potential to perturb the cirrus background. Thickly coated soot particles, freezing homogeneously, are also atmospherically relevant. Bhandari et al. (2019) collected ambient soot particles under different weather conditions in a valley and reported that most ambient soot particles are covered with a thick coating or are embedded in external materials. If these thickly coated soot particles can be activated

as droplets and transported to high altitudes at lower $T$ conditions, it is likely that ice crystals can be induced by them via homogeneous freezing. In water supersaturated conditions at $T$ < HNT, it is reported that the homogeneous freezing of water droplets nucleated by soot particles is the dominant ice crystal formation pathway for cirrus clouds (Kärcher and Yu, 2009). Consequently, the water interaction enhancement of soot particles with thick H$_2$SO$_4$ coatings in this study resembles this case. According to Ditas et al. (2018), BC particles emitted by wildfires can reach the lowermost stratosphere and the authors

suggested that a BC particle with a 120 nm average mass equivalent diameter can be coated by volatile organic compounds (VOCs) and/or H$_2$SO$_4$ with a thickness of 150 nm. In this case, our findings imply that homogeneous freezing will be the dominant mechanism for ice nucleation as even the very active FW200 soot loses their ice nucleation ability with such a thick H$_2$SO$_4$ coating. As such, knowing the ageing history (trajectory) of soot particles in the upper troposphere would be crucial to accurately determine the role of soot particles in cirrus cloud formation. Mahrt et al. (2020a) suggested that cloud processing

induced aggregate compaction of soot samples significantly enhances their ice nucleation ability in the cirrus cloud regime. However, if the compaction arises from thick coatings of H$_2$SO$_4$ no enhancement or even a suppression of ice nucleation ability is to be expected in the cirrus cloud regime.

**5 Summary**

The ice nucleation ability of two samples, an organic carbon poor (mCASTblack) and a porous commercial carbon black

(FW200) soot were investigated. The soot particles coated with H$_2$SO$_4$ to different degrees are systemically studied at $T$ = 243-218 K under RH conditions from ice saturation to well above water supersaturation. Auxiliary measurements for soot sample property characterization, including particle mobility size and mass, H$_2$SO$_4$ coating $wt$ %, soot-aggregate morphology and surface chemical composition, were performed. Overall, H$_2$SO$_4$ coating on soot particles can modify soot-aggregate morphological properties depending on the coating $wt$ % and enhance soot-water interaction ability, thereby impacting soot

particle ice nucleation abilities. The C$_3$H$_8$ flame fuel-lean soot (mCASTblack), viewed as an aviation soot proxy in laboratory studies, is originally a poor INP at $T$ < HNT. H$_2$SO$_4$ coating makes 200 and 400 nm mCASTblack soot particles form ice crystals via homogeneous freezing and low mass H$_2$SO$_4$ coating even depresses their ice nucleation activity at $T$ < HNT to require RH > RH$_{hom}$ conditions. Porous FW200 bare soot particle and FW200 soot particles with thin or moderate H$_2$SO$_4$ coating can nucleate ice crystals via PCF below homogeneous freezing conditions at $T$ < HNT. However, thick H$_2$SO$_4$ coatings

exert a size dependent effect suppressing small size (200 nm) FW200 soot PCF activation more than that of larger size particles (400 nm). Thickly coated 200 nm particles freeze homogeneously requiring a smaller H$_2$SO$_4$ coating $wt$ % than that for 400





nm particles. Based on above results, we propose a three-step $H_2SO_4$ coating process on soot-aggregates with increasing $H_2SO_4$ coating masses, inclusion of pore filling, $H_2SO_4$ spreading over soot particle surfaces and soot particle structure collapse.

As demonstrated in this study, $H_2SO_4$ coating changes soot particle morphology and modifies its water interaction ability simultaneously. Differentiating these two coating effects to identify the dominant effect on ice nucleation is still an open question for future studies. In addition to $H_2SO_4$ coating, atmospheric ageing altering soot particle ice nucleation abilities can include chemical oxidation, organics coating and clouding processing in the atmosphere. Only a small proportion of studies in the literature have examined these topics systemically. Experimental studies on these soot ageing processes will benefit the

understanding of soot role in cirrus cloud formation.

## Appendix A The mixing state of soot and $H_2SO_4$

In order to ensure the absence of pure nucleated $H_2SO_4$ particles in the mixing state aerosol flow, the particle size distribution evolution of the aerosol in the coating apparatus was measured. Here, we present the internal mixing process of 200 nm

mCASTblack soot with a $H_2SO_4$ coating *wt* % = 15.6 % as an example. As shown in Fig. A1, particle size distribution for the 0-minute state of pure $H_2SO_4$ aerosol in the coating apparatus (red line) and 200 nm size selected bare soot particles (black dashed line) were conducted firstly and respectively. Afterwards, 200 nm mCASTblack bare soot aerosol was fed into the coating apparatus. The mixed aerosol particle size distribution was monitored until it reached a stable state (20-minutes black line, Fig. A1). In addition, a comparison between the aerosol particle mass distribution of bare soot particles and $H_2SO_4$ coated

particles was made to further demonstrate that pure nucleated $H_2SO_4$ particles are absent in the aerosol flow produced from the coating apparatus (Fig. A2).

Figure A1 depicts the particle size distribution evolution process of the mixing state of soot particles with $H_2SO_4$. At the very beginning of the measurement, pure nucleated $H_2SO_4$ aerosol shows a high particle number concentration and a small size

distribution with a mode peak at ~ 40 nm. After 200 nm bare soot particles are introduced into the coating apparatus for several minutes, the mixed aerosol particle number concentration dramatically decreases. With more time, the pure $H_2SO_4$ particle size distribution mode is absent and a size distribution peak around 200 nm becomes more distinct, indicating that the mixing state of soot particles and $H_2SO_4$ is changing over time. Finally, the size distribution of the mixed particles reaches a stable state as shown in the figure denoted by solid black line. Comparing the bare (dashed black line, Fig. A1) to well coated (solid

black line, Fig. A1) particle size distribution curve, $H_2SO_4$ coating in this case results in particle number concentration losses and a small size growth.



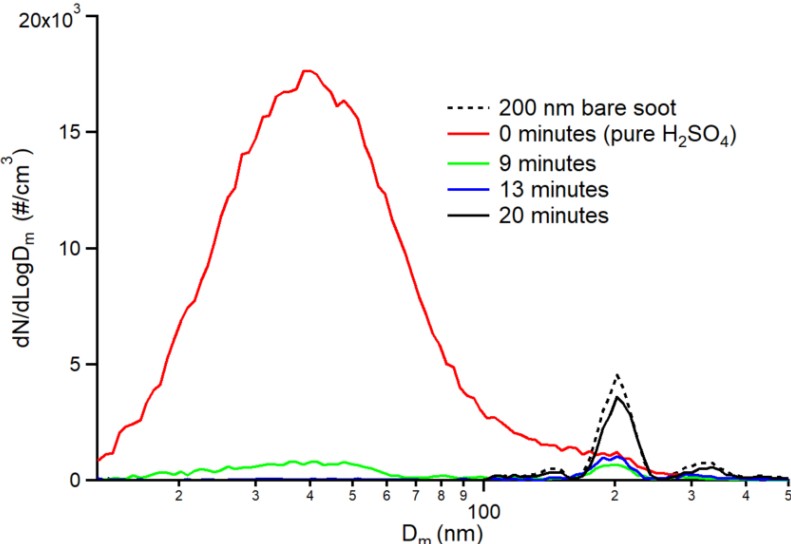

**Figure A1. The raw size distribution of the particles coming out of the coating apparatus in 20 minutes after 200 nm mCASTblack**
**soot aerosol sample flow is connected to the coating apparatus. The red line denotes the initial state of H₂SO₄ aerosol, where only**
**nucleation mode H₂SO₄ exists in the coating system without soot particles. The dashed black line indicates bare 200 nm black soot**
**particles. The green line presents the particle size distribution 9 minutes later when the soot particles were fed into the coating**
**apparatus. The blue line shows the particle size distribution 13 minutes later. The black line stands for the stable mixing state of**
**soot particle with H₂SO₄.**

Figure A2 shows the mass distribution of 200 nm bare and H₂SO₄ coated mCASTblack soot particles. The larger mass
distribution mode value of coated soot particles doubtlessly demonstrates the adsorption/condensation of H₂SO₄ on the soot-
aggregates. The mass distribution curve also provides evidence for the internal mixing state of soot particles with H₂SO₄, given
that the curve only shows one single distinct peak with a log-normal fitting peak value at 1.93 fg. It is conceivable that there
are no 200 nm pure H₂SO₄ particles in the aerosol flow because a 200 nm pure H₂SO₄ particle has a mass of 7.7 fg.


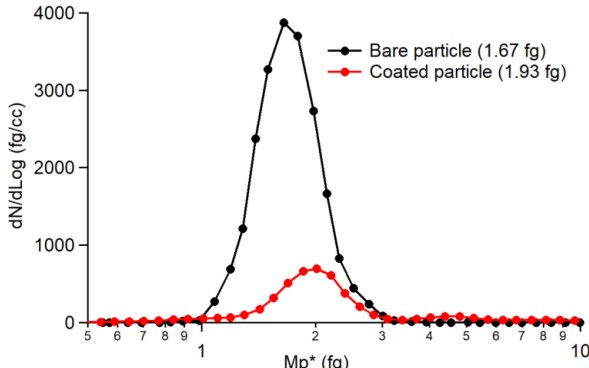





**Figure A2. The raw data mass distribution of 200 nm mCASTblack soot bare particles and H$_2$SO$_4$ coated particles. The black line shows the mass distribution of bare particles with a mass mode value of 1.67 fg derived from the log-normal distribution. The red line presents the mass distribution of H$_2$SO$_4$ coated particles with a mass mode value of 1.93 fg.**


Because of the heterogeneity of soot particles and the poor assumption for its spherical structure, DMA cannot perfectly select monodisperse soot particles based on the electrical mobility size. Therefore, we measured the number size distribution of the size selected bare soot particles using an SMPS system with multiple charged particle correction, downstream of the DMA. The DMA and SMPS system were operated with the same configurations as for all ice nucleation experiments. For 200 nm

soot particles, the DMA was running with a sheath to aerosol flow ratio of 10 : 1 and the downstream SMPS system with a flow ratio of 4 : 1. The system was pulled by a CPC 3772 with a sample flow rate of 1 L min$^{-1}$. For 400 nm soot particles, the DMA sheath to aerosol flow ratio was 7 : 1 and SMPS system operating with a CPC 3776 in low flow mode (aerosol sample flow 0.3 L min$^{-1}$) had a sheath to aerosol flow ratio 6 : 1. As shown in Figs. A3 and A4, SMPS measurements are able to cover the double-charged size for 200 and 400 nm soot particles and there exists a significant proportion (> 15 %) of soot particles

with larger sizes than the size selection value.

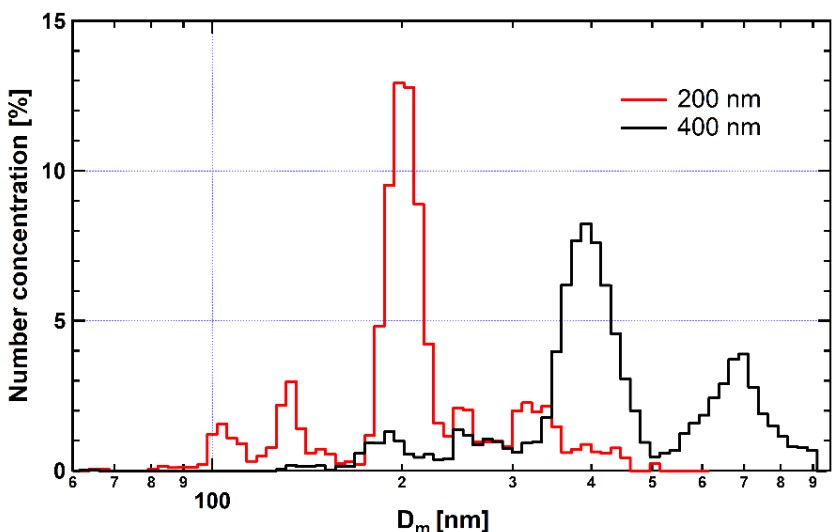

**Figure A3. Number size distribution in terms of particle concentration percentage as a function of mobility size (D$_m$) for 200 and 400 nm size selected mCASTblack bare soot particles.**





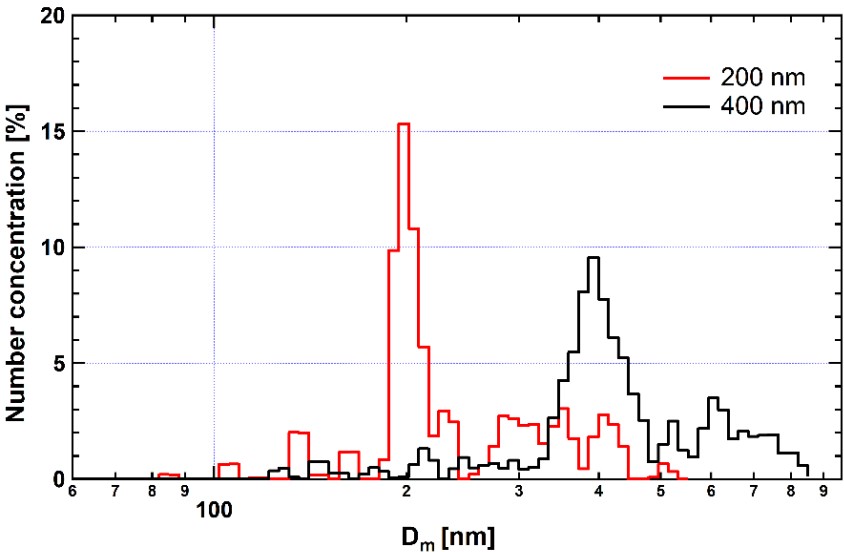

**795** **Figure A4. Number size distribution in terms of particle concentration percentage as a function of mobility size ($D_m$) for 200 and 400 nm size selected FW200 bare soot particles.**

**Appendix B The ice nucleation activity of bare and $H_2SO_4$ coated soot particles**

Soot particles coated by $H_2SO_4$ with progressively increasing coating masses were systematically investigated for their ice
**800** nucleation activities. The coating thickness was from less than a complete $H_2SO_4$ monolayer coverage up to several equivalent monolayers. Figures B1 to B4 present the AF plots derived from 1 μm OPC channel data of all the soot samples investigated as a function of RH ($RH_i$ and $RH_w$) and AF curves based on 5 μm OPC channel data are presented in Figs. B5 to B8. These figures show that at 243 and 238 K the signal in the 5 μm OPC channel at $RH_w < 105\%$ is absent, thus suggesting no ice crystals form at these temperatures and only water droplets are present in the 1 μm channel. This is because if ice can be
**805** nucleated at 243 and 238 K, the ice crystals grow sufficiently large to be reliably detected in the 5 μm channel for $RH_w < 105$ % but water droplets do not. As such, we can confidently conclude no ice nucleation for our soot particles at $T = 243$ and 238 K. For both mCASTblack and FW200 soot, at least eight $H_2SO_4$ coating $wt\%$ were tested for size selected soot particles of a size 200 or 400 nm.





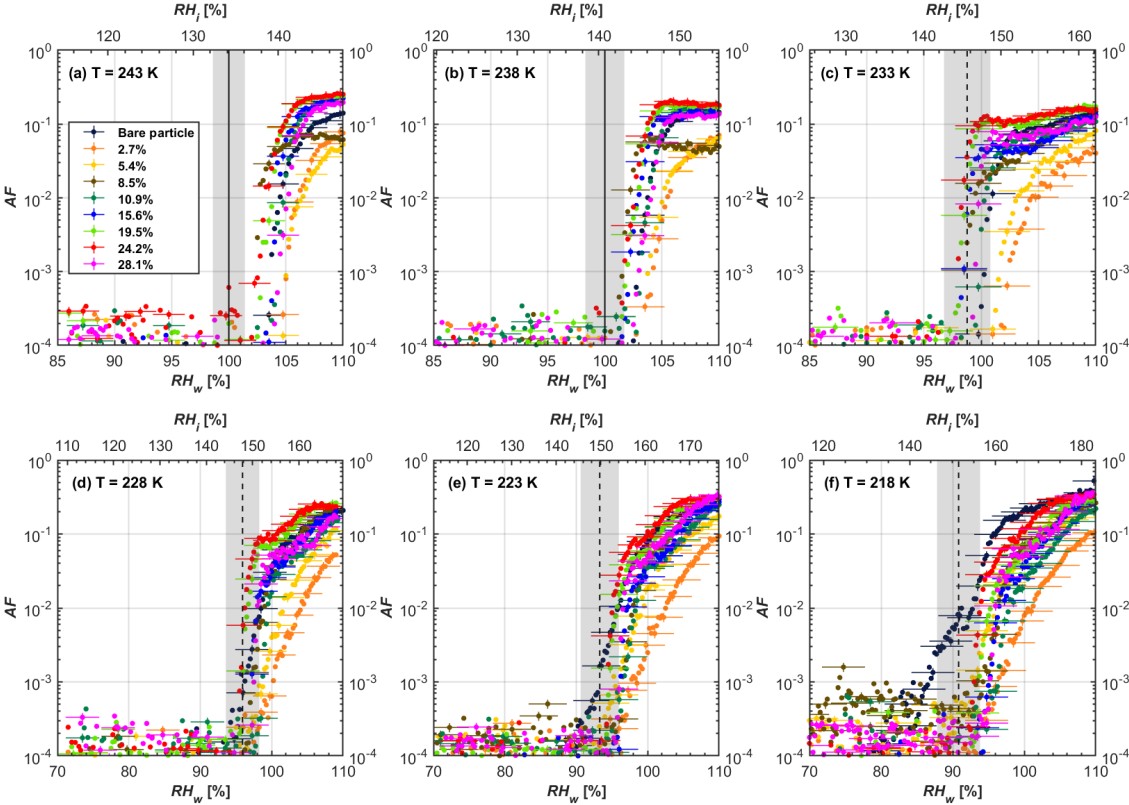

**Figure B1.** RH scans for bare and coated 200 nm mCASTblack soot particles at eight different coating $wt\%$ corresponding to 1 µm OPC channel, presented as AF as a function of RH. Black solid lines represent water saturation conditions according to Murphy and Koop (2005). Black dashed lines denote the expected RH values for homogeneous freezing at $T <$ HNT (Koop et al., 2000). The grey shading areas show the possible RH uncertainty calculated for water saturation and homogeneous freezing conditions. The percentage number represents the $H_2SO_4$ coating mass ratio to the bare particle mass.



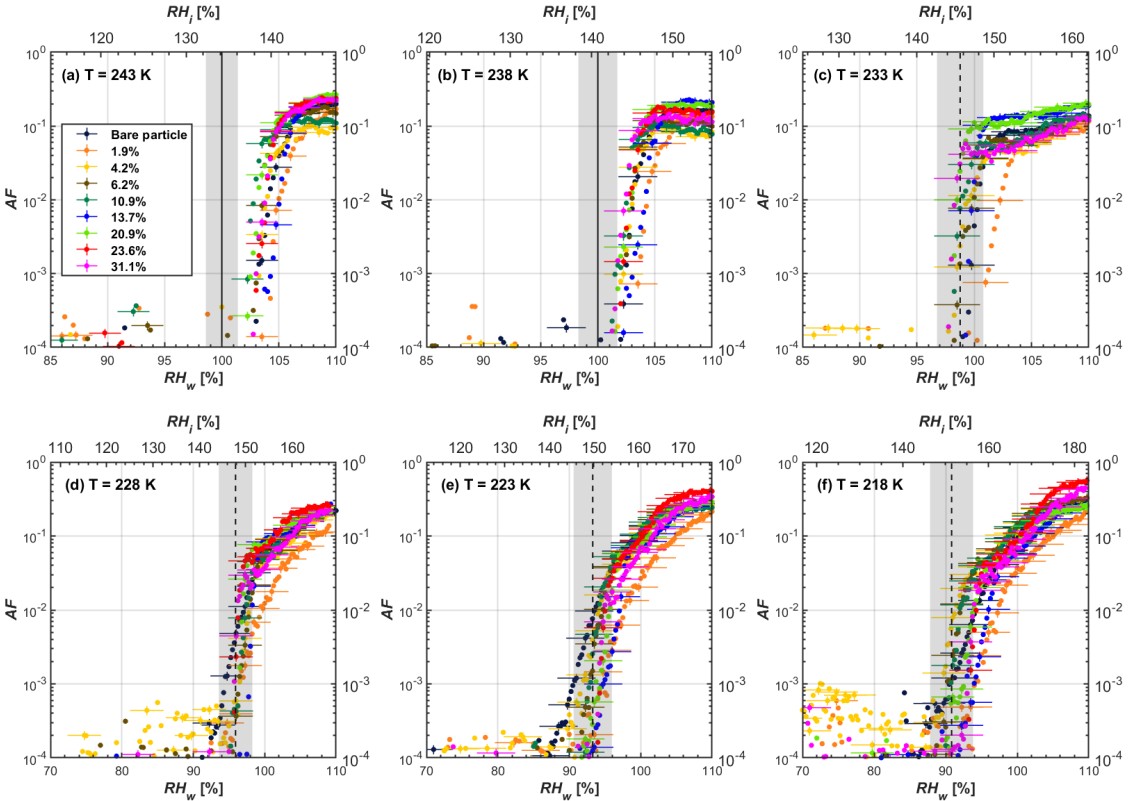

815

**Figure B2. RH scans for bare and coated 400 nm mCASTblack soot particles at eight different coating $wt\%$ corresponding to 1 μm OPC channel, presented as AF as a function of RH. Black solid lines represent water saturation conditions according to Murphy and Koop (2005). Black dashed lines denote the expected RH values for homogeneous freezing at $T <$ HNT (Koop et al., 2000). The grey shading areas show the possible RH uncertainty calculated for water saturation and homogeneous freezing conditions. The**
820 **percentage number represents the $H_2SO_4$ coating mass ratio to the bare particle mass.**



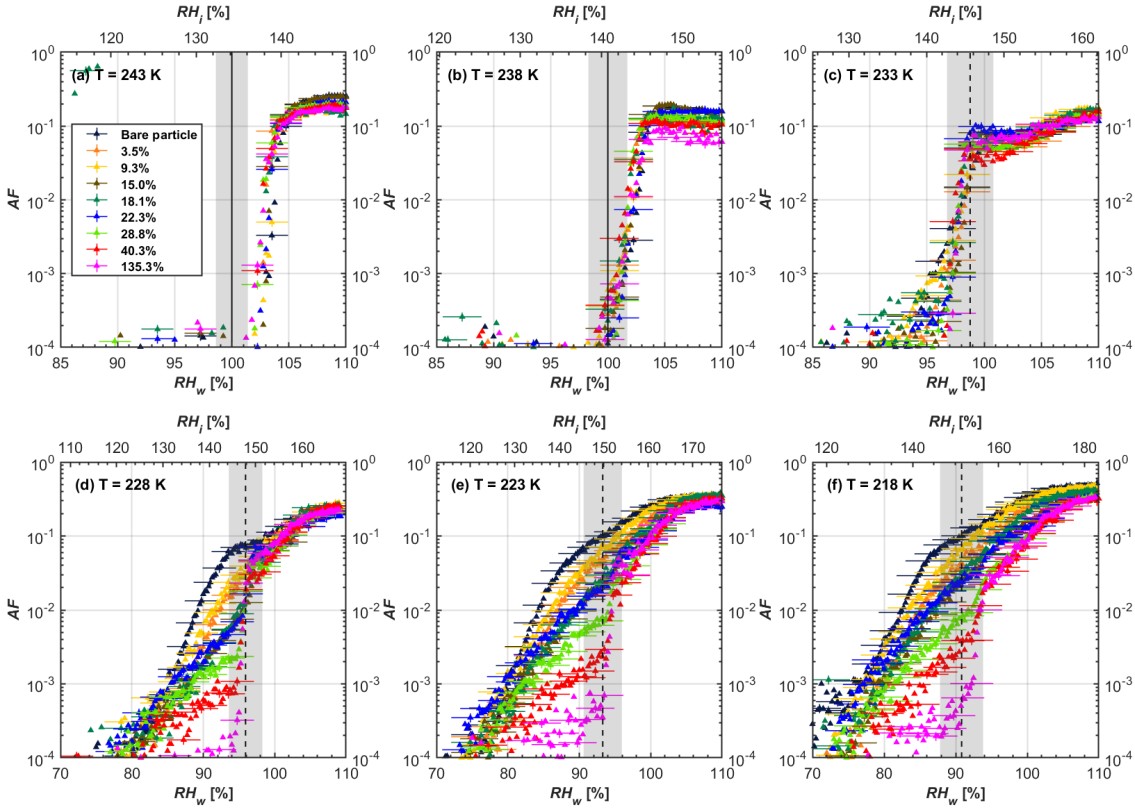

**Figure B3. RH scans for bare and coated 200 nm FW200 soot particles at eight different coating $wt$ % corresponding to 1 μm OPC channel, presented as AF as a function of RH. Black solid lines represent water saturation conditions according to Murphy and Koop (2005). Black dashed lines denote the expected RH values for homogeneous freezing at $T$ < HNT (Koop et al., 2000). The grey shading areas show the possible RH uncertainty calculated for water saturation and homogeneous freezing conditions. The percentage number represents the $H_2SO_4$ coating mass ratio to the bare particle mass.**



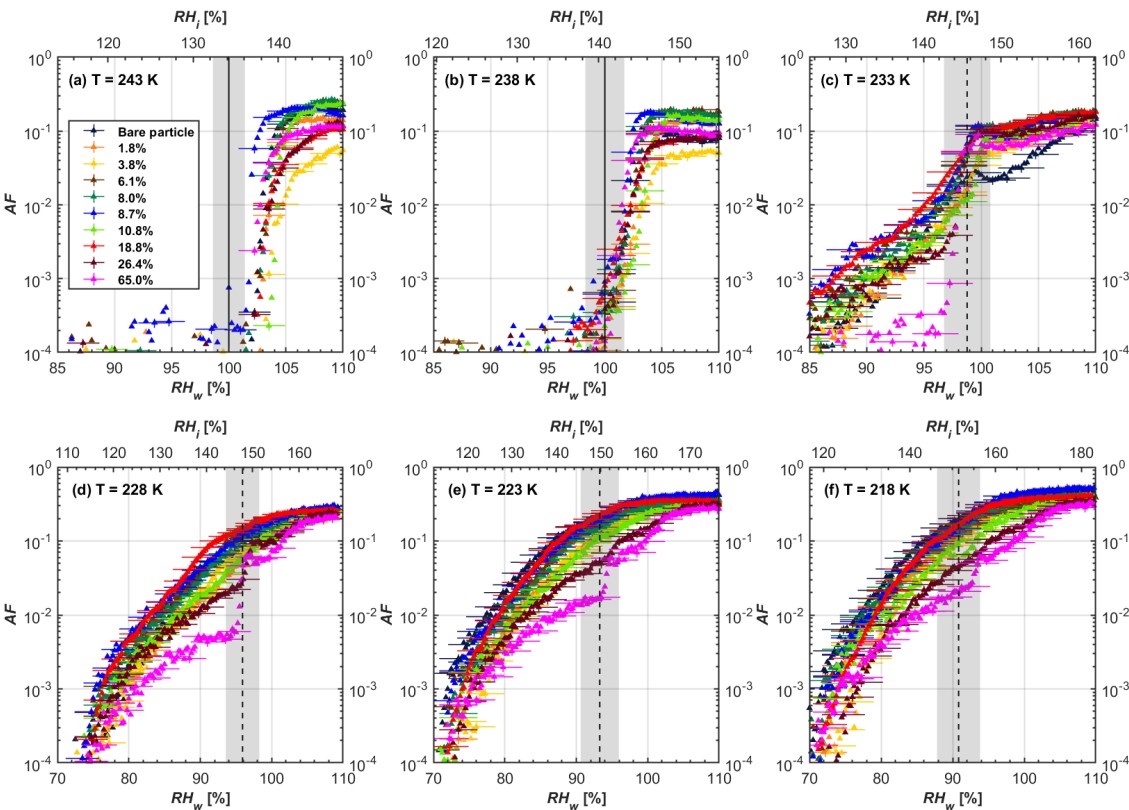

**Figure B4. RH scans for bare and coated 400 nm FW200 soot particles at nine different coating $wt$% corresponding to 1 μm OPC**
830 **channel, presented as AF as a function of RH. Black solid lines represent water saturation conditions according to Murphy and Koop (2005). Black dashed lines denote the expected RH values for homogeneous freezing at $T <$ HNT (Koop et al., 2000). The grey shading areas show the possible RH uncertainty calculated for water saturation and homogeneous freezing conditions. The percentage number represents the $H_2SO_4$ coating mass ratio to the bare particle mass.**





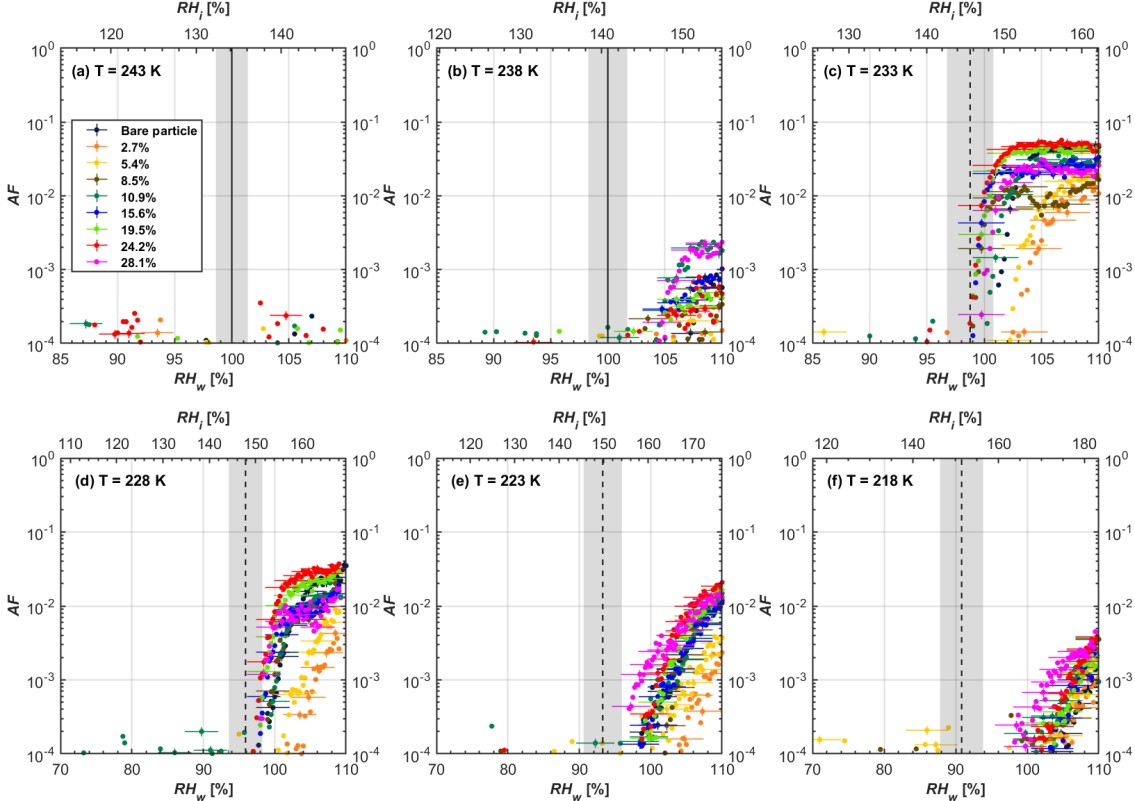

**Figure B5. RH scans for bare and coated 200 nm mCASTblack soot particles at eight different coating $wt\%$ corresponding to 5 μm OPC channel, presented as AF as a function of RH. Black solid lines represent water saturation conditions according to Murphy and Koop (2005). Black dashed lines denote the expected RH values for homogeneous freezing at $T$ < HNT (Koop et al., 2000). The grey shading areas show the possible RH uncertainty calculated for water saturation and homogeneous freezing conditions. The percentage number represents the $H_2SO_4$ coating mass ratio to the bare particle mass.**

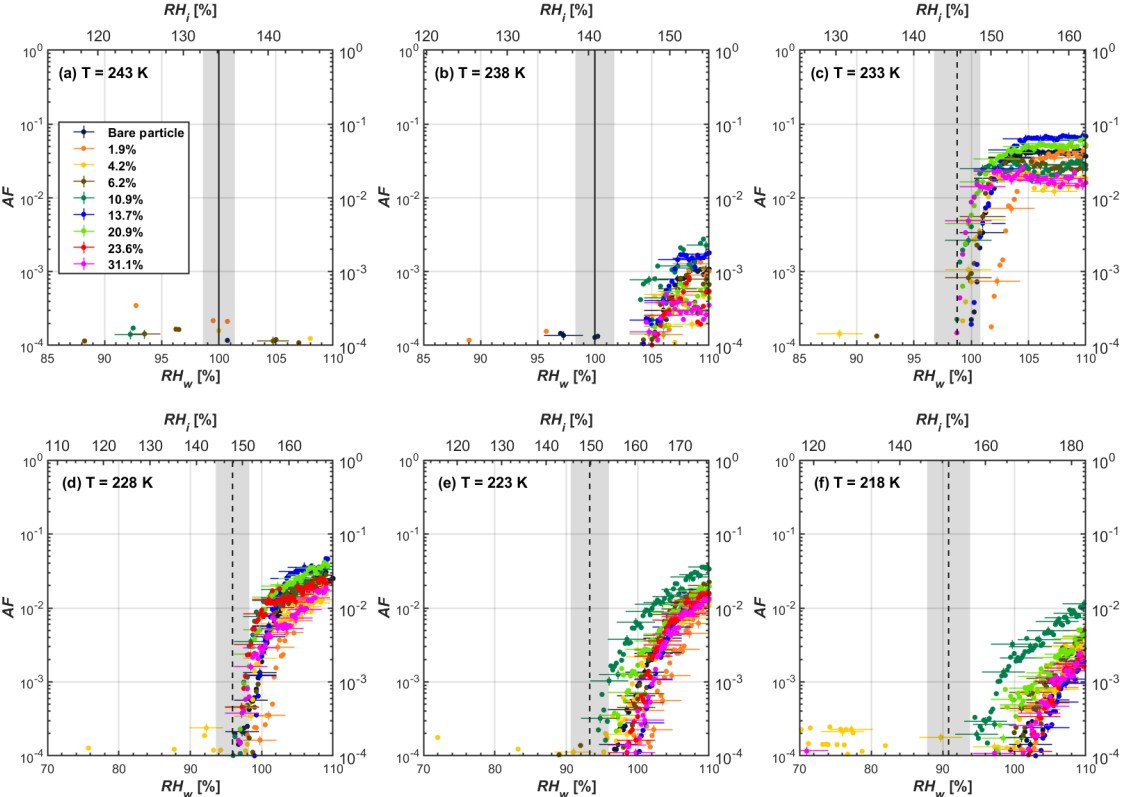

**Figure B6. RH scans for bare and coated 400 nm mCASTblack soot particles at eight different coating $wt\%$ corresponding to 5 μm OPC channel, presented as AF as a function of RH. Black solid lines represent water saturation conditions according to Murphy and Koop (2005). Black dashed lines denote the expected RH values for homogeneous freezing at $T$ < HNT (Koop et al., 2000). The grey shading areas show the possible RH uncertainty calculated for water saturation and homogeneous freezing conditions. The percentage number represents the $H_2SO_4$ coating mass ratio to the bare particle mass.**





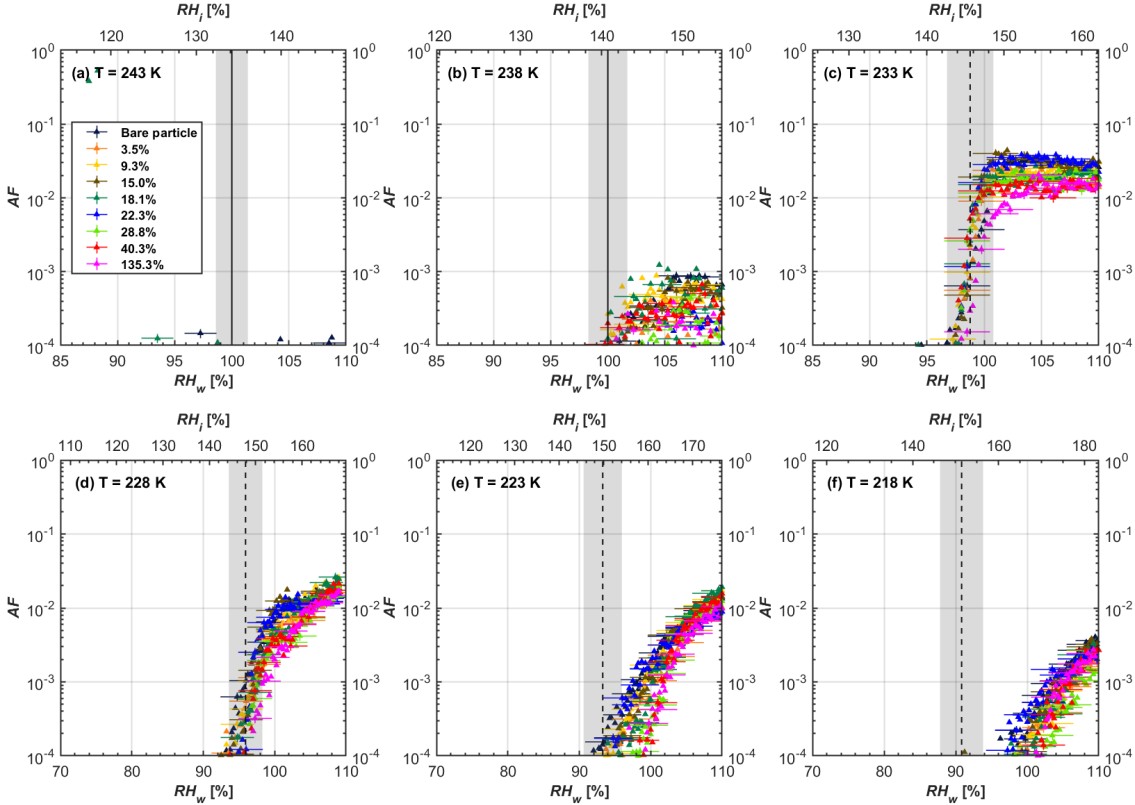

**Figure B7.** RH scans for bare and coated 200 nm FW200 soot particles at eight different coating $wt\%$ corresponding to 5 μm OPC
channel, presented as AF as a function of RH. Black solid lines represent water saturation conditions according to Murphy and
Koop (2005). Black dashed lines denote the expected RH values for homogeneous freezing at $T$ < HNT (Koop et al., 2000). The grey
shading areas show the possible RH uncertainty calculated for water saturation and homogeneous freezing conditions. The
percentage number represents the $H_2SO_4$ coating mass ratio to the bare particle mass.



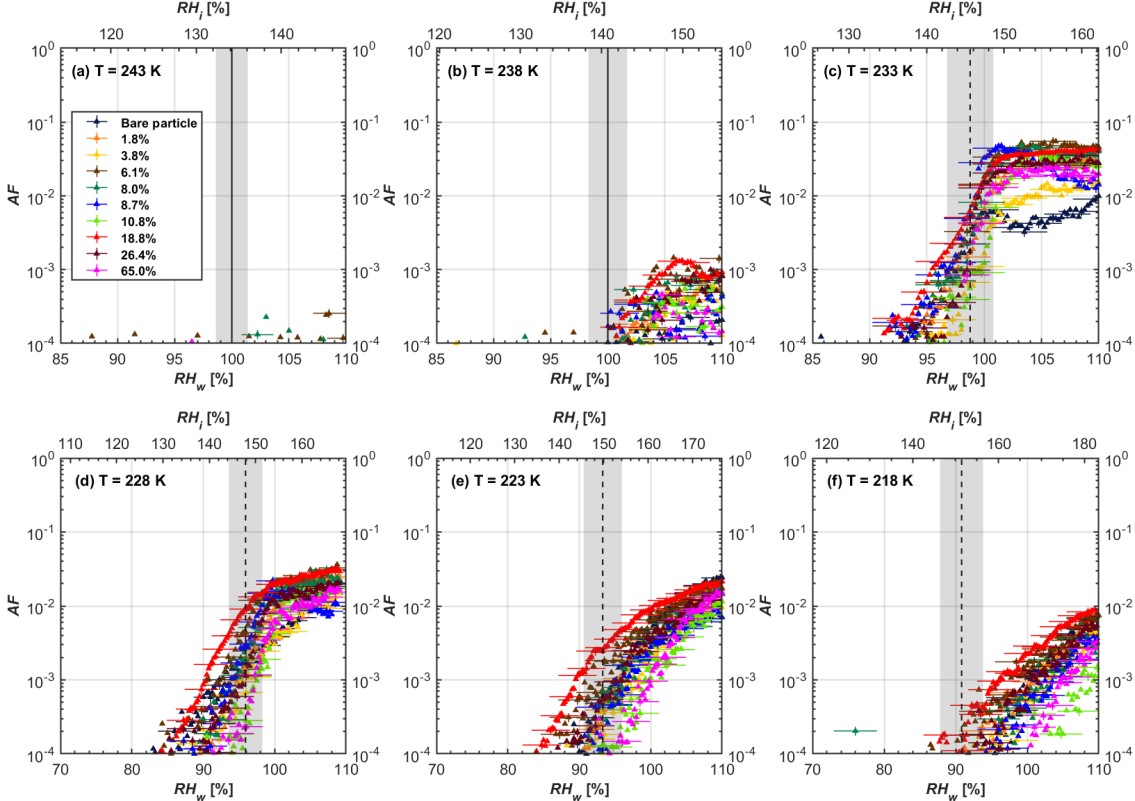


**Figure B8. RH scans for bare and coated 400 nm FW200 soot particles at nine different coating $wt$ % corresponding to 5 μm OPC channel, presented as AF as a function of RH. Black solid lines represent water saturation conditions according to Murphy and Koop (2005). Black dashed lines denote the expected RH values for homogeneous freezing at $T$ < HNT (Koop et al., 2000). The grey shading areas show the possible RH uncertainty calculated for water saturation and homogeneous freezing conditions. The**
**percentage number represents the $H_2SO_4$ coating mass ratio to the bare particle mass.**

## Appendix C Supplementary SEM, TEM and EDX results to characterize particle morphology

SEM and TEM images of low resolutions or with larger scale bars showing an overview of particle morphologies are presented in Figs. C1 (400 nm soot-aggregates) and C2 (200 nm soot-aggregates), respectively. The SEM images at 20k magnification
in Fig. C1 show that thickly coated soot particles, both for mCASTblack and FW200, are more compacted whereas thinly coated particles are similar to the bare ones. The same findings can be supported by TEM images at low magnification values in Fig. C2. Additionally, some small residues, surrounding single soot-aggregates in Fig. C2f, are believed to be small $H_2SO_4$ droplets generated during the impaction/drying of these soot-aggregates.





However, the visualized particles at low magnifications are larger than the particle sample selected size. As explained in Sect. 3.3.1, the most probable reason is the soot-aggregate agglomeration during impaction onto the grid, particularly when the particle concentration is high. This can explain why mCASTblack soot-aggregates in Fig. C1a, b and c present to be much larger than 400 nm, i.e. the size selection value. During TEM sample grids collection, the particle number concentration of 400 nm mCASTblack soot particles was as high as 3,000 cm$^{-3}$, providing a high probability for these mCASTblack soot-

aggregates to agglomerate into larger clusters (> 400 nm). In order to exclude other possible artifacts, the optical size of suspended 200 and 400 nm mCASTblack size selected bare soot particle sample was measured by the OPC, operated in a same manner as for HINC experiments. The measurement aerosol flow, also with a particle number concentration ~ 3,000 cm$^{-3}$, was just sampled upstream of ZEMI. Each OPC channel logged the number of particles larger than the channel threshold value in a 5 s interval. After measuring for 5 minutes, the percentage of 200 and 400 nm mCASTblack bare soot particles in different

OPC counting channels could be calculated. As shown in Fig. C3a, only 2.08 % of 200 nm size selected mCASTblack soot particles are detected with an optical size larger than 0.3 μm, suggesting larger soot-aggregates in Fig. 9a and b are agglomerates of 200 nm aggregates. Similarly, 400 nm size selected mCASTblack soot aerosol only contains 0.04 % of particles larger than 1 μm optical size whereas 99.96 % of the particles are of an optical size between 0.3 and 1 μm (see Fig. C3b). This further supports our claim that the super micron size soot particles shown in Figs. C1 and C2 are agglomerates of small soot-

aggregates.

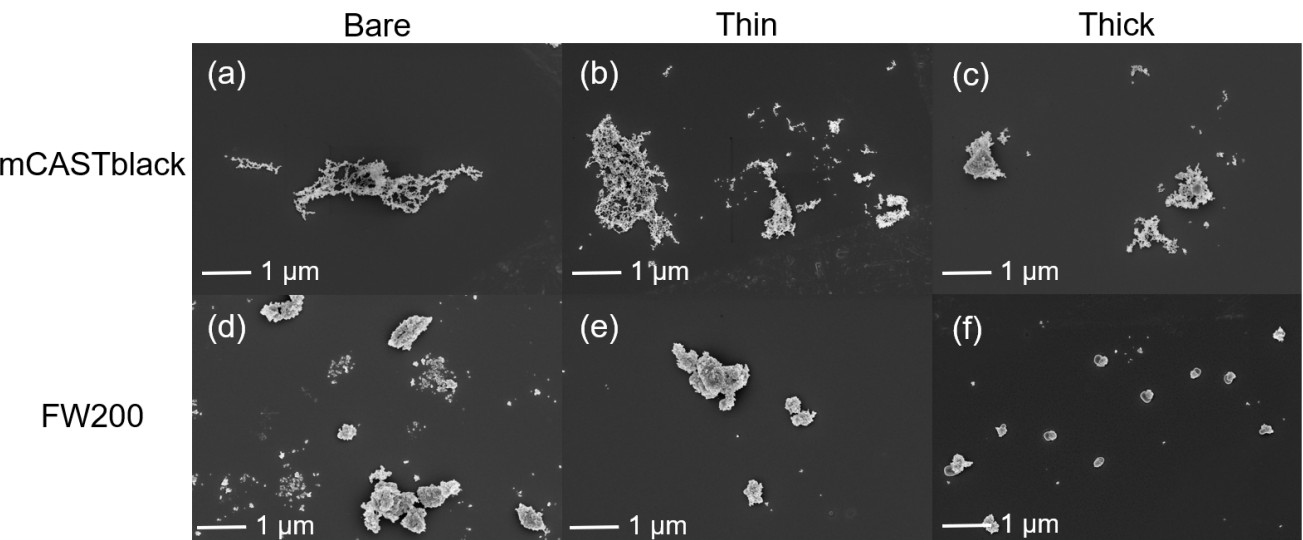

**Figure C1. SEM images (Zeiss Leo 1530, Signal = InLens, EHT = 3 kV) for 400 nm size selected bare and coated mCASTblack and FW200 soot particles. Scale bars are indicated in each image. (a) bare mCASTblack, (b) mCASTblack with a thin coating (coating**

***wt*** **= 1.9 %), (c) mCASTblack with a thick coating (coating *wt* = 31.1 %), (d) bare FW200, (e) FW200 with a thin coating (coating *wt* =1.8 %), (f) FW200 with a thin coating (coating *wt* = 65.0 %).**





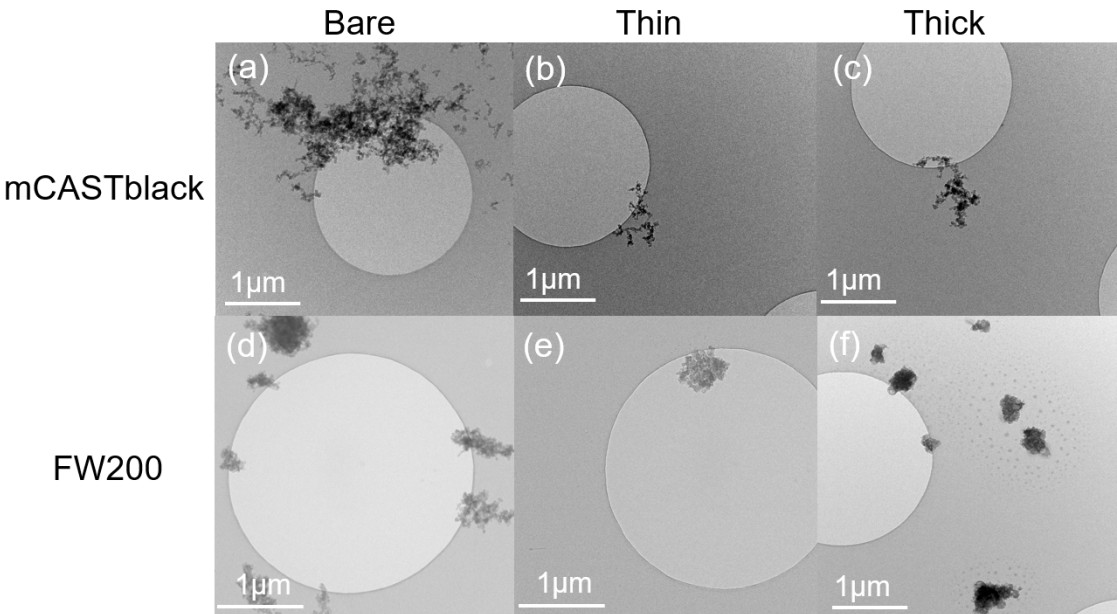

**Figure C2. TEM images for 200 nm size selected bare and coated mCASTblack and FW200 soot particles with a lower magnification as shown in each image. Scale bars are indicated in each image. (a) bare mCASTblack, (b) mCASTblack with a thin coating (coating**

**$wt$ = 2.9 %), (c) mCASTblack with a thick coating (coating $wt$ = 30.2 %), (d) bare FW200, (e) FW200 with a thin coating (coating $wt$ = 2.3 %), (f) FW200 with a thin coating (coating $wt$ = 139.3 %). Microscopes used: (a)-(c) and (f) TFS F30, (d) and (e) Hitachi HT7700.**

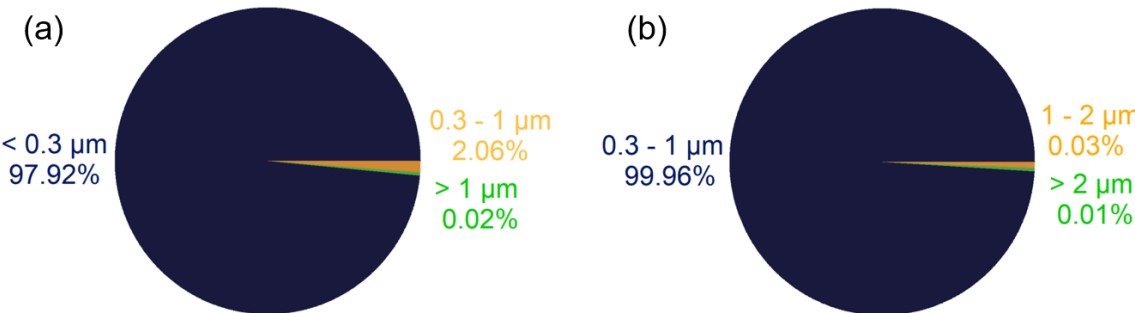

**Figure C3. The percentage of 200 (a) and 400 (b) nm size selected mCASTblack bare soot particles in three optical size ranges measured by the OPC operating in different counting channels, including 0.3, 1 and 2 μm.**

HR-TEM images are also collected to determine how the $H_2SO_4$ coating changes the fine structures of soot-aggregates and the network among primary particles, i.e. soot spheres showing a distinct onion-like structure comprising of graphitic layers in

Fig. C4. In order to avoid the background noise from the carbon film on the Cu grid, soot-aggregates hanging outside of the mesh edge (i.e. in the holes) were selected to obtain these HR-TEM images. The primary particle profiles in these 2D projected





soot-aggregate images are still distinguishable in Fig. C4d for bare and Fig. C4e for thinly coated soot particles, but single primary particle edges are more ambiguous in Fig. C4f for thickly coated soot particles. The similar results can be seen from Fig. C5, in which the outline of FW200 soot primary particles with thick coating (Fig. C5f) is smoother than those of bare and thinly coated particles (Fig. C5d and e). This can be attributed to the pore filling effect of $H_2SO_4$ coating, which fills in voids or wedges among primary particles and reduces the roughness of the soot-aggregate surface.

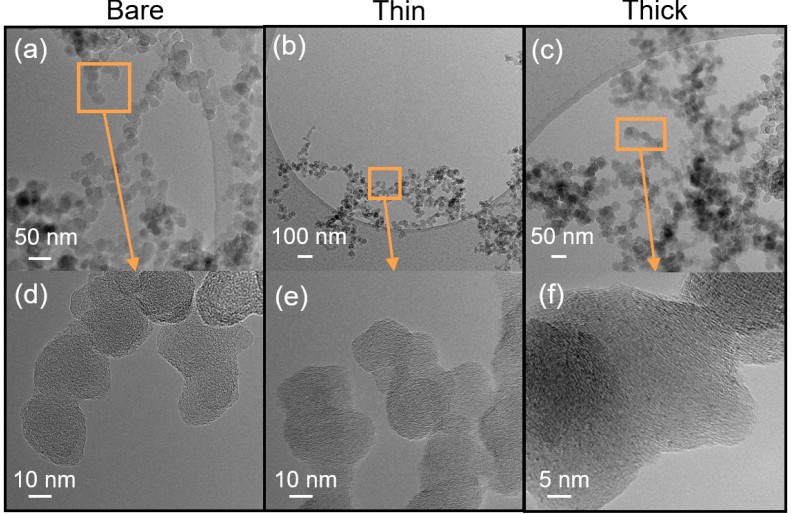

**Figure C4. HR-TEM images for fine structures of 200 nm size selected bare and coated mCASTblack soot particles. Scale bars are indicated in each image. (a) and (d) Bare particles, (b) and (e) Thinly coated particles with a coating $wt$ = 2.9 %, (c) an (f) Thickly coated particles with a coating $wt$ = 30.2 %. All images taken with TFS F30.**

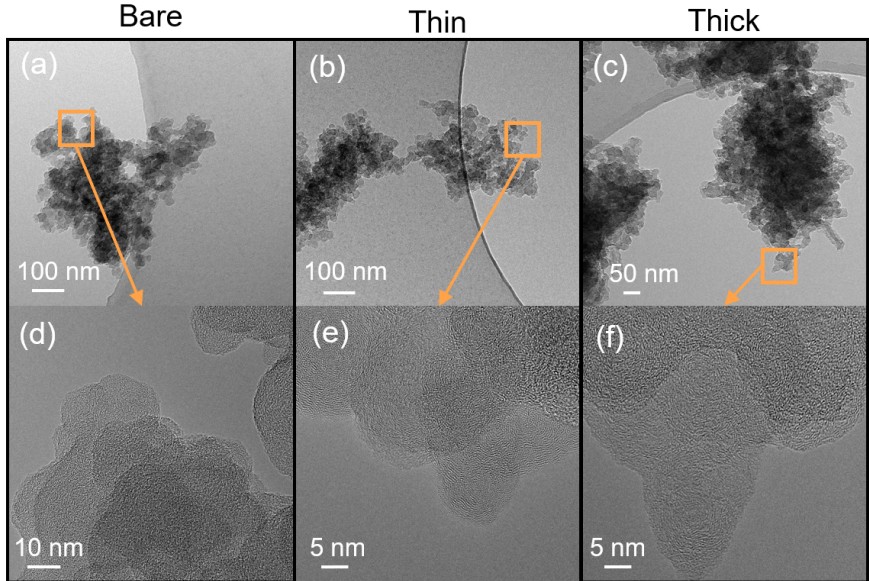





**Figure C5. HR-TEM images for fine structures of 200 nm size selected bare and coated FW200 soot particles. Scale bars are indicated in each image. (a) and (d) Bare particles, (b) and (e) Thinly coated particles with a coating *wt* = 2.3 %, (c) an (f) Thickly coated particles with a coating *wt* = 139.3 %. All images taken with TFS F30.**


As shown in Figs. 10 and 11, C, O and S normalized mass percentages in aera of interests (AOI) on the soot-aggregate surface are presented. These results are calculated from the EDX spectra shown below in Figs. C6 to C11. The three peaks, at 0.93 (Copper, Kα), 1.74 (Silicon, Kα) and 8.10 (Copper, Lα) eV (electron volt), are from the microscopy grid material and are not included in the analysis.

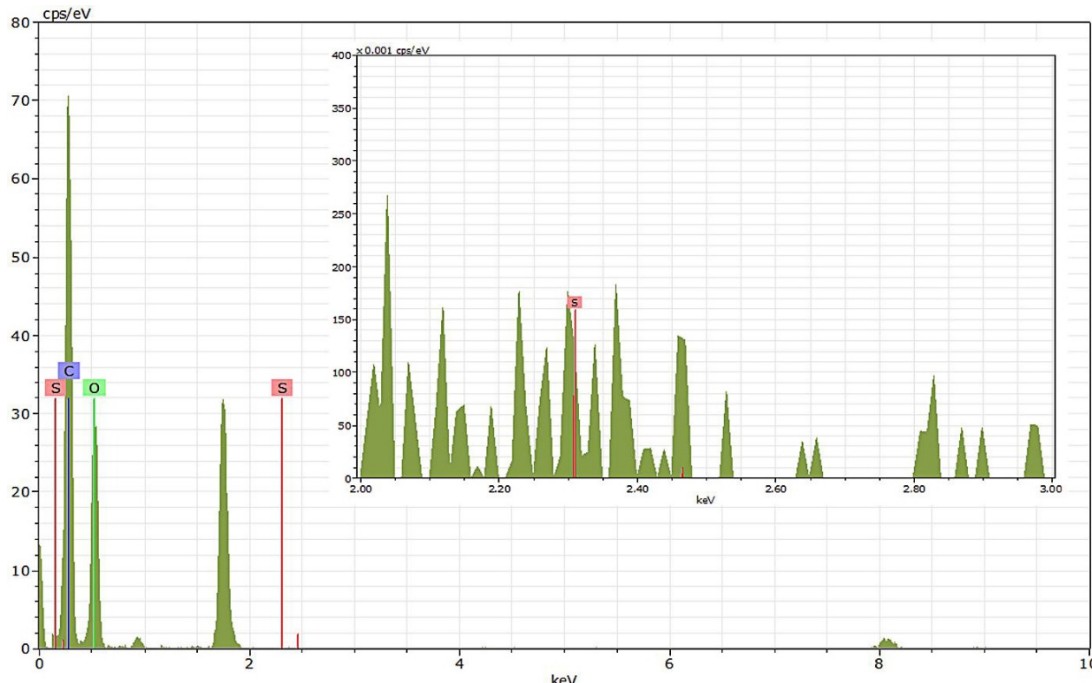


**Figure C6. Energy dispersive X-ray spectroscopy (EDX) spectra for AOI of bare mCASTblack soot presented in Fig. 10. The *x*-axis stands for the electron energy in thousand electron volt (keV) and the *y*-axis represents counts per second per eV (cps/eV). The position of carbon (C), oxygen (O) and sulphur (S) is indicated in the figure. The zoom-in patch shows the sulphur region of the spectra which is not legible in the main spectrum.**




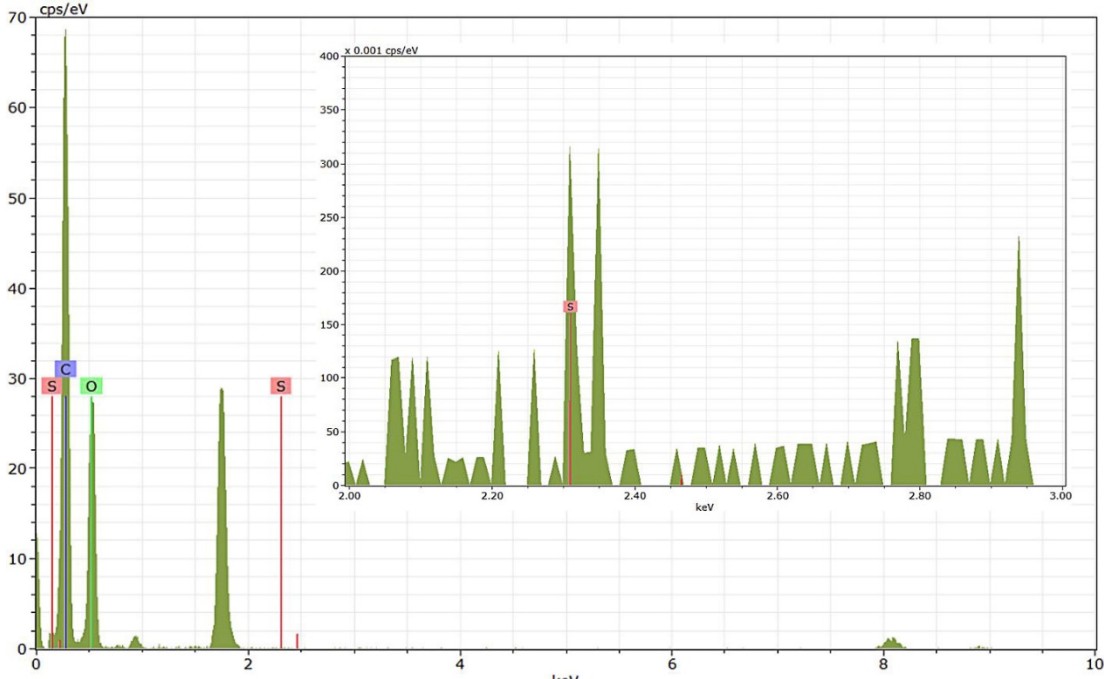

**Figure C7.** Energy dispersive X-ray spectroscopy (EDX) spectra for AOI of thinly coated mCASTblack soot ($H_2SO_4$ coating $wt =$ 2.9 %) presented in Fig. 10. The *x*-axis stands for the electron energy in thousand electron volt (keV) and the *y*-axis represents counts per second per eV (cps/eV). The position of carbon (C), oxygen (O) and sulphur (S) is indicated in the figure. The zoom-in patch shows the sulphur region of the spectra which is not legible in the main spectrum.






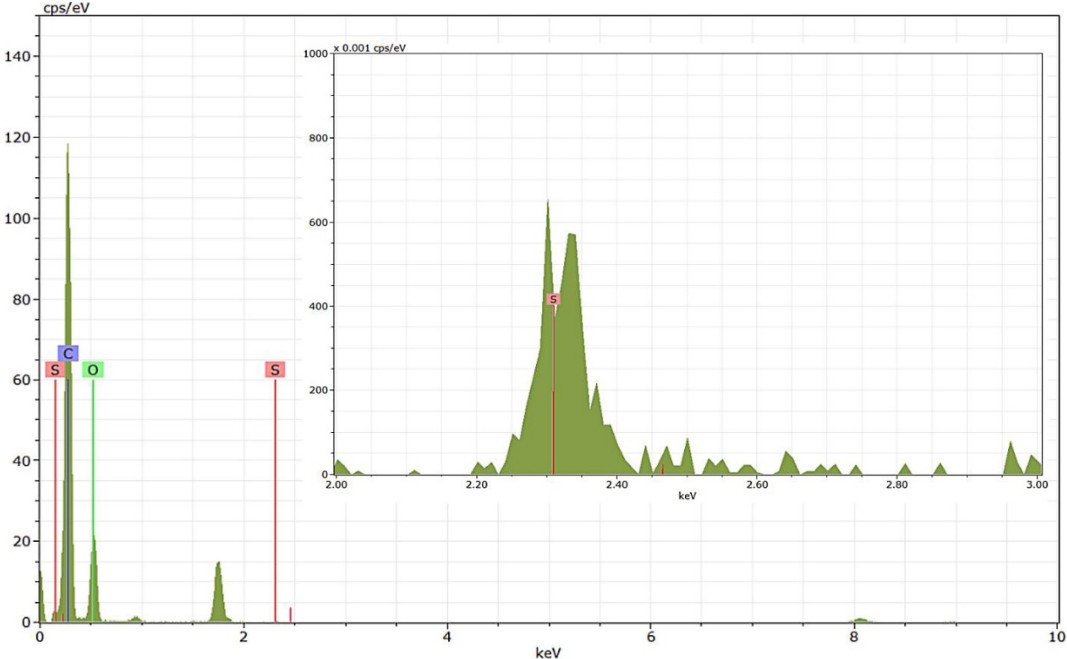

**Figure C8. Energy dispersive X-ray spectroscopy (EDX) spectra for AOI of thickly coated mCASTblack soot (H$_2$SO$_4$ coating *wt* = 30.2 %) presented in Fig. 10. The *x*-axis stands for the electron energy in thousand electron volt (keV) and the *y*-axis represents counts per second per eV (cps/eV). The position of carbon (C), oxygen (O) and sulphur (S) is indicated in the figure. The zoom-in patch shows the sulphur region of the spectra which is not legible in the main spectrum.**




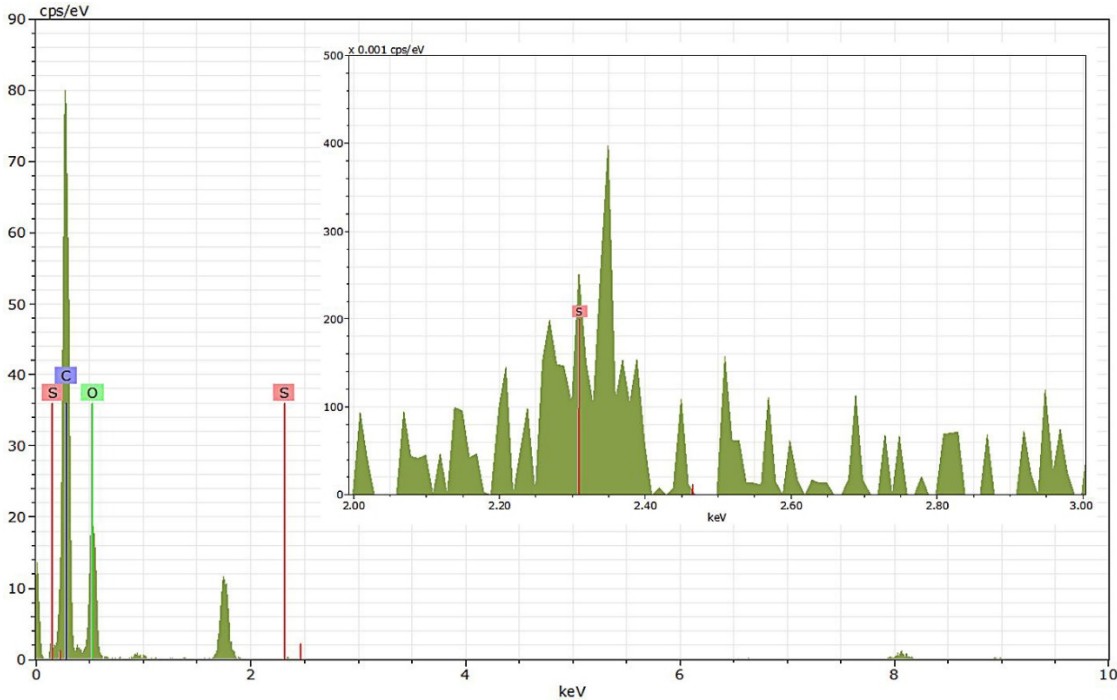

**Figure C9. Energy dispersive X-ray spectroscopy (EDX) spectra for AOI of bare FW200 soot presented in Fig. 11. The *x*-axis stands for the electron energy in thousand electron volt (keV) and the *y*-axis represents counts per second per eV (cps/eV). The position of carbon (C), oxygen (O) and sulphur (S) is indicated in the figure. The zoom-in patch shows the sulphur region of the spectra which is not legible in the main spectrum.**






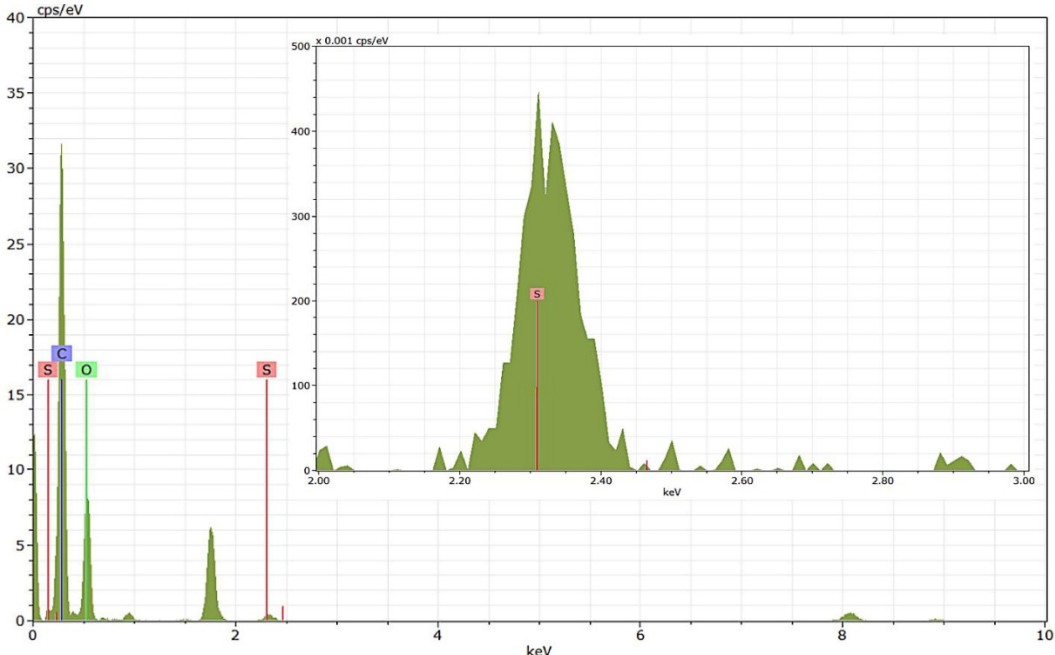

**Figure C10. Energy dispersive X-ray spectroscopy (EDX) spectra for AOI of thinly coated FW200 soot ($H_2SO_4$ coating *wt* = 2.3 %) presented in Fig. 11. The *x*-axis stands for the electron energy in thousand electron volt (keV) and the *y*-axis represents counts per second per eV (cps/eV). The position of carbon (C), oxygen (O) and sulphur (S) is indicated in the figure. The zoom-in patch shows the sulphur region of the spectra which is not legible in the main spectrum.**



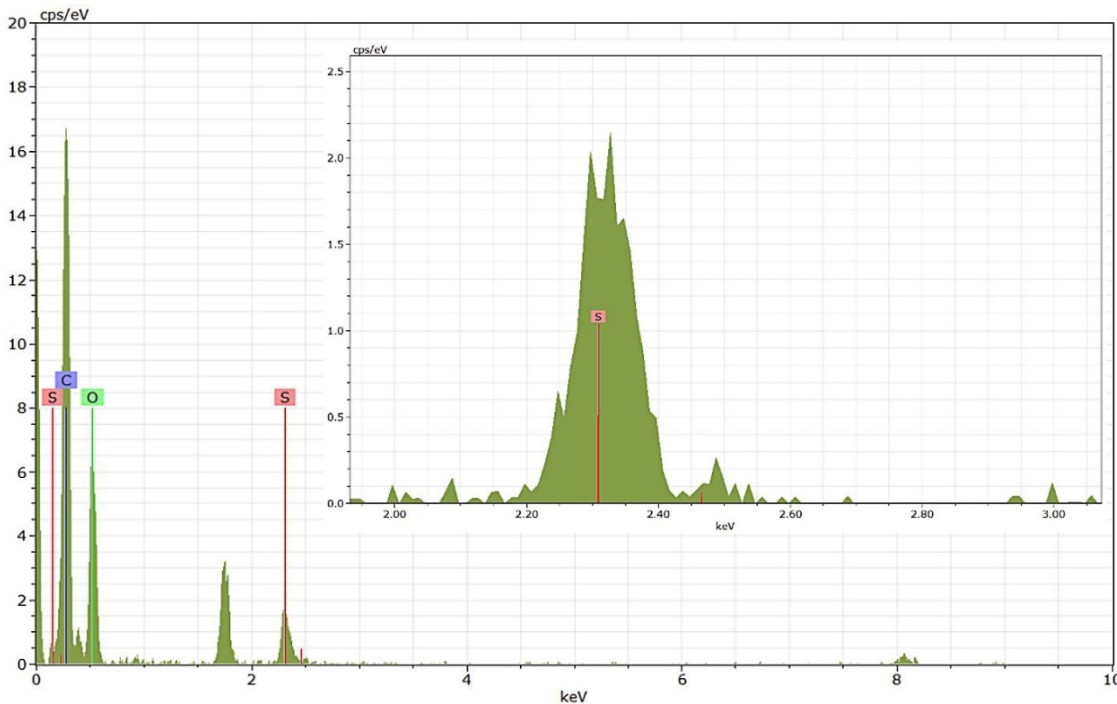


**Figure C11. Energy dispersive X-ray spectroscopy (EDX) spectra for AOI of thickly coated FW200 soot ($H_2SO_4$ coating $wt$ = 139.3 %) presented in Fig. 11. The *x*-axis stands for the electron energy in thousand electron volt (keV) and the *y*-axis represents counts per second per eV (cps/eV). The position of carbon (C), oxygen (O) and sulphur (S) is indicated in the figure. The zoom-in patch shows the sulphur region of the spectra which is not legible in the main spectrum.**


*Data availability:* The data presented in this publication will be made available at DOI 10.3929/ethz-b-000498786. Note by the authors: Data and DOI link will be activated for public access upon acceptance of publication.

*Author contributions:* KG and ZAK designed the experiment and interpreted the data. KG wrote the manuscript and prepared
the figures with contributions from ZAK. EJBM collected SEM, TEM and EDX results. All authors discussed and reviewed the manuscript. ZAK supervised the project.

*Competing interests:* The authors declare that they have no conflict of interest.

*Acknowledgements:* We are grateful to Franz Friebel who originally designed the coating apparatus and to the experimental atmospheric physics group at ETHZ for their help with instrument trouble shooting. Franz Friebel is thanked for his valuable comments on the initial manuscript. The authors also would like to thank Fabian Mahrt for sharing his HINC data processing code. This work was supported by China Scholarship Council (Grant No. 201906020041) and the atmospheric physics professorship at ETH.



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
