# Peer review of "Laboratory studies of ice nucleation onto bare and internally mixed soot-sulphuric acid particles"

_Atmospheric Chemistry and Physics, 2021_

## Referee Comment (RC1)

This study presents and analyzes carefully designed laboratory measurements describing the ice nucleation behavior of soot particles internally mixed to various degrees with sulfuric acid. Data are discussed at cirrus temperatures ranging from 218 K to 243 K, for relative humidities from ice saturation to above water saturation and for two selected soot particle sizes (200 nm and 400 nm). Importantly, soot particle coatings are varied from thin to thick, corresponding to about 3% to 30% weight fractions of soluble material, respectively. The study finds that $H_2SO_4/H_2O$ coatings suppress the ice nucleation ability of soot particles at cirrus levels to a degree that depends on the coating thickness.

The in-depth discussion of the coating process confirms how crucial it is to take disparate soot particle morphologies (i.e. sources and atmospheric ageing processes) into account when assessing their cirrus-forming potential. The characterization of particle morphologies with the help of high-resolution imagery in Appendix C is particularly valuable supporting data interpretation. The study fills an important gap in the literature by presenting a systematic investigation of how soot particle coatings affect heterogeneous ice nucleation and offering a physical explanation (based on pore condensation and freezing (PCF) and immersion freezing processes) for the observed behavior. Existing data sets are valuable in their own right, but have not led to a consistent picture regarding the role of soot particles in cirrus cloud formation.

In my opinion, this is an excellent contribution to the literature on this topic, and I recommend publication after the authors considered my following comments. Line numbers refer to the initially submitted manuscript (made available for technical review).

Major

l81-83: To me, it is unclear if contrail processing had actually happened in the measurements of Petzold et al. (1998) or whether the peculiar signature of the probed soot particles (bimodality) was a feature of the fresh emissions. Please check.

l354: Please define what a mesopore is. This is especially relevant as pore properties feature prominently in the data discussion (mostly in section 3.1). In l402, the term 'micropore' is used and it is unclear what sets it apart from a mesopore.

l668ff: The authors may want to more clearly state in this paragraph that current data suggest that only soot particles with sizes 200 nm and 400 nm actually contribute to cirrus formation, hence are 'atmospherically relevant'. In the case of aviation soot (l683f), most particles emitted at altitude belong to the Aitken mode, i.e., they are much smaller.

Even poor INPs can be efficient in altering cirrus if present in sufficiently high concentrations. BC particles are often present in high number concentrations close to their sources. It may be good to add that besides size-dependent ice nucleation ability the soot particle number concentration must be known in order to judge their potential to affect cirrus formation notably.

I was also wondering whether the authors like to add the importance of further measurements probing smaller soot particles (if feasible, down to 50 nm) to confirm the trend of ice nucleation ability diminishing with decreasing size. This would be especially important for the case of aviation soot, but possibly for other high-temperature combustion sources, too.

Minor

l11: I believe you mean 'net warming'. Same in l39.

l38: The plural of aircraft is still aircraft.

l39: As written, it is unclear whether the climate warming effect relates to cirrus or to aviation soot.

l48: Please clarify that for T<235 K, homogeneous ice formation occurs in liquid aerosol droplets only, setting cirrus apart from mixed-phase clouds.

l63: Sentence ending with 'where aeroengines exhaust sulphur emissions' appears to be incomplete.

l73: Sentence ends with 'but remaining unconstrained' sounds awkward.

l118: What is a 'potential' pore?

l292+293: Sentence sounds awkward.

l331: Define AF.

l418: Please clarify that water uptake is reduced to the inverse Kelvin effect, which, what I believe, is what the authors mean to say.

l686ff: A recent study* concluded that even a small number of ice-active aviation soot particles is capable of modifying the total number and mean size of cirrus ice crystals when nucleating ice alongside homogeneous freezing, albeit with a minor impact on cirrus optical depth. It was found that, based on soot-PCF, only uncoated (barely coated) soot particles with sizes >100 nm contributed to enhanced ice nucleation activity (after contrail-processing). This information may be used in this paragraph relating to mCAST black soot to better explain what is meant by 'may not inhibit or compete with (aerosol) droplet homogeneous freezing' (l689f).

In sum, while not making 'a significant contribution to cirrus cloud formation via PCF' (l693), soot particles may still perturb cirrus microphysical properties. This cautionary note also applies to the subsequent paragraph (l697ff) discussing FW200 soot samples.

*https://www.nature.com/articles/s43247-021-00175-x

l706: consider replacing 'nucleated' with 'water-activated' to avoid confusion with ice nucleation

l727: 'require RH >$RH_{hom}$ conditions' — for what?

l738: 'systemically' -> systematically

l739: 'soot role' -> role of soot

Bernd Kärcher, DLR-IPA

---

## Referee Comment (RC2)

**Review for a manuscript titled: Ice nucleation activities of soot particles internally mixed with sulphuric acid at cirrus cloud conditions, by Gao et al., 2021, ACPD**

Gao et al. present laboratory experiments where they tested the ice nucleation activity of soot particles. Propane combustion particles were generated in a controlled laboratory setup using a burner. In addition a commercial soot sample was used. The ice nucleation activity was tested with a HINC chamber, operated at fixed temperature steps between -55 C and -30 C while performing *RH*-scans at each temperature step. These conditions are relevant for ice formation in cirrus and mixed-phase clouds. The ice nucleation activity is tested for bare soot particles and compared to the ice nucleation activity of soot with varying degrees of H2SO4 coating. The ice nucleation experiments are supported by a suite of auxiliary measurements to characterize the chemical and physical properties of the soot particles and the impact of coating thickness on these properties. The authors explain their observation using the pore condensation and freezing mechanism, its inhibition and enhancement with respect to coating. The authors then present a hypothesis of a 3 step coating process.
Overall, the authors find the tested combustion soot particles to be worse ice nucleation particles in comparison to the commercial soot and demonstrate how coating thickness can affect some of these properties.
The presented study results fall within the scope of Atmospheric Chemistry and Physics (ACP) journal. The observations in this study are largely in-line with previous work and further help to understand the role of soot in cloud formation processes. The manuscript is clearly written however, the conclusions drawn are not sufficiently supported by evidence and some ambiguity remains. These issues need further clarifications. Below, I list my comments and suggestions that should be addressed upon revising the manuscript.

**Major comments:**
The study is well structured and has the suitable experimental design to conduct ice nucleation measurements. However, the study lacks novelty, with some portions of the study being a repetition of previous experiments with the same chamber or in other institutes e.g. L349,L429, L522.

The results, as is, have high degree of uncertainty. Discrepancies between EDX, optical sizes, mobility sizes and electron microscopy sizing on grids and equivalent ML are mentioned but waived aside without a deeper discussion on the uncertainties of the study. Lack of statistical analysis presentation, in particular in electron microscopy, create high ambiguity for the derived conclusions. Some of

the uncertainties, like chemical reactions, are mentioned e.g. L581 but not sufficiently discussed while some are not mentioned at all. The authors mention the presence of the doubly charged particles however, it's not clear which sizes are playing the main role in the nucleation process. Since size is a key factor both for coagulation, condensation and ice nucleation, this aspect of the study should be clarified. A deeper discussion on the physicochemical properties of H2SO4 to the same extent as water freezing is absent.

**Minor, specific and technical comments:**

L1: 'Activities' replace with activity. Also I'd recommend to amend the title to include the bare soot measurements and to clarify that these are laboratory experiments. Also see comment L625.

L10: There is a problem with this sentence, if they are only candidates, what makes them so important? Perhaps use potential instead of important.

L10: comma after formation

L14: I think a sentence of justification is missing here for the choice of propane flame soot or a commercial BC. e.g. "Generic BC surrogates are often used in laboratory experiments. In this study…."

L16: activities - in line 13 you say ability, but study activity? Please keep consistency or explain the differences throughout the text between ability and activity.

L25: abilities – same comment

L35: please add https://doi.org/10.1029/2021JD034649 to Liu et al., 2020.

L36: The numbers cited here are from a review by Bond et al. published 8 years ago and data collected more than a decade ago. Please see if there are additional newer reports. Otherwise, short-lived climate forcers suggest a continuous decline in carbonaceous emissions. Moreover, there is a lack of accurate global quantification of this short lived element (IPCC AR6 report, Chapter 6). These caveats should be mentioned.

L38: aircraft plural is still aircraft

L65: "…vigorous convection" – reference missing.

L69: abilities – see comment for L16 and throughout the manuscript

L72: "potentially… potential", I suggest to reword. For example: Thus, it is possible that the mixing state of soot particle with H2SO4 coating may regulate its INP activity.

L75: remove abilities, remove in-situ

L75: some references could be useful here e.g. Brown et al. DOI: 10.2514/6.2018-3188

L76: please include some references for lab studies of aviation soot.

L77: "contrail dissipation" – please add reference

L77: stick with one form 'in-situ' like in line 75 or 'in situ'.

L88: in line 85 you talk about aviation soot particle surrogates then here you mention graphite, It is not clear how graphite is a surrogate of aviation soot and how this is relevant here.

L90: Mahrt et al. 2018 – please replace with Marcolli C. 2014 and others. I believe it was suggested also much earlier by others.

Marcolli, C.: Deposition nucleation viewed as homogeneous or immersion freezing in pores and cavities, Atmos. Chem. Phys., 14, 2071-2104, 2014.

L116: I'd remove the text after wettability and refrain from mentioning contact angle in the manuscript for two main reasons: First and foremost, you don't report or use contact angle measurements in this study. Secondly, the concept of contact angle measurements and inference to nano-scale dynamic processes is questionable to the least. No surface energy component calculations are truly quantitative nor are they necessarily based upon universally accepted theoretical considerations (see Marmur et al. 2017;Marmur 2006; Strobel&Lyons 2011). Most theories of solid surface energy have a basis in Young's equation, which employs the equilibrium contact angle. In surface energy calculations, many in the surface science field tend to use the so-called static contact angle, which we now know to be meaningless (Marmur et al. 2017), or else the advancing angle, assuming that one of these angles is the equilibrium angle or at least very close to the equilibrium angle. However, the equilibrium contact angle cannot be determined on practical BC surfaces.

Contact angle is indeed a parameter that can reflect surface functionality however it was and continues to be a highly biased method for getting absolute characterization of the surface. If experimentally measured, it is often highly dependent on the methodology of the measurement. Hoose and Mohler (2012) suggested to use a probability distribution function for contact angle. Marcolli (2016) when using contact angle ignored the correction for the size dependence of surface tension and assumed a zero contact angle, she also mentioned that "accurate values for the contact angles between water and the pore walls of our investigated particles are not available".

Marmur, Abraham, Claudio Della Volpe, Stefano Siboni, Alidad Amirfazli, and Jaroslaw W. Drelich. "Contact angles and wettability: towards common and accurate terminology." Surface Innovations 5, no. 1 (2017): 3-8.

Marmur, A.: Soft contact: measurement and interpretation of contact angles, Soft Matter, 2(1), 12–17, doi:10.1039/B514811C, 2006.

Strobel, M. and Lyons, C. S. (2011), An Essay on Contact Angle Measurements. Plasma Processes Polym., 8: 8-13. doi:10.1002/ppap.201000041

Marcolli, C. (2016), 'Pre-activation of aeosol particles by pore condensation and freezing', Atmos. Chem. Phys., 2016, 1-48.

Hoose, C, and O Möhler. "Heterogeneous Ice Nucleation on Atmospheric Aerosols: A Review of Results from Laboratory Experiments." Atmospheric Chemistry and Physics 12, no. 20 (2012): 9817-54.

L127: replace 'need' with 'could benefit from'

L134: in the abstract you also mention mixed-phase, but the conclusions are only about ice nucleation in cirrus clouds?

L137: The samples are not 'experimental' perhaps 'test samples'?

L142: Switch places between the name and the HINC acronym like you did in line146 for DMA

L143: part of the exhaust air

L148: mobility diameter – change throughout the manuscript.

L164: remove 'Experimental'

L166: mention it's a diameter in this sentence.

L167: The aerosol sample – does it contain only the soot aerosol? Are there byproducts in the propane combustion process e.g. gases/vapors? If there are any, where do they go, do they follow the aerosol coating and IN path? Would this direct injection affect your measurements?

L168: home change to in-house

L174: CPMA - see comment L142

L175: sucked changed to pulled

L176: microscopic grids change to microscopy grids

L177: Rephrase. Not clear distinction from what? Other studies in this field? Your previous studies?

Is the "generation of coated soot...and the real time analysis of their properties..." are the only novelty components of this study? why is it important to mention this distinction? Also see my major comment on the significance/novelty of this study.

L178: not sure what online means here? Real-time?

L179: see comment in L16.

L186: finally change to consequently

L191: LabVIEW

L207: indicating no pure nucleated H2SO4 - add reference to subsection where these results are presented.

L211: becomes absent - what does that mean? There is no longer homogeneous nucleation or the mode moves below your detection threshold or rate of coagulation is higher than homogeneous nucleation?

L211: decreases dramatically - please elaborate, is it the difference between nucleation and coagulation rates? What portion coagulated and how much condensed? have you tracked the temperature in this coating process? how mixing the flows with different temperature affects the evaporation rate of H2SO4? could it affect the repeatability?

L215: we are confident change to we conclude

L220: a known changed to defined?

L252: 'and' change to 'so that'

L258: what's the reasoning for using a water CPC?  would you expect differences in comparison to butanol CPC for coated and bare soot counting?

L260: see comment L148

L262: remove mathematical

L262: see comment L148

L266: remove 'by the'

L274: see comment L148

L289: What is the reasoning for bringing this issue up here if there is no solution provided? Stating that it provides still relatively comparable information doesn't resolve the issue. This should be moved into discussions of uncertainties in the study. What type of BET was used for the specific surface area of your samples?

L292: Whereas 400 nm size selected soot particle does not show apparent size growth – why?

L302: home – change to in-house

L331: see comment in L16.

L338: what aerosol is used to calibrate the 780nm diode in the OPC? In appendix C, have you evaluated the response of the OPC to different soot types and diameters directly without passing through the chamber? Is it possible that the OPC mis-sizes non spherical, light absorbing soot with higher refractive index with a complex component?

The coated soot is often studied for its enhanced light absorption and other optical property changes, would that affect the OPC sizing? Bhandari et al. 2019 wrote that compaction affects the soot optical properties. Light absorption and scattering change when a soot particle undergoes morphological transformations.

The recommended operating temperature of the OPC by the manufacturer is 0º C to +50º C, what are the expected biases in detection and sizing for air/aerosol flows at low temperature, down to -55? Would you expect humidity condensation on OPC windows?

L360: see comment L116. You can cite here the earlier studies. It can be attributed to wettability but not necessarily contact angle.

L360: There is no mention of the solvents/organic content pre-existing in FW200 as indicated by the manufacturer. Volatile content gas chromotography or DTA-TGA analysis could shed some light on the chemistry or its wt%. Without it, some discussion is needed about the possible properties of this content and whether glass transition of those organic compounds could impact the IN activity in these temperatures.

L409: Is there a possibility of homogeneous nucleation of the acid and coagulation with soot, Coagulation is strongly affected by residence time. would that be possible on these time scales?

L413: T – spell out temperature throughout the manuscript

L413: inhibits – change to 'could be the main cause for inhibition of the'

L414: see comment L413.

L414: From – change to 'In'

L415: shows coherence- change to 'is coherent'

L429: see comment L413

L430: see comment L413

L440: more uniformly - doesn't that contradict what you wrote in L405?

L455: see comment L116. I think this proves further that the contact angle measurement is not the best choice of technique to include in such studies.

L460: see comment L413.

L461-462: not clear who you are citing here Koehler 2009 or Henson 2007?

L463: doesn't that contradict L440?

L478: soot aggregate - how do you define an aggregate here? is it a single aggregate of spherules, please elaborate. Is it consistent with definitions by Long et al. 2013?

Long, C. M., Nascarella, M. A., and Valberg, P. A.: Carbon black vs. black carbon and other airborne materials containing elemental carbon: Physical and chemical distinctions, Environ. Pollut., 181, 271–286, https://doi.org/10.1016/j.envpol.2013.06.009, 2013

L480: see comment in L148. Small particles with a small mobility size contain less pore volume - Is this generalization of mobility diameter accurate? doesnt that depend also on spherule size? Did you mean 'Small FW200 soot particles'?

L489: In this interpretation of observation, I couldn't find a discussion about the effect of cooling rate, and hydration of the sulfuric acid on its glass transition and changes around the glass transition range, see Zobrist et al 2008; Williams & Long 1995 and others. Also see my last Major comment.

Zobrist, B., Marcolli, C., Pedernera, D. A., and Koop, T.: Do atmospheric aerosols form glasses?, Atmos. Chem. Phys., 8, 5221–5244, https://doi.org/10.5194/acp-8-5221-2008, 2008.

Williams, Leah R., and Forrest S. Long. "Viscosity of supercooled sulfuric acid solutions." The Journal of Physical Chemistry 99.11 (1995): 3748-3751.

L506: sentence too long.

L506: remove 'measurement results'

L506: remove 'characterization results'

L516-519: is that the "collapse" you described in L293? it sounds less obvious here.

L522: fractal dimension - have you explained what this is or how 1.86 was calculated?

L524: coating does not result in a significant soot surface topography change – it doesn't look like there is any morphological difference between 8d,e,f except the addition of a large amount of acid. No evidence to indicate a collapse mentioned in L293 and shown in Fig. 3c.

L529: resulted from the impaction of soot-aggregates on the Cu grid - how do you know that? Further in the text, you've discussed possible other biases of sample transfer, of measurement with SEM in vacuum. Also, how would electron beam interaction with nonconductive particles affect the sample and imaging?

L531: could you also describe what this figure shows in C2a,C2b, C2c?

L542: Small aggregates can also coagulate while transporting in the aerosol flow - coagulation is highly dependent on residence time, if you suggest coagulation is

possible, what about coagulation between soot and small droplets of sulfuric acid below your detection range or those that are neutral in charge?

Also do you see this same extent of coagulation with thinly or thickly coated?

L547: why did you decide to show this aggregate? is that a typical image?

L551: some extent aggregate compaction - previously I believe you called it collapse? Consider toning down the previous description.

L554: the shrinkage, collapse, and compaction terminology should be consistent throughout the text. Could you explain the differences between the percentage of decrease in diameter in Fig 3a, and the shrinkage observed in Fig. 8c?

L559: a supplementary video recording starting with the initial state could provide valuable qualitative information about the structure for the readers.

L569:  demonstrated that

L569: shows – change to has

L570: 'to indicate' – change to 'to characterize the'

L571: S – spell out sulfur and all the other elements throughout the text

L576: see comment L571

L580: S,O see comment L571

L581: This may suggest that there is a reaction between mCASTblack soot and $H_2SO_4$ depleting some O content in the process - If you are suggestion that there is a chemical reaction taking place, wouldn't that have an impact on IN activity? On partial vapour pressure? This is in addition to the changes in $H_2SO_4$ viscosity (see comment L489). These processes, which are barely discussed may affect your hypothesis. See Major comments.

L584: see comment L571

L586: Does that contradict your interpretation of observations in L529? Is it possible that evaporation of coating would induce morphological changes like compaction, similar to droplet evaporation in cloud processes? See also comment to Fig.12.

L590: be $H_2SO_4$ droplets – Fig. 11a? not clear where it's shown.

L596: would be good to attach a datasheet with all the known properties of FW200 provided by the manufacturer as a supplement. The product may be discontinued in the future and the information from the manufacturer will be lost.

L596: If all the reservations in this paragraph are true, it is not clear how the statement in L614 (robust evidence) is supported. Or how EDX analysis can be used at all to explain any coating observations in this study.

L610: Firstly, this is not unequivocally evident from your analysis. Secondly, there could be other possible interpretations that you haven't discussed, see comment L489. I suggest to tone down these statements.

L625: This section should be revised or even removed. The title/focus of the manuscript is ice nucleation. Is the hypothesis brought up here is highly relevant to the main topic? Perhaps the title should be changed.

L645: The fatty acid example is irrelevant to the hypothesis of sulfuric acid coating process. What these two have in common?

L652: Im not sure I understand the physics of this process, could you elaborate further? See comment to Fig.12.

L654: please avoid drama in science, "supported by the compaction of thickly coated..."

L658: remove a dramatic

L663: robust support from measurements in this study – I'd highly recommend to tone down the statement and properly address the major comments.

L674: the sentence here is fairly general and doesn't require citation, unless you want to refer the readers e.g. 'see examples in Kanji et al 2017'?

L678: morphological properties and size?

L688-L697: wouldn't it also depend on the dynamic expansion pulse into the low pressure environment at high altitude and induced supersaturation?

L699: similar to our – change to similar to the FW200 soot sample used here

L705: it is likely that ice crystals can be induced by them – reword.

L712: loses change to loose

L719: carbon poor?

L774: because a 200 nm pure H2SO4 particle has a mass of 7.7 fg - what about the bottom edge of the PSD for pure H2SO4?

L805: to be reliably detected in the 5 μm channel - Korhonen et al 2021 selected 6 micron as their bottom threshold. Would lamina flow peripheral regions have slightly different RH conditions?

Korhonen, K., Kristensen, T. B., Falk, J., Malmborg, V. B., Eriksson, A., Gren, L., Novakovic, M., Shamun, S., Karjalainen, P., Markkula, L., Pagels, J., Svenningsson, B., Tunér, M., Komppula, M., Laaksonen, A., and Virtanen, A.: Particle emissions from a modern heavy-duty diesel engine as ice-nuclei in immersion freezing mode: an experimental study on fossil and renewable fuels, Atmos. Chem. Phys. Discuss. [preprint], https://doi.org/10.5194/acp-2021-111, in review, 2021.

L867: believed - In my opinion EDX is not a suitable technique to analyze organics, it is more suited for the analysis of inorganic materials, rather than materials that contain for the most part all the same organic elements bonded differently. Especially in the case of a commercial FW200.

L868: impaction/drying - you also mentioned earlier the electron beam evaporated the acid.

L872: particle concentration is high - why this was not addressed during the experiments? This is a known issue, which is controlled by grid size selection, collection time, and flow settings.

L874: a high probability - I believe the collection efficiency is something that can be calculated. See previous comment. Was the ZEMI instrument calibrated for proper collection in previous studies?

L877: 3000 cm-3 - what is the highest number of particles the instrument can size without coincidence bias that will cause mis-sizing?

L881: 'suggesting' - see comments L338, L877.

A better way would be to analyze the volatile components while heating the collected particles, or analyzing the exhaust of a thermal denuder.

L908: primary particle edges are more ambiguous in Fig. C4f for thickly coated soot particles - this is not obvious, a higher magnification is shown in C4f. what differences one would expect to see for organic volatile content in ultra-high vacuum for HRTEM and high energy beams?

Figures like this of lower significance and their description can be moved to a supplement.

L921: see comment in L867. I think figures C4-C11 can be moved to a supplement. A more informative figure/table would be of a statistical analysis for these measurements, which is the standard practice in electron microscopy for extrapolation of properties to a broader sample area.

L921: the areas of interest

L976: is this a thesis? please indicate.

L1068 and others: The DOI links dont work. All those should be changed to the right format e.g. https://doi.org/10.5194/acp-17-15199-2017.

L1994: 2020b should be listed after 2020a?

Fig. 1: Exhaust filter, flow splitter – correct labels,

   1.5 L/min does that belong to the CPMA on the right?

Fig. 2: remove the turquoise box in the CPC, irrelevant

Table 1: first column coating T – change to H2SO4 evaporation T

Fig. 3: The different soot types should be indicated on the plots too.

caption: size – change to diameter measured by SMPS.

   Effective – change to calculated effective

Table 2: first column, please elaborate what is that temperature. Also, is this table really necessary or the numbers can be mentioned in the main text?

What's the significance of mentioning 30C results, which suggest, if I understood correctly, that there might be no coating?

Fig. 10: I suspect multiple measurements were done to make this figure rather than one single measurement for each nanoparticle type and inference for the whole grid. Please include some statistical analysis of sampling in different locations on same particle and/or on different particles on the grid in each category and add stdev values to elemental percentage that you present. With the currently presented values, the difference significance between all 3 is not obvious.

why did you choose to use a precision of 1 decimal for carbon and oxygen but 2 decimals for sulfur?

Fig. 11: see comment for Fig. 10.

Fig. 12: interestingly, you illustrate here the sulfuric acid as homogeneously nucleated droplets rather than vapour condensing in monolayers? Or perhaps I got it wrong?

I'm not sure I understand this conceptual collapse in Fig. 12d. what forces are causing it? I would expect the particle to collapse upon drying due to surface tension exerted on the aggregate rather than during the coating process.

Also, in all references it's sulfuric while you use sulphuric, consider changing throughout the text for consistency.

Fig. B8: After this figure, or in the main text, I'd recommend to add a composite figure that will include all the results of onset saturation ratio with respect to ice freezing where 1 % of the aerosol particles are activated as a function of temperature.

Fig. C3: what are the implications of these charts on the ice nucleation results and analysis? A discussion is missing in the main text.

---

## Author Comment (AC1)

Response to acp-2021-645 reviews for RC1

Dear Bernd Kärcher,

Thank you very much for reviewing our manuscript acp-2021-645. We are grateful for your comments and constructive suggestions. Please find our itemized response in below and corrections in the revised version. Your comments are reproduced **in bold** and our responses are given directly afterward in normal font. *The original text in previous manuscript version is reproduced in red italic* and *revised text is added in blue italic*.

All the best,

Kunfeng and Zamin

- **Major:**

1  **L81-83: To me, it is unclear if contrail processing had actually happened in the measurements of Petzold et al. (1998) or whether the peculiar signature of the probed soot particles (bimodality) was a feature of the fresh emissions. Please check.**

R:  Petzold et al. (1998) sampled contrail and cirrus crystals using a counter-flow virtual impactor (CVI) and characterized the chemical composition, morphology and size distribution of ice-residual particles. It is evident that cloud processing occurred in the study, because characterized ice-residual particles (majorly soot particles, ~87 %) are leftover substances from the ice crystal after sublimation or melting followed by evaporation. Petzold et al. (1998) only characterized the size distribution of ice-residual particles by statistically analyzing the size of sampled particles on filters and divided the residual particles into three size bins, including fine particles (0.1-0.5 μm), intermediate (0.5-1.5 μm) and coarse particles (1.5-5.0 μm). However, the authors did not measure the size distribution of interstitial aerosol particles or fresh soot particles. This description therefore suggests contrail processing.

We added two references (see L86-87 in revised manuscript) to support that cloud processing can modify the morphology of freshly emitted soot particles. The first literature is from Colbeck et al. (1990) who observed a fractal dimension value increase for fresh soot particle (~ 1.8) after exposing to cloud processing simulation conditions (> 2.0-2.5). The second literature is from Petzold and Schröder (1998) who suggested the fresh soot particle size distribution is dominated by fine particles (< 0.1 μm), which is different from cloud processed soot particles which have larger sizes. We present the corrected text in L85-87 in the revised manuscript:

'*In addition, soot with a more compacted morphology than the freshly emitted particles are detected in aviation contrail ice crystal residues (Colbeck et al., 1990; Petzold and Schröder, 1998; Petzold et al., 1998) suggesting a change in shape and size of the soot-aggregates due to contrail processing.*'

2  **L354: Please define what a mesopore is. This is especially relevant as pore properties feature prominently in the data discussion (mostly in section 3.1). In l402, the term 'micropore' is used and it is unclear what sets it apart from a mesopore.**

R:  Thanks, we agree this is a valuable definition to include. We updated this in revised manuscript L362-364 as below:

'*...homogeneous freezing conditions in the cirrus cloud regime (Mahrt et al., 2018). Herein, mesopores are defined as pore structures with a width between 2 and 50 nm whereas pore structures with a width less than 2 nm refer to micropores (Thommes et al., 2015).*'

3  **L668ff: The authors may want to more clearly state in this paragraph that current data**

**suggest that only soot particles with sizes 200 nm and 400 nm actually contribute to cirrus formation, hence are 'atmospherically relevant'. In the case of aviation soot (l683f), most particles emitted at altitude belong to the Aitken mode, i.e., they are much smaller.**

R:    Thank you for raising the concern on the size dependence of soot ice nucleation. In the referred paragraph we state that "*Field studies detected soot-aggregates with 0.1-1 µm sizes in ice crystal residuals sampled from cirrus clouds (Petzold et al., 1998; Twohy and Poellot, 2005; Cziczo and Froyd, 2014) and the presence of 100-800 nm BC particles in the interstitial aerosol samples increases with increasing altitude from 8 to 11 km in the cirrus cloud regime (Petzold et al., 1998).*", which implies soot particles in the size range of 200 and 400 nm are relevant to cirrus formation. We agree, that fine particles (< 100 nm) dominate the number concentration of aviation soot particle population, but soot particles larger than ~ 100 nm in the aviation emission take a large proportion of volume/mass concentration (Brito et al., 2014; Fushimi et al., 2019). In addition, Zhang et al. (2020) reported that fine particles coagulate into larger soot aggregates downstream of the emission, upon evolution of the aviation plume. Therefore, our study of 200 and 400 nm aviation soot proxies are relevant for the evaluation of aviation soot contribution on cirrus cloud formation, considering the size dependence of soot ice nucleation, i.e. larger soot particles are more active INPs. We updated the manuscript text at the end of the paragraph (L697-700 in revised manuscript) as following:

'... *Soot particles emitted from biomass burning and aviation activities are dominated by fine particles (< 100 nm) in number concentration, however, large particles (> 100 nm) dominate in volume concentration (Brito et al., 2014; Fushimi et al., 2019). In addition, Zhang et al. (2020) reported that fine particles coagulate into large soot aggregates downstream of the emission upon aviation plume evolution. Thus, we believe that the results presented in this study are of atmospherically relevant sizes (200 and 400 nm).*'

4    **Even poor INPs can be efficient in altering cirrus if present in sufficiently high concentrations. BC particles are often present in high number concentrations close to their sources. It may be good to add that besides size-dependent ice nucleation ability the soot particle number concentration must be known in order to judge their potential to affect cirrus formation notably.**

R:    Thank you for this comment. We agreed and modified the statement in between the second paragraph in Sect. 4 as below (L707-710 in revised manuscript):

'*Despite the minor ice nucleation ability of mCASTblack soot via PCF even at 218 K as presented in this study, uncoated aviation soot analogous to fractal and hydrophobic mCASTblack may still influence cirrus cloud properties if soot particles are in present in sufficiently high number concentration near their sources because a non-negligible number will nucleate ice crystals (Kärcher et al., 2021).*'

5    **I was also wondering whether the authors like to add the importance of further measurements probing smaller soot particles (if feasible, down to 50 nm) to confirm the trend of ice nucleation ability diminishing with decreasing size. This would be especially important for the case of aviation soot, but possibly for other high-temperature combustion sources, too.**

R:    Thank you for this suggestion. Yes, we agree that the investigation of size threshold for soot ice nucleation is important. In fact, we have just submitted a manuscript to ACP where we present ice nucleation onto soot particles down to 60 nm (Gao et al., 2021). A new statement was added at the end of the Sect. 5 as below (L767-770 in revised manuscript):

'...*Moreover, measuring the lower size limit to identify the threshold of soot ice formation will be*

*important for future laboratory studies, given the strong particle size dependence of (soot) ice nucleation and that the Aitken mode dominates size distribution from aviation soot emissions and also other high temperature combustion sources.*'

- **Minor:**

**1 L11: I believe you mean 'net warming'. Same in L39.**

R: Yes. Corrected (now L11 and L40 respectively in revised manuscript)

**2 L38: The plural of aircraft is still aircraft.**

R: Corrected (L40 in revised manuscript).

**3 L39: As written, it is unclear whether the climate warming effect relates to cirrus or to aviation soot.**

R: Thank you. The sentence was restructured as below (L39-41 in revised manuscript):

'*...Aviation soot particles, directly emitted by commercial aircraft in the upper troposphere, exert net-warming effects on climate (Liou, 1986) by acting as potential ice nucleating particles (INPs) at high altitudes where cirrus clouds usually form.*'

Original sentence: '*Aviation soot particles, directly emitted by commercial aircraft in the upper troposphere, are potential ice nucleating particles (INPs) at high altitudes where cirrus clouds usually form, and exert net-warming effects on climate (Liou, 1986).*'

**4 L48: Please clarify that for T<235 K, homogeneous ice formation occurs in liquid aerosol droplets only, setting cirrus apart from mixed-phase clouds.**

R: Thanks. The sentence (now L52-53 in revised manuscript) was modified as below:

'*...Homogeneous ice nucleation can only be triggered at $T < 235$ K (homogeneous nucleation temperature, HNT) and relative humidity with respect to ice (RH$_i$) higher than 140 % (Koop et al., 2000), where the nucleation rates of liquid aerosol particles are large enough to freeze spontaneously leading to cirrus cloud formation.*'

**5 L63: Sentence ending with 'where aeroengines exhaust sulphur emissions' appears to be incomplete.**

R: Thank you. Agreed and corrected as below (now L66 in revised manuscript):

'*...It is conceivable that soot particles and $H_2SO_4$ can be internally mixed forming $H_2SO_4$ coated soot in the atmosphere, especially in high altitude aircraft corridors where aeroengines emit fossil fuel combustion aerosol.*'

**6 L73: Sentence ends with 'but remaining unconstrained' sounds awkward.**

R: Thanks. The sentence was revised as following (now L76-77 in revised manuscript):

'*...Thus, the internal mixing of soot particles with $H_2SO_4$ may regulate its ability to be a potential INP but the mixing state is unconstrained.*'

**7 L118: What is a 'potential' pore?**

R: In this context, 'a potential pore for PCF' means the pore structures in soot particles which have appropriate pore width and wettability, thus can induce capillary condensation under sub-saturation

conditions. But the activation of such a pore structure also depends on temperature condition, that is why we termed it as 'a potential pore'.

**8    L292+293: Sentence sounds awkward.**

R:    We reorganized as below (now L296-297 in revised manuscript):

'*However, such a small size increase is absent for the case of coating 400 nm soot particles, and the size starts to decrease due to collapse when the coating mass percentage reaches ~ 20 %.*'

Original statement: '*Whereas 400 nm size selected soot particle does not show apparent size growth and starts to collapse when the coating mass percentage reaches ~ 20 %.*'

**9    L331: Define AF.**

R:    The AF was already defined in L238 (in original manuscript, now L242 in revised manuscript).

**10    L418: Please clarify that water uptake is reduced to the inverse Kelvin effect, which, what I believe, is what the authors mean to say.**

R:    Here, we intend to present that excess $H_2SO_4$ coating for pore filling will spread over soot surface and enhance soot surface hygroscopicity. To make a clear statement, we corrected the sentence as below (now L426-429 in revised manuscript):

'*With more coating (in medium and thick cases), hygroscopic $H_2SO_4$ adsorption on soot surfaces may occur simultaneously with $H_2SO_4$ pore filling and thus enhance soot surface hygroscopicity. Hence, soot particles with medium or thick coating will not only sustain pore filling but also form bulk water droplets more readily.*'

Original statement: '*With more coating (in medium and thick cases), hygroscopic $H_2SO_4$ adsorption on soot surfaces competes with $H_2SO_4$ pore filling and promotes the water uptake ability of soot particles to form a bulk water droplet more readily.*'

**11    L686ff: A recent study\* concluded that even a small number of ice-active aviation soot particles is capable of modifying the total number and mean size of cirrus ice crystals when nucleating ice alongside homogeneous freezing, albeit with a minor impact on cirrus optical depth. It was found that, based on soot-PCF, only uncoated (barely coated) soot particles with sizes >100 nm contributed to enhanced ice nucleation activity (after contrail-processing). This information may be used in this paragraph relating to mCASTblack soot to better explain what is meant by 'may not inhibit or compete with (aerosol) droplet homogeneous freezing' (l689f). In sum, while not making 'a significant contribution to cirrus cloud formation via PCF' (l693), soot particles may still perturb cirrus microphysical properties. This cautionary note also applies to the subsequent paragraph (l697ff) discussing FW200 soot samples.**
**\*https://www.nature.com/articles/s43247-021-00175-x**

R:    Thank you for this constructive comment. Following your suggestion, a short paragraph was added at the end of Sect. 3.4 as below (now L736-743 in revised manuscript):

'*In brief, $H_2SO_4$ coating modifies soot particle ice nucleation in the cirrus cloud regime by decreasing the pore availability for PCF activation thereby inhibiting ice formation below $RH_{hom}$ conditions. This finding is consistent with a model simulation study based on soot-PCF framework (Marcolli et al., 2021) performed by Kärcher et al. (2021) who reported that only barely coated soot particles larger than ~ 100 nm with sufficient mesopore structures can contribute to cirrus clouds ice formation. Nonetheless, the*'

*authors also suggested that even a small number of active uncoated soot particles (~ 11 L$^{-1}$) can decrease the total number of cirrus ice crystals but increase the ice crystal mean size by forming ice via PCF processes while competing with the homogeneous freezing of aerosol droplets (Kärcher et al., 2021). Therefore, soot particles may still play an important role in regulating cirrus cloud properties, despite the suppressed ice nucleation ability caused by H$_2$SO$_4$ coating.'*

**12   L706: consider replacing 'nucleated' with 'water-activated' to avoid confusion with ice nucleation**

R:   Thanks. Agreed and corrected (now L723 in revised manuscript).

**13   L727: 'require RH > RHhom conditions' — for what?**

R:   To make a clear statement, we revised the statement as below (now L753-754 in revised manuscript):

'*High wt/wt% H$_2$SO$_4$ coated 200 and 400 nm mCASTblack soot particles form ice crystals via homogeneous freezing and low wt/wt% H$_2$SO$_4$ coating even depresses their ice nucleation to RH > RH$_{hom}$ conditions at T < HNT.*'

Original statement: '*H$_2$SO$_4$ coating makes 200 and 400 nm mCASTblack soot particles form ice crystals via homogeneous freezing and low mass H$_2$SO$_4$ coating even depresses their ice nucleation activity at T < HNT to require RH > RH$_{hom}$ conditions.*'

**14   L738: 'systemically' -> systematically**

R:   Thanks. Agreed and corrected (now L766 in revised manuscript).

**15   L739: 'soot role' -> role of soot**

R:   Agreed and corrected (now L767 in revised manuscript).

**Reference**:

Brito, J., Rizzo, L. V., Morgan, W. T., Coe, H., Johnson, B., Haywood, J., Longo, K., Freitas, S., Andreae, M. O., and Artaxo, P.: Ground-based aerosol characterization during the South American Biomass Burning Analysis (SAMBBA) field experiment, Atmos. Chem. Phys., 14, 12069-12083, https://10.5194/acp-14-12069-2014, 2014.

Colbeck, I., Appleby, L., Hardman, E. J., and Harrison, R. M.: The optical properties and morphology of cloud-processed carbonaceous smoke., J. Aerosol Sci., 21, 527–538, https://doi.org/10.1016/0021-8502(90)90129-L, 1990.

Cziczo, D. J. and Froyd, K. D.: Sampling the composition of cirrus ice residuals, Atmos. Res., 142, 15-31, https://doi.org/10.1016/j.atmosres.2013.06.012, 2014.

Fushimi, A., Saitoh, K., Fujitani, Y., and Takegawa, N.: Identification of jet lubrication oil as a major component of aircraft exhaust nanoparticles, Atmos. Chem. Phys., 19, 6389-6399, https://10.5194/acp-19-6389-2019, 2019.

Gao, K., Friebel, F., Zhou, C.-W., and Kanji, Z. A.: Enhanced soot particle ice nucleation ability induced by aggregate compaction and densification, Atmos. Chem. Phys. Discuss., http://10.5194/acp-2021-883, 2021.

Kärcher, B., Mahrt, F., and Marcolli, C.: Process-oriented analysis of aircraft soot-cirrus interactions constrains the climate impact of aviation, Commun. Earth Environ., 2, https://10.1038/s43247-021-00175-x, 2021.

Koop, T., Luo, B., Tsias, A., and Peter, T.: Water activity as the determinant for homogeneous ice nucleation in aqueous solutions, Nature, 406, 4, https://doi.org/10.1038/35020537, 2000.

Liou, K.-N.: Influence of cirrus clouds on weather and climate processes: A global perspective, Mon. Weather Rev., 114, 1167–1199, 1986.

Marcolli, C., Mahrt, F., and Kärcher, B.: Soot-PCF: Pore condensation and freezing framework for soot aggregates, Atmos. Chem. Phys., https://doi.org/10.5194/acp-21-7791-2021, 2021.

Petzold, A. and Schröder, F. P.: Jet engine exhaust aerosol characterization, Aerosol Sci. Tech., 28, 62-76, https://doi.org/10.1080/02786829808965512, 1998.

Petzold, A., Strom, J., Ohlsson, S., and Schröder, F. P.: Elemental composition and morphology of ice-crystal residual particles in cirrus clouds and contrails, Atmos. Res., 49, 21-34, http://doi.org/10.1016/S0169-8095(97)00083-5, 1998.

Twohy, C. H. and Poellot, M. R.: Chemical characteristics of ice residual nuclei in anvil cirrus clouds: evidence for homogeneous and heterogeneous ice formation, Atmos. Chem. Phys., 5, 2289–2297, 2005.

Zhang, X., Karl, M., Zhang, L., and Wang, J.: Influence of Aviation Emission on the Particle Number Concentration near Zurich Airport, Environ. Sci. Technol., 54, 14161-14171, https://10.1021/acs.est.0c02249, 2020.

---

## Author Comment (AC2)

We thank Reviewer 2 for their effort and feedback on our manuscript ACP-2021-645. In response to the questions and suggestions, please find our answers and corrections listed below. **Reviewer 2 comments are extracted in bold from original review supplement** and our responses are given directly below in normal font. *The original text in previous manuscript is repeated in red italic* and *corrected text in revised manuscript typed is in blue italic*.

- **Major:**

**1     The study is well structured and has the suitable experimental design to conduct ice nucleation measurements. However, the study lacks novelty, with some portions of the study being a repetition of previous experiments with the same chamber or in other institutes e.g. L349, L429, L522.**

R:     We acknowledge that Mahrt et al. (2018) (L349 and L522, now L355 and L526 in revised manuscript respectively) studied the ice nucleation activity of freshly generated organic lean $C_3H_8$ (propane) flame soot (mCASTblack) and commercial black carbon soot (FW200) soot particles using the same chamber HINC. Our study however, focuses on how the mixing state of soot with $H_2SO_4$ regulates soot ice nucleation activity in the mixed-phase and cirrus cloud regime. By using the same ice nucleation chamber to perform RH (relative humidity) scans for size selected mCASTblack and FW200 soot particles with various $H_2SO_4$ coating thicknesses, we ensure repeatability and direct comparability, which we believe is important to the research process. The experiments with uncoated samples set the benchmark for further comparison and thus are necessary to repeat, also part of a robust research process. Therefore, the measurement of bare mCASTblack and FW200 soot particles is a repetition of Mahrt et al. (2018) but it is necessary and reasonable as these are needed to compare to the results of coated particles.

In our study, we coated each size selected soot particle sample with at least eight different $H_2SO_4$ coating thicknesses, ranging from really thin (*wt %* < 3 %) to thick *wt %* (> 100 %) coatings which are atmospherically relevant (Bhandari et al., 2019). Thus, the novelty is in the atmospheric relevance of the coating thicknesses that cover cleaner regions to very polluted regions. Previous studies typically achieved only very thick coating conditions. In addition, we discussed the mechanism of ice nucleation in detail by means of the varying coating thickness. These discussions and results have not been presented before with a systematic control of size and coating thickness parameters. The detailed comparison between our study and previous studies (Möhler et al., 2005; Crawford et al., 2011; Mahrt et al., 2018; Mahrt et al., 2020a) was performed to understand and put into context the ice nucleation activity of our soot samples rather than highlight similarities in the studies. We have done this in the main text in Sect. 3 (L348-355, now L355-362 in revised manuscript) and Sect. 3.1 (L429, L432 and L434, now L435, L438 and L440 in revised manuscript).

**2     The results, as is, have high degree of uncertainty. Discrepancies between EDX, optical sizes, mobility sizes and electron microscopy sizing on grids and equivalent ML are mentioned but waived aside without a deeper discussion on the uncertainties of the study. Lack of statistical analysis presentation, in particular in electron microscopy, create high ambiguity for the derived conclusions. Some of the uncertainties, like chemical reactions, are mentioned e.g. L581 but not sufficiently discussed while some are not mentioned at all. The authors mention the presence of the doubly charged particles however, it's not clear which sizes are playing the main role in the nucleation process. Since size is a key factor both for coagulation, condensation and ice nucleation,**

**this aspect of the study should be clarified. A deeper discussion on the physicochemical properties of H₂SO₄ to the same extent as water freezing is absent.**

R:    We acknowledge the reviewer's comment and respond to each aspect as below. First, we respond to the size measurement discrepancies raised. As addressed in Appendix C of the manuscript, micron size soot agglomerates were observed for uncoated mCASTblack soot from its 200 nm mobility diameter TEM (transmission electron microscopy) images and 400 nm mobility diameter SEM (scanning electron microscopy) images, resulting from soot-aggregates coagulating when depositing the sample particles onto the microscopy grids under high particle number concentration ($\sim$ 3000 # cm$^{-3}$) conditions. When collecting soot particles onto grids, such a large particle concentration provides a high probability for submicron size single soot aggregates to coagulate. However, micron size agglomerates were not observed for coated mCASTblack and bare and uncoated FW200 soot samples whose TEM/SEM grids were collected with a much lower particle concentration ($\sim$ 200-300 # cm$^{-3}$). On the other hand, optical size measurements of 200 and 400 nm soot particles ($\sim$ 3000 # cm$^{-3}$ in aerosol phase) were used to confirm if the DMA (differential mobility analyzer) could select soot particles with correct size and the results demonstrated the absence of micron size particle in the aerosol phase under different particle number concentration conditions (see Fig. C3 in original manuscript). This measurement further confirms that micron size particles are absent for the same soot sample (uncoated) used for aerosol particle ice nucleation experiments with similar or lower particle number concentrations.

As for the uncertainty of coating equivalent monolayer (ML), we used this concept from Wyslouzil et al. (1994) to evaluate the coating state and to investigate the relation between ML values and the corresponding soot ice nucleation activity. In Sect. 2.2.3 (L284-286, now L288-290 in the revised manuscript), we addressed the ML values for coated soot particles depend on the specific surface area and the technique used for measuring the surface area besides the coating masses, which leads to the ML value calculation uncertainty. In Sect.3.3.1 (L614-624, now L623-632 in the revised manuscript), we discussed the relation between soot ice nucleation and corresponding ML values and concluded that the equivalent ML value is not a reliable proxy to describe the distribution of H₂SO₄ coating material on soot-aggregate surfaces (L631 in revised manuscript Sect. 3.3.1). Nonetheless, the ML values of coated soot particle samples provide nominal coating thicknesses to understand the amounts of coating and its effect on soot particle morphology change. Therefore, we used coating *wt* %, particle mobility diameter change, as well as microscopy images as descriptors to evaluate coated soot particle mixing state and to understand the coated soot particle ice nucleation activity, in addition to the ML value (see Sect. 2.2.3 and Sect. 3.3).

In order to address the concern raised by the reviewer on the statistics of the electron microscopy images, we provided more microscopy images of bare and coated soot particles. We took more than 40 images randomly from each soot sample grid and we present 8 images randomly selected from these images, as shown in Figs. RC2-1 to RC2-6. Other than the sample grid for bare mCASTblack soot, TEM sample grids were collected on the same date and the same condition as for soot samples presented in Fig. 9 in the main text. The figures below should provide more confidence that the images shown in the manuscript are representative. Given obtaining TEM and SEM images is time consuming and resource costly (time due to shared facility, expensive equipment), we obtained as many images as possible within the above constraints to convince us that the sample we produce is represented by these images.

[Figure]

**Figure RC2-1. TEM images (microscope TFS F30) for 200 nm size selected bare mCASTblack soot particles. Scale bars are indicated in each image.**

[Figure]

**Figure RC2-2. TEM images (microscope TFS F30) for 200 nm size selected mCASTblack with a thin coating (coating *wt* = 2.9 %). Scale bars are indicated in each image.**

[Figure]

**Figure RC2-3. TEM images (microscope TFS F30) for 200 nm size selected mCASTblack with a thick coating (coating *wt* = 30.2 %). Scale bars are indicated in each image.**

[Figure]

**Figure RC2-4. TEM images (microscope TFS F30) for 200 nm size selected bare FW200 soot particles. Scale bars are indicated in each image.**

[Figure]

**Figure RC2-5. TEM images (microscope TFS F30) for 200 nm size selected FW200 with a thin coating (coating *wt* = 2.3 %). Scale bars are indicated in each image.**

[Figure]

**Figure RC2-6. TEM images (microscope TFS F30) for 200 nm size selected FW200 with a thick coating (coating *wt* = 139.3 %). Scale bars are indicated in each image.**

Fourthly, regarding possible chemical reactions on soot surfaces with $H_2SO_4$ coating as the carbon to oxygen ratio before and after coating was changed (Fig. 10 in the manuscript), detailed chemical reactions or mechanisms (L581, now L589 in revised manuscript) are out the scope of this study which focuses on the ice nucleation of soot particles. We infer that a reaction took place based on the O:C ratio, without designing the experiment to study the reaction between acid and soot. As such it would be completely speculative to state anything more than is already stated in the manuscript.

Multiple charged soot particles persist in soot samples when using a DMA (differential mobility analyzer) because of the nature of the charge theory. Moreover, the heterogeneity and fractal nature of soot particles lead to a decreased resolution for soot particle size selection. To our knowledge, this is a recalcitrant issue to the use of DMAs for size selecting soot particles (Burkert-Kohn et al., 2017; Mahrt et al., 2018; Nichman et al., 2019; Zhang et al., 2020) in combination with ice nucleation. This is also

the reason why we presented size selection quality evaluation for each sot sample used in this study. Now, we add a statement in Appendix A after the results for particle size selection evaluation (L820-821 in revised manuscript), as below:

'*It is likely that these double-charged particles of larger size than the size selection value first nucleate ice during the RH scan experiments, given the size dependence of soot ice nucleation.*'

Finally, this study focuses on how the mixing state of soot particle with $H_2SO_4$ modifies the ice nucleation ability of coated soot particles. In each soot particle activation fraction plot (Figs. 4-7 and B1-B8) as a function of RH (relative humidity), the homogeneous freezing RH of solution droplets ($H_2SO_4$ solution), calculated based on the water activity parameterization presented by Koop et al. (2000), was indicated and used to compare the ice nucleation ability of soot particles (Sect. 3.1). In Sect. 4, we evaluated the atmospheric impacts of coated soot particles, compared to the homogeneous freezing of droplets, e.g. $H_2SO_4$ (L679 and L691, now L689-690 and L705-706 in revised manuscript).

- **Minor:**

**1    L1: 'Activities' replace with activity. Also, I'd recommend to amend the title to include the bare soot measurements and to clarify that these are laboratory experiments. Also see comment L625.**

R:    Thanks. The title is changed to '*Laboratory studies of ice nucleation onto bare and internally mixed soot–sulphuric acid particles*'.

**2    L10: There is a problem with this sentence, if they are only candidates, what makes them so important? Perhaps use potential instead of important.**

R:    Thanks for your suggestion. We changed it to "potential candidates" (now L10 in revised manuscript).

**3    L10: comma after formation**

R:    Thanks. In this sentence, it is cirrus clouds that finally exerts a net-warming effect on the climate. Therefore, we think we do not need a comma here.

**4    L14: I think a sentence of justification is missing here for the choice of propane flame soot or a commercial BC. e.g. 'Generic BC surrogates are often used in laboratory experiments. In this study….'.**

R:    Thanks. This sentence was changed to '*In this laboratory study, two samples, a propane ($C_3H_8$) flame soot and a commercial carbon black were used as atmospheric soot surrogates and coated with varying wt % of sulphuric acid ($H_2SO_4$)*.' (now L14 and L15 in revised manuscript)

**5    L16: activities - in line 13 you say ability, but study activity? Please keep consistency or explain the differences throughout the text between ability and activity.**

R:    Both 'ice nucleation ability' and 'ice nucleation activity' are used to describe soot particle ice nucleation in simulation and laboratory studies (Koehler et al., 2009; Lupi and Molinero, 2014; Mahrt et al., 2020b; Mahrt et al., 2020a; Zhang et al., 2020; Kilchhofer et al., 2021).

**6    L25: abilities – same comment.**

R:    See response above.

**7**     **L35: please add https://doi.org/10.1029/2021JD034649 to Liu et al., 2020.**

R:     Thanks for the suggestion. Agreed and added (now L35 in revised manuscript).

**8**     **L36: The numbers cited here are from a review by Bond et al. published 8 years ago and data collected more than a decade ago. Please see if there are additional newer reports. Otherwise, short-lived climate forcers suggest a continuous decline in carbonaceous emissions. Moreover, there is a lack of accurate global quantification of this short lived element (IPCC AR6 report, Chapter 6). These caveats should be mentioned.**

R:     Thanks a lot for this suggestion. We cited some recent studies reported in IPCC AR6 report and added two sentences as following (L37-39 in revised manuscript):

'*Recently, some studies suggested that emission-based radiative forcing from BC has been reduced (Takemura and Suzuki, 2019; Lee et al., 2021) because of carbon emission mitigation. However, the estimate is of a low confidence (Mcgraw et al., 2020).*'

**9**     **L38: aircraft plural is still aircraft**

R:     Thanks. Agreed and corrected (L40 and 43 in revised manuscript)

**10**     **L65: '…vigorous convection' – reference missing.**

R:     Thanks. Relevant references were added as following (L68 and 69 in revised manuscript):

'*In addition, some soot particles, generated by incomplete combustion from natural and anthropogenic sources contaminated by sulphur material during industrial processes, can get advected to the upper troposphere by vigorous convection (Pósfai et al., 1999; Okada et al., 2005; Motos et al., 2020).*'

**11**     **L69: abilities – see comment for L16 and throughout the manuscript**

R:     See response to comment 5 above.

**12**     **L72: 'potentially… potential', I suggest to reword. For example: Thus, it is possible that the mixing state of soot particle with $H_2SO_4$ coating may regulate its INP activity.**

R:     Thanks. Agreed and accepted. The sentence was restructured as following (see L76 and 77 in revised manuscript):

'*Thus, the internal mixing of soot particle with $H_2SO_4$ may regulate its ability to be a potential INP but the mixing state is unconstrained.*'

**13**     **L75: remove abilities, remove in-situ**

R:     Thanks. Agreed and accepted. The sentence was change to be as following (see L79 in revised manuscript):

'*Ice nucleation of aviation soot particles and their surrogates have been investigated both in field measurements (Brown, 2018) and laboratory studies.*'

**14**     **L75: some references could be useful here e.g. Brown et al. DOI: 10.2514/6.2018-3188**

R:     Agreed and added (L79 in revised manuscript, also see response to comment 13 above).

**15**     **L76: please include some references for lab studies of aviation soot.**

R:   Laboratory studies on the ice nucleation of aviation fuel generated soot particles (Popovicheva et al., 2004; Koehler et al., 2009) were already introduced in the next paragraph (now in L90 and L98 in revised manuscript). However, ice nucleation studies of aviation soot emitted from real aircraft engines are lacking.

**16   L77: 'contrail dissipation' – please add reference**
R:   A relevant paper is cited in L81 (in revised manuscript) and is given below:
Kärcher, B., Kleine, J., Sauer, D., and Voigt, C.: Contrail Formation: Analysis of Sublimation Mechanisms, Geophys.Res. Lett., 45, 10.1029/2018gl079391, 2018.

**17   L77: stick with one form 'in-situ' like in line 75 or 'in situ'.**
R:   Agreed, we decided to use "in situ" and applied this form to the whole manuscript. (L81 in revised manuscript)

**18   L88: in line 85 you talk about aviation soot particle surrogates then here you mention graphite, it is not clear how graphite is a surrogate of aviation soot and how this is relevant here.**
R:   Soot particles are of different level of graphitization. The study of graphite ice nucleation is of similarity to that of soot particles and the results can help understand how soot particle graphitization level changes its ice nucleation ability.

**19   L90: Mahrt et al. 2018 – please replace with Marcolli C. 2014 and others. I believe it was suggested also much earlier by others. Marcolli, C.: Deposition nucleation viewed as homogeneous or immersion freezing in pores and cavities, Atmos. Chem. Phys., 14, 2071-2104, 2014.**
R:   Marcolli (2014) suggested PCF may be the possible pathway for soot ice nucleation and Mahrt et al. (2018) performed ice nucleation experiments for soot particles with different porosity and surface wettability below homogeneous freezing temperature and demonstrated PCF is responsible for soot ice nucleation. (L94-97 in revised manuscript)
*'Mahrt et al. (2018) demonstrated that pore condensation and freezing (PCF) (Koehler et al., 2009; Marcolli, 2014) rather than deposition nucleation is responsible for soot particle ice nucleation activities, given that porous soot particles are able to form ice crystals at RHi values lower than homogenous freezing conditions at T < HNT.'*

**20   L116: I'd remove the text after wettability and refrain from mentioning contact angle in the manuscript for two main reasons: First and foremost, you don't report or use contact angle measurements in this study. Secondly, the concept of contact angle measurements and inference to nano-scale dynamic processes is questionable to the least. No surface energy component calculations are truly quantitative nor are they necessarily based upon universally accepted theoretical considerations (see Marmur et al. 2017; Marmur 2006; Strobel&Lyons 2011). Most theories of solid surface energy have a basis in Young's equation, which employs the equilibrium contact angle. In surface energy calculations, many in the surface science field tend to use the so-called static contact angle, which we now know to be meaningless (Marmur et al. 2017), or else the advancing angle, assuming that one of these angles is the equilibrium angle or at least very close to the equilibrium angle. However, the equilibrium contact angle**

cannot be determined on practical BC surfaces. Contact angle is indeed a parameter that can reflect surface functionality however it was and continues to be a highly biased method for getting absolute characterization of the surface. If experimentally measured, it is often highly dependent on the methodology of the measurement. Hoose and Mohler (2012) suggested to use a probability distribution function for contact angle. Marcolli (2016) when using contact angle ignored the correction for the size dependence of surface tension and assumed a zero contact angle, she also mentioned that 'accurate values for the contact angles between water and the pore walls of our investigated particles are not available'.

Marmur, Abraham, Claudio Della Volpe, Stefano Siboni, Alidad Amirfazli, and Jaroslaw W. Drelich. "Contact angles and wettability: towards common and accurate terminology." Surface Innovations 5, no. 1 (2017): 3-8.

Marmur, A.: Soft contact: measurement and interpretation of contact angles, Soft Matter, 2(1), 12–17, doi:10.1039/B514811C, 2006.

Strobel, M. and Lyons, C. S. (2011), An Essay on Contact Angle Measurements. Plasma Processes Polym., 8: 8-13. doi:10.1002/ppap.201000041

Marcolli, C. (2016), 'Pre-activation of aeosol particles by pore condensation and freezing', Atmos. Chem. Phys., 2016, 1-48.

Hoose, C, and O Möhler. "Heterogeneous Ice Nucleation on Atmospheric Aerosols: A Review of Results from Laboratory Experiments." Atmospheric Chemistry and Physics 12, no. 20 (2012): 9817-54.

R:    Thank you for this comment. We agreed that contact angle cannot be accurately measured and the contact angle over heterogeneous soot powder surface perhaps has a distribution other than a single value. However, a wettable surface always shows a lower contact angle value and this relative outcome remains unchanged. Here in L116 in the original manuscript (L120 in revised manuscript), we use contact angle as a measure for relative soot surface wettability. In addition, the inverse Kelvin effect occurrence, which is an important step for soot ice nucleation via pore condensation and freezing (PCF), relies on the contact value according to Kelvin equation. From the literature (Marcolli, 2014; David et al., 2019; David et al., 2020; Marcolli, 2020; Marcolli et al., 2021), contact angle is used for PCF description and parameterization other than wettability. Marcolli et al. (2021) noted that contact angle can be viewed as a measure to quantify particle surface wettability and enumerated two cases (contact angle 180 and 0º) from Lohmann et al. (2016) to present the role of contact angle in surface wettability. To stay coherence with the literature and understand soot particle ice nucleation via PCF, we think it is necessary to mention the contact angle.

**21    L127: replace 'need' with 'could benefit from'**

R:    Thank you. Agreed (see L131 in revised manuscript)

**22    L134: in the abstract you also mention mixed-phase, but the conclusions are only about ice nucleation in cirrus clouds?**

R:    Thanks for the comment. We discussed and concluded that both uncoated and coated soot particles cannot nucleate ice in the mixed-phase cloud regime (see Sect. 3 in L349-350 in revised manuscript). Now, a statement is added in the conclusion part (see Sect. 5 in L751-752 in revised manuscript) as following:

'*For T > HNT, i.e. in the mixed-phase cloud regime, soot particles with/without H₂SO₄ coating do not*

*nucleate ice.*'

**23  L137: The samples are not 'experimental' perhaps 'test samples'?**

R:  We agree, and now, we use 'soot samples' instead of 'experimental samples (see L141 in revised manuscript).

**24  L142: Switch places between the name and the HINC acronym like you did in line146 for DMA**

R:  Thanks. Agreed and changed (see L146-147 in revised manuscript).

**25  L143: part of the exhaust air**

R:  We mean exhaust and not specifically exhaust air, since the exhaust contains air and soot particles, as such we do not believe exhaust air suggested by the reviewer reflects what we mean to say.

**26  L148: mobility diameter – change throughout the manuscript.**

R:  Thanks. Agreed and changed (L152 in revised manuscript).

**27  L164: remove 'Experimental'**

R:  Thanks. Agreed and removed (L167 in revised manuscript).

**28  L166: mention it's a diameter in this sentence.**

R:  Thanks. The sentence was corrected as following (L169-170 in revised manuscript):

'*Firstly, the DMA selects soot particles with a mobility diameter of 200 nm (sheath to sample flow ratio 13 : 1) or 400 nm (sheath to sample flow ratio 7 : 1) and…*'

Original sentence: '*Firstly, the DMA selects 200 nm (sheath to sample flow ratio 13 : 1) or 400 nm (sheath to sample flow ratio 7 : 1) soot particles and…*'

**29  L167: The aerosol sample – does it contain only the soot aerosol? Are there byproducts in the propane combustion process e.g. gases/vapors? If there are any, where do they go, do they follow the aerosol coating and IN path? Would this direct injection affect your measurements?**

R:  The combustor exhaust partitioning will influence soot particle properties. For example, Corporan et al. (2008) suggested the aircraft engine particulate and gaseous emissions vary among different sampling positions, implying that soot samples downstream of the soot generator with different ageing time may have different properties and ice nucleation abilities. In this study, we used a 125 L mixing volume to allow soot particle evolution and mixing with gaseous species and to generate a more homogeneous soot particle population before ice nucleation experiments and sample characterization. Before sampling size selected soot particles, water vapor was removed by a diffusion dryer (down to < 5 % $RH_w$, see L149-152 and L247 in revised manuscript).

**30  L168: home change to in-house**

R:  Thanks. Agreed and changed (L172 in revised manuscript).

**31  L174: CPMA - see comment L142**

R:  Thanks. Agreed and changed (L179 in revised manuscript).

**32    L175: sucked changed to pulled**

R:    Thanks. Agreed and changed (L180 in revised manuscript).

**33    L176: microscopic grids change to microscopy grids**

R:    Thanks. Agreed and changed (L181 in revised manuscript).

**34    L177: Rephrase. Not clear distinction from what? Other studies in this field? Your previous studies? Is the 'generation of coated soot...and the real time analysis of their properties...' are the only novelty components of this study? why is it important to mention this distinction? Also see my major comment on the significance/novelty of this study.**

R:    This sentence was changed to the following (L181-183 in revised manuscript):

'*The purpose of this study is generating $H_2SO_4$ coated size-selected soot particles and studying their physical properties including mobility diameter, mass (density), morphology and ice nucleation ability.*'

**35    L178: not sure what online means here? Real-time?**

R:    It was deleted (from L182 in revised manuscript) and the sentence was revised. Please see response to comment 34.

**36    L179: see comment in L16.**

R:    Please see response to comment 5.

**37    L186: finally change to consequently**

R:    Thanks. Agreed and accepted (L190 in revised manuscript).

**38    L191: LabVIEW**

R:    Thanks. Agreed and accepted (L195 in revised manuscript).

**39    L207: indicating no pure nucleated $H_2SO_4$ - add reference to subsection where these results are presented.**

R:    Thanks. Agreed and accepted we added the reference to Appendix A (see L211 in revised manuscript).

**40    L211: becomes absent - what does that mean? There is no longer homogeneous nucleation or the mode moves below your detection threshold or rate of coagulation is higher than homogeneous nucleation?**

R:    We mean the absence of small size mode (~ 40 nm) of $H_2SO_4$ read directly from the Fig. A1. When soot particles present in the nucleation and condensation process of $H_2SO_4$ vapor, soot particles serve as nuclei and significantly decrease the $H_2SO_4$ vapor pressure by depleting the acid vapor. Therefore, nucleation mode $H_2SO_4$ and coagulated pure small $H_2SO_4$ particles number concentration can be dramatically decreased. We have now rephrased "becomes absent" to "is absent" (see L215 in revised manuscript)

**41    L211: decreases dramatically - please elaborate, is it the difference between nucleation and coagulation rates? What portion coagulated and how much condensed? have you tracked the**

**temperature in this coating process? how mixing the flows with different temperature affects the evaporation rate of H₂SO₄? could it affect the repeatability?**

R: Thanks for the comment. As in L211 in the original manuscript, it is the number concentration of pure H₂SO₄ particles that decreases progressively and after 20 minutes the small mode ~ 40 nm is completely absent. To make this clear, we have now rephrased the sentence to read '*…and the number concentration of H₂SO₄ particles reduces to effectively zero for the ~40 nm peak (see Fig. A1)*' (see L215 and 216 in revised manuscript)

As shown in Fig. 2 and described in Sect. 2.2.1, the temperature of H₂SO₄ vapor flow, dilution flow and cooling system was continuously monitored and maintained for each soot sample with a H₂SO₄ coating *wt* %. The flow rates for each stream were controlled as indicated in Fig. 2. By increasing the H₂SO₄ vapor saturation and dilution flow temperatures, more H₂SO₄ vapor can be generated and the coating thickness for per coated soot sample can be controlled. To demonstrate the repeatability, the coated particle mass and mobility diameter were monitored during the course of ice nucleation experiments, and ice nucleation tests were also performed by conducting two RH (relative humidity) scans at per fixed temperature. In addition, each size selected soot sample was coated with at least 8 different coating thicknesses and the coating thickness increase monotonically with increasing saturation flow and temperature (see Table 1 and Fig.3).

**42   L215: we are confident change to we conclude**

R: Agreed and changed (see L220 in revised manuscript).

**43   L220: a known changed to defined?**

R: Agreed and changed (see L224 in revised manuscript).

**44   L252: 'and' change to 'so that'.**

R: Agreed and changed (see L256 in revised manuscript).

**45   L258: what's the reasoning for using a water CPC? would you expect differences in comparison to butanol CPC for coated and bare soot counting?**

R: For particle mass measurement by using the CPMA (centrifugal particle mass analyzer), a high flow rate CPC (condensation particle counter) is required to reach a better resolution than a low flow rate CPC. Therefore, water CPC 3787 (Model 3787, TSI Inc.) with 1.5 L min⁻¹ flow rate was used. Before the experiment, we compared our butanol CPCs 3776/3772 and water CPC 3787 by measuring the same soot sample (both propane flame soot and FW200 black carbon) with/without H₂SO₄ coating. The differences for particle counting from these CPCs were less than 10 %.

**46   L260: see comment L148**

R: Agreed and changed (see L264 in revised manuscript).

**47   L262: remove mathematical**

R: Agreed and removed (see L266 in revised manuscript).

**48   L262: see comment L148**

R:    Agreed and changed (see L266 in revised manuscript).

**49    L266: remove 'by the'**

R:    Agreed and removed (see L270 in revised manuscript).

**50    L274: see comment L148**

R:    Agreed and changed (see L278 in revised manuscript).

**51    L289: What is the reasoning for bringing this issue up here if there is no solution provided? Stating that it provides still relatively comparable information doesn't resolve the issue. This should be moved into discussions of uncertainties in the study. What type of BET was used for the specific surface area of your samples?**

R:    Here, we presented particle size, mass (effective density) and equivalent monolayer changes in Fig. 3 to show the coating effect on soot particle morphological properties. This helps the understanding and discussion of the coating effect on soot particle ice nucleation. More detailed coating effects on soot particle mixing states and ice nucleation were discussed in Sect. 3. We used the $N_2$ (nitrogen) based BET (Brunauer-Emmett-Teller) surface area for uncoated soot samples reported by Mahrt et al. (2018) (see L275 in revised manuscript).

**52    L292: Whereas 400 nm size selected soot particle does not show apparent size growth – why?**

R:    After L292 in the original manuscript (L297-300 in revised manuscript), a new sentence is added to explain this.

'*This may result from a lower effective density of 400 nm soot particle compared to that of 200 nm soot particle, which suggests larger soot particles are less densified and contain more pore volume for $H_2SO_4$ filling rather than surface accumulation of the acid which would lead to a detectable size increase.*'

**53    L302: home – change to in-house**

R:    Agreed and changed (L309 in revised manuscript).

**54    L331: see comment in L16.**

R:    See response to comment 5.

**55    L338: what aerosol is used to calibrate the 780nm diode in the OPC? In appendix C, have you evaluated the response of the OPC to different soot types and diameters directly without passing through the chamber? Is it possible that the OPC mis-sizes non spherical, light absorbing soot with higher refractive index with a complex component? The coated soot is often studied for its enhanced light absorption and other optical property changes, would that affect the OPC sizing? Bhandari et al. 2019 wrote that compaction affects the soot optical properties. Light absorption and scattering change when a soot particle undergoes morphological transformations. The recommended operating temperature of the OPC by the manufacturer is 0º C to +50º C, what are the expected biases in detection and sizing for air/aerosol flows at low temperature, down to -55? Would you expect humidity condensation on OPC windows?**

R:    Our GT-526S OPC (optical particle counter) was calibrated with PSL particles by MET ONE

Instrument Inc. before the experiments. In Appendix C, we used the OPC to measure 200 and 400 nm size selected bare mCASTblack soot particles directly without passing through the chamber to demonstrate that majority of the size selected unactivated soot particles are detected in the size bin below 1 μm. As we do not use the OPC to detect soot particles directly but rather use it to detect nucleated ice crystals larger than 1 μm, we do not believe mis-sizing the soot particles in the range below 1 μm is an issue. As long as unactivated soot particles are not detected in the OPC > 1 μm channel, we do not expect interference of signal. Finally for such small freshly nucleated ice crystals (1 – 5 μm), we believe the ice crystals appear almost spherical (Mahrt et al., 2019) in HINC. As such we do not expect significant biases from asphericity. More details on the characterization of sizes are available in Lacher et al. (2017) and Mahrt et al. (2019).

We did not observe vapor condensation on the OPC internal optics. If this was to occur, it typically results in no counts or unchanged counts over time and thus an indication for cleaning and recalibration. Furthermore, the OPC optics will be at temperatures warmer than the sample temperature, as such condensation is not likely to occur. Detection is merely based on being at size larger than the injected soot particles, so as long as the ice crystals do not shrink to below 1 μm, our counts should not be affected. It is possible that the OPC mis-sizes the particles – however, we are not using the OPC to report a size distribution as such this should not affect our results. The OPC is merely used to separate small unactivated soot particles from larger ice crystals.

**56   L360: see comment L116. You can cite here the earlier studies. It can be attributed to wettability but not necessarily contact angle.**

R:   See response to comment 20.

**57   L360: There is no mention of the solvents/organic content pre-existing in FW200 as indicated by the manufacturer. Volatile content gas chromotography or DTA-TGA analysis could shed some light on the chemistry or its *wt* %. Without it, some discussion is needed about the possible properties of this content and whether glass transition of those organic compounds could impact the IN activity in these temperatures.**

R:   We in L348 and L353 (in original manuscript, now in L355 and L360 in revised manuscript) refer to Mahrt et al. (2018) who studied 200 and 400 nm bare FW200 soot particles generated by a fluidized bed aerosol generator (FBG, Model 3400A, TSI Inc.) The authors also conducted thermogravimetric analysis (TGA, model Pyris 1 TGA, PerkinElmer) and suggested that FW200 shows ~ 7 % mass losses for water vapor and/or low-molecular-weight organics below 100 ℃ and there is no significant mass loss for FW200 between 100 and 400 ℃ (Mahrt et al., 2018). In addition, flushing FW200 sample at 25 ℃ for 1000 min in $N_2$ flow leads to a ~ 9 % mass loss, suggesting the mass loss during TGA measurement below 100 ℃ is due to loss of adsorbed water (Mahrt et al., 2018). The TGA results suggest that organic content is negligible for this soot sample.

**58   L409: Is there a possibility of homogeneous nucleation of the acid and coagulation with soot, Coagulation is strongly affected by residence time. would that be possible on these time scales?**

R:   We addressed in L187 and L188(in original manuscript, now in L189 and L190 in revised manuscript), the acid coating process onto soot particles may include the direct condensation of supersaturated $H_2SO_4$ vapor and the adsorption of small $H_2SO_4$ particles formed by homogeneous nucleation (Bambha et al., 2013; Pei et al., 2018). We also measured the particle size and mass

distribution of coated soot particles as presented in Appendix A and concluded that there is no pure acid droplet in coated soot particle aerosol. Furthermore, we did not observe any single acid droplet (smaller than bare soot particle mobility diameter or comparable) in presence in the microscopy images.

**59    L413: T – spell out temperature throughout the manuscript.**

R:    Thank you for your suggestion. As temperature is a variable, we use its acronym and use italics to indicate it, following the ACP manuscript guideline (*https://www.atmospheric-chemistry-and-physics.net/submission.html#templates*).

**60    L413: inhibits – change to 'could be the main cause for inhibition of the'.**

R:    Thanks for your suggestion. We prefer to leave it as is (see L418 in revised manuscript).

**61    L414: see comment L413.**

R:    See response to comment 59 (see L418 in revised manuscript).

**62    L414: From – change to 'In'.**

R:    Agreed and changed (see L419 in revised manuscript).

**63    L415: shows coherence- change to 'is coherent'.**

R:    Agreed and changed (see L420 in revised manuscript).

**64    L429: see comment L413.**

R:    See response to comment 59 (see L435 in revised manuscript).

**65    L430: see comment L413.**

R:    See response to comment 59 (see L436 in revised manuscript).

**66    L440: more uniformly - doesn't that contradict what you wrote in L405?**

R:    This does not contradict to the content in L405. Mahrt et al. (2020a) generated $H_2SO_4$ coated soot particles by dispersing soot samples in aqueous bulk acid solution. The acid can be coated over soot surfaces upon water evaporation and the solute (acid) can be distributed more uniformly than that in this study.

The statement now is in L446-448 in revised manuscript (also as below):

'*Furthermore, the $H_2SO_4$ coating generated in this way can be distributed more uniformly over the soot particle surface, compared to a nonuniform $H_2SO_4$ coating in this study (see Sect. 3.3).*'

**67    L455: see comment L116. I think this proves further that the contact angle measurement is not the best choice of technique to include in such studies.**

R:    According to the Kelvin equation, both small enough pore size and contact angle are required to induce the inverse Kelvin effect, i.e. capillary condensation (Marcolli, 2014). The statement in L455 (in original manuscript, now in L459-461 in revised manuscript) suggests only if there exists pore volume capable of inducing capillary condensation can soot particle surface with a low enough contact angle facilitate its ice nucleation via pore condensation and freezing (PCF). In other words, this suggests contact angle is the second most important factor for soot PCF activation. However, if the contact angle

is too large (> 90º), the inverse Kelvin effect will not be fulfilled and capillary condensation of supercooled water will not be possible for PCF.

**68  L460: see comment L413.**

R:  See response to comment 59 (see L466 in revised manuscript).

**69  L461-462: not clear who you are citing here Koehler 2009 or Henson 2007?**

R:  Thanks for checking this. We cited Koehler et al. (2009) for their report on that hydrophilic aviation soot particles need relative humidity (RH) conditions higher than the Kelvin RH limit to activate as water droplets. Herein, the Kelvin RH limit we referred to the work reported by Henson (2007) (see L466-469 in revised manuscript).

'*The AF curves of bare and coated FW200 soot particles in Figs. 6 and 7 are consistent with the study conducted by Koehler et al. (2009) who reported that the CCN activation of hydrophilic aviation kerosene soot particles require higher RH conditions than the Kelvin RH limit (Henson, 2007), which is required by wettable particles to show CCN activation.*'

**70  L463: doesn't that contradict L440?**

R:  In L440, we described the work from Mahrt et al. (2020a) who dispersed propane flame soot samples in aqueous acid bulk solution to generate soot particles with more uniform $H_2SO_4$ coating than $H_2SO_4$ coated propane flame soot particles in this study. Whereas L463 (in original manuscript, now in L466-469 in revised manuscript) describes the activation curves of bare and coated FW200 soot particles in Figs. 6 and 7. Therefore, there is no contradiction.

**71  L478: soot aggregate - how do you define an aggregate here? is it a single aggregate of spherules, please elaborate. Is it consistent with definitions by Long et al. 2013?**
**Long, C. M., Nascarella, M. A., and Valberg, P. A.: Carbon black vs. black carbon and other airborne materials containing elemental carbon: Physical and chemical distinctions, Environ. Pollut., 181, 271–286,**
**https://doi.org/10.1016/j.envpol.2013.06.009, 2013**

R:  In this study, FW200 particles are generated from a typical carbon black material (see L154-155 in revised manuscript) and thus a FW200 soot-aggregate (in L478 in original manuscript, now in L484 in revised manuscript) is defined as a discrete, rigid and individual entity of extensively coalesced primary particles. And it is consistent with the definition given by Long et al. (2013) who noted that a carbon black particle consists of acini-form (grape-like) aggregates of highly fused spherical primary particles.

**72  L480: see comment in L148. Small particles with a small mobility size contain less pore volume - Is this generalization of mobility diameter accurate? Doesn't that depend also on spherule size? Did you mean 'Small FW200 soot particles'?**

R:  Yes, here we mean small FW200 soot particles, i.e. a 200 nm mobility diameter. Now, we clarify this in L486 in revised manuscript.

**73  L489: In this interpretation of observation, I couldn't find a discussion about the effect of cooling rate, and hydration of the sulfuric acid on its glass transition and changes around the glass transition range, see Zobrist et al 2008; Williams & Long 1995 and others. Also see my**

**last Major comment.**

**Zobrist, B., Marcolli, C., Pedernera, D. A., and Koop, T.: Do atmospheric aerosols form glasses? Atmos. Chem. Phys., 8, 5221–5244, https://doi.org/10.5194/acp-8-5221-2008, 2008.**

**Williams, Leah R., and Forrest S. Long. "Viscosity of supercooled sulfuric acid solutions." The Journal of Physical Chemistry 99.11 (1995): 3748-3751.**

R:    Thanks for the comment. The aerosol sample was cooled down to the RH (relative humidity) scan experiment temperature ($218 < T < 243$ K) in a few seconds before ice nucleation. During this fast transit period, soot particles only experience a low RH condition since a diffusion drier is used upstream of the ice nucleation chamber to decrease the $RH_w$ down to $< 5$ % (see L151-152 in revised manuscript). The actual RH scan experiment for $H_2SO_4$ coated soot particles was conducted at a fixed $T$ with a $RH_i$ slope 2 % min$^{-1}$ from $RH_i = 100$ % to $RH_w > 110$ %, during which the ice nucleation will actually occur. Therefore, the fast cooling process occurs prior to ice nucleation events and does not affect soot ice nucleation in this study.

Secondly, according to Williams and Long (1995) and references therein (Vuillard, 1954; Eicher and Zwolinski, 1971), the glass transition temperature for aqueous $H_2SO_4$ solution of different concentration is less than 180 K which is below the temperature ($> 218$ K) for ice nucleation experiments in this study. Zobrist et al. (2008) also suggested that the glass transition temperature of aqueous solutions of $H_2SO_4$ is too low to be of atmospheric importance. Frey et al. (2013) also reported that the glass transition temperature of $H_2SO_4$ solution of $\sim 65$ *wt* % is as low as 164 K and the transition temperature decreases with decreasing $H_2SO_4$ *wt* %. Therefore, the glass transition is not a possible case in this study.

However, the hydration of coated $H_2SO_4$ in soot-aggregate structures has a lower water activity can lead to depressed homogeneous freezing, which inhibits soot particle ice nucleation, as discussed in Sect. 3.1.

**74   L506: sentence too long.**

R:    Thanks for this comment. The sentence is restructured as following (now L511-512 in revised manuscript):

'*Here we use the single particle size and mass results coupled with microscopy of the particle mixing state and the ice nucleation activities of $H_2SO_4$ coated soot particles to propose a hypothesis on the internal mixing state of $H_2SO_4$ and soot particles.*'

The original sentence is as following:

'*Coupling single particle size and mass measurement results with microscopy of the particle mixing state characterization results, as well as with the ice nucleation activities of $H_2SO_4$ coated particles with different coating wt %, a hypothesis depicting the internal mixing states of $H_2SO_4$ and soot particles is proposed.*'

**75   L506: remove 'measurement results'**

R:    See response to comment 74 (L511-512 in revised manuscript).

**76   L506: remove 'characterization results'**

R:    See response to comment 74 (L511-512 in revised manuscript).

**77   L516-519: is that the 'collapse' you described in L293? it sounds less obvious here.**

R:    Yes. The sentence is changed as following (L522-523 in revised manuscript):

'*In addition, thickly coated 400 nm mCASTblack soot-aggregates tend to* collapse by showing *a more compacted and less fractal 2D projection shape, compared to bare and thinly coated aggregates*.'

**78   L522: fractal dimension - have you explained what this is or how 1.86 was calculated?**

R:   A statement for the meaning of fractal dimension was added after this sentence in L522 (in original manuscript, now L527-530 in revised manuscript), as below:

'*The particle fractal dimension is used to quantitatively describe soot particle morphology, considering that the number of primary particles scales as a power law with the radius, thus a chain agglomerate would have a fractal dimension of 1 and a sphere of primary particles would have a fractal dimension of 3.*'

**79   L524: coating does not result in a significant soot surface topography change – it doesn't look like there is any morphological difference between 8d, e, f except the addition of a large amount of acid. No evidence to indicate a collapse mentioned in L293 and shown in Fig. 3c.**

R:   As addressed in L523-525 (in original manuscript, now L530-532 in revised manuscript), '*There are no distinguishable morphological feature differences between Fig. 8d and e, which means thin $H_2SO_4$ coating does not result in a significant soot surface topography change* with a 1.8 % $H_2SO_4$ coating mass.'. However, a larger $H_2SO_4$ coating mass ($\sim$ 65.0 % as for Fig. 8f) can lead to a part of soot aggregate immersed in the $H_2SO_4$ droplet and results in particle size shrinkage comparing the scale bar in Fig. 8d and f.

**80   L529: resulted from the impaction of soot-aggregates on the Cu grid - how do you know that? Further in the text, you've discussed possible other biases of sample transfer, of measurement with SEM in vacuum. Also, how would electron beam interaction with nonconductive particles affect the sample and imaging?**

R:   A ring of spray in Fig. 8f results from the impaction of soot-aggregate with thick $H_2SO_4$ coating on the Cu (copper) grid because coated $H_2SO_4$ droplet can be split off from the soot-aggregate. For example, Virtanen et al. (2010) observed solid particle bouncing using an aerosol impactor for electron microscopy sampling. For the case of this study also with an impactor for electron microscopy sampling, solid soot-aggregates can possibly bounce and coated soot-aggregates may sputter coated $H_2SO_4$ when impacting on the Cu grid. Soot particles are conductive and soot particles coated with $H_2SO_4$ are still conductive unless they are completely embedded into the acid droplet which we did not observe from our SEM (scanning electron microscopy) images.

**81   L531: could you also describe what this figure shows in C2a, C2b, C2c?**

R:   Figure C2a, b and c are presenting TEM (transmission electron microscopy) images for 200 nm bare and coated mCASTblack soot particles with a low magnification which are taken from the same sample grid as for TEM images in Fig. 9 but with a large magnification (now L894 in revised manuscript). The description for these 200 nm mCASTbalck soot-aggregates can also be found in section 3.3.1.

**82   L542: Small aggregates can also coagulate while transporting in the aerosol flow - coagulation is highly dependent on residence time, if you suggest coagulation is possible, what about coagulation between soot and small droplets of sulfuric acid below your detection range or those that are neutral in charge? Also do you see this same extent of coagulation with thinly or**

**thickly coated?**

R:    As addressed in L187 and L188 in original manuscript (now L189-190 in revised manuscript), the coating process of acid onto soot particles may include the direct condensation of supersaturated $H_2SO_4$ vapor and the adsorption of small $H_2SO_4$ particles formed by homogeneous nucleation (Bambha et al., 2013; Pei et al., 2018). Here, the adsorption of small $H_2SO_4$ particles (nucleation) can be viewed as soot particles coagulate with $H_2SO_4$ droplets. However, the coagulation of soot particles with larger $H_2SO_4$ particles may not occur because there is no significant particle mass and size change for coated soot particles as shown in Fig. 3 (thin and medium coating *wt* %). Moreover, we demonstrated that there is no pure acid droplet in coated soot particle aerosol (please see Appendix A). Furthermore, we did not observe any single acid droplet (smaller than bare soot particle mobility diameter or comparable) in the microscopy images. Therefore, the coagulation of soot-aggregates with large $H_2SO_4$ particles does not occur.

Secondly, we did not see coagulation for the case of coated mCASTblack soot particles and only observed significant coagulation for bare mCASTblack soot samples because of very high soot particle number concentration ($\sim 3000$ cm$^{-3}$) when sampling, as addressed in Appendix C.

**83    L547: why did you decide to show this aggregate? is that a typical image?**

R:    No, this is not a typical image. Please see more randomly selected images in our response to your second major comment.

**84    L551: some extent aggregate compaction - previously I believe you called it collapse? Consider toning down the previous description.**

R:    We stick to 'collapse' and 'some extent' was deleted (see L558 in revised manuscript).

**85    L554: the shrinkage, collapse, and compaction terminology should be consistent throughout the text. Could you explain the differences between the percentage of decrease in diameter in Fig 3a, and the shrinkage observed in Fig. 8c?**

R:    As discussed in Sect. 3.3.2, the coating effect of $H_2SO_4$ on soot particle morphology depends on soot type and its pristine properties, apart from the coating mass. Therefore, coating induced soot-aggregate collapse is in gradual progression with increasing $H_2SO_4$ coating masses. This is also the reason why we propose a three-step coating process. Now, we stick to 'collapse' for describing soot-aggregate morphology changed induced by $H_2SO_4$ coating through the manuscript (see L297, L522, L558, L560, L663, L670, in revised manuscript). The content around L554 in original manuscript is revised as below (now in L560-562 in revised manuscript):

'*Clearly, a thick coating for FW200 soot particle results in a more collapsed aggregate projection with the edge boundary being smoother than the bare and thinly coated ones (see Fig. 9f). This process of soot-aggregate collapse can also be explained by the decrease in mobility diameter of soot particles with thick coatings shown in Fig. 3c.*'

**86    L559: a supplementary video recording starting with the initial state could provide valuable qualitative information about the structure for the readers.**

R:    Unfortunately, our microscopy does not allow us to take videos.

**87    L569: demonstrated that**

R:    Agreed and changed (L576 in revised manuscript).

**88  L569: shows – change to has**

R:    Agreed and changed (L576 in revised manuscript).

**89  L570: 'to indicate' – change to 'to characterize the'**

R:    Agreed and changed (L577 in revised manuscript).

**90  L571: S – spell out sulfur and all the other elements throughout the text.**

R:    Thanks for your suggestion. We are following the IUPAC Gold Book suggested in the submission webpage at https://www.atmospheric-chemistry-and-physics.net/submission.html and the website at https://goldbook.iupac.org/indexes/exact

**91  L576: see comment L571**

R:    See response to comment 90.

**92  L580: S, O see comment L571**

R:    See response to comment 90.

**93  L581: This may suggest that there is a reaction between mCASTblack soot and $H_2SO_4$ depleting some O content in the process. If you are suggestion that there is a chemical reaction taking place, wouldn't that have an impact on IN activity? On partial vapour pressure? This is in addition to the changes in $H_2SO_4$ viscosity (see comment L489). These processes, which are barely discussed may affect your hypothesis. See Major comments.**

R:    If there is a chemical reaction on the soot particle surface, it does not necessitate influencing soot particle ice nucleation. For example, Friedman et al. (2011) reported that ozonolysis increased soot particle surface oxidation level does not significantly change uncoated soot particle ice nucleation activity. Nichman et al. (2019) also suggested that small amounts of surface oxidation do not change soot ice nucleation activity.

Secondly, the chemical reaction between soot and $H_2SO_4$, if any, perhaps modifies the functional groups on soot surfaces. Some new functional groups may influence the soot-water interaction abilities (Mahrt et al., 2020a) and consequently change the water vapor pressure in the soot nearby environment. Then, referring to vapor pressure change induced $H_2SO_4$ viscosity change and thereof potential impacts on soot ice nucleation change, we attributed this to a decreased water activity of $H_2SO_4$ solution (Koop et al., 2000) inside soot-aggregate pore structures in Sect. 3.1 (see L422-425 in revised manuscript), which leads to depressed homogeneous freezing of supercooled $H_2SO_4$ solution in pores.

**94  L584: see comment L571**

R:    See response to comment 90.

**95  L586: Does that contradict your interpretation of observations in L529? Is it possible that evaporation of coating would induce morphological changes like compaction, similar to droplet evaporation in cloud processes? See also comment to Fig.12.**

R:    Therein, we mentioned that the loss of coating material by electron beam heating may be a possible

case and may partly contribute to a lower EDX (energy dispersive X-ray) AOI (area of interest) detected coating material percentage than the global particle $H_2SO_4$ coating *wt* %. This does not contradict the content in L529 in the original manuscript (now L535-538 in revised manuscript), and this is different from the evaporation of water vapor from droplet activated soot particles, i.e. cloud processing. Firstly, the $H_2SO_4$ coating *wt* % for soot particles in Fig. 10 is small. Whereas cloud processing is the case during which a super micron size droplet evaporates and leaves a residual particle core. Thus, the amount of vapor evaporated during cloud processing is much more than the case in the microscope of a small (few monlolayers) amount of coating material evaporation resulting from electron beam heating. Furthermore, statistical analysis of soot aggregate images does not suggest all cloud processed soot-aggregate morphology can be modified, according to Mahrt et al. (2020b) (Fig.7). Finally, if the evaporation of coating material could significantly change soot-aggregate morphology by compaction, thinly coated soot-aggregate morphology should have also been changed. However, thinly coated soot-aggregate shows similar morphology to uncoated soot-aggregates. Therefore, the statement in L586 (in the original manuscript, now L594 in revised manuscript) does not contradict our interpretation in L529 (in the original manuscript, now L535-538 in revised manuscript) and the evaporation of coating material should not induce morphological changes like compaction.

**96  L590: be $H_2SO_4$ droplets – Fig. 11a? not clear where it's shown.**

R:    We changed the sentence as below (L595-598 in revised manuscript):

'*In Fig. 11a for FW200 soot, intensive S signals (red markings) on coated particle surfaces also suggest random distribution of coating material and that there are preferred sites for the interaction of soot and $H_2SO_4$. Red markings for S attached with and distributed around thickly coated soot-aggregate are shown to be the footprint of $H_2SO_4$ droplets.*'

**97  L596: would be good to attach a datasheet with all the known properties of FW200 provided by the manufacturer as a supplement. The product may be discontinued in the future and the information from the manufacturer will be lost.**

R:    Thanks for your good suggestion. The information provided from the manufacturer for FW200 can be found in Mahrt et al. (2018) in Table 1. As we already mentioned several places in the main text that we used the same black carbon material for FW200 as Mahrt et al. (2018), we did not repeat the information from the manufacturer.

**98  L596: If all the reservations in this paragraph are true, it is not clear how the statement in L614 (robust evidence) is supported. Or how EDX analysis can be used at all to explain any coating observations in this study.**

R:    The EDX (energy dispersive X-ray) results clearly provide the evidence on the presence of $H_2SO_4$ coating for soot particles, shown that there is no S signal for uncoated soot-aggregates and the S signal increases with increasing $H_2SO_4$ coating *wt* % (see Figs. 10 and 11). Moreover, S map from EDX measurement shows the distribution of $H_2SO_4$ coating is not uniform and there are preference sites on soot surfaces for accommodating the $H_2SO_4$ coating material. We consider this robust evidence of the presence of S in the coated particles and absence of S in the bare particles.

**99  L610: Firstly, this is not unequivocally evident from your analysis. Secondly, there could be other possible interpretations that you haven't discussed, see comment L489. I suggest to tone**

**down these statements.**

R:    The comment on L489 (in the original manuscript) is about the interpretation of the $H_2SO_4$ coating effects on soot particle ice nucleation and the glass transition. As response to L489 (see comment 73), glass transition of hydrated $H_2SO_4$ with increasing RH (relative humidity) requires much lower temperatures than the ice nucleation temperatures addressed in this study. In this study, the SEM images are collected for $H_2SO_4$ coated particles before low temperature ice nucleation experiments (see Fig.1 and Sect. 2.2). Considering some other possibilities not addressed in the literature and this study, we tone down these statements as per reviewer suggestion in L610 (in the original manuscript, now in L618 in revised manuscript) as below:

'*Overall, SEM images show that thick $H_2SO_4$ coatings may change soot-aggregate surface topography. Likely, the coating material either can be distributed over the soot surfaces unevenly and change the connectivity of adjacent primary particles or will pile up over primary particle clusters to form chunks and finally form a $H_2SO_4$ shell.*'

**100 L625: This section should be revised or even removed. The title/focus of the manuscript is ice nucleation. Is the hypothesis brought up here is highly relevant to the main topic? Perhaps the title should be changed.**

R:    We agree, and our title is changed to '*Laboratory studies of ice nucleation onto bare and internally mixed soot–sulphuric acid particles*' (also see the response to minor comment 1). We agree that the coating steps are not the main focus of this manuscript but we are able to decipher the mechanism and show that coatings conducted in situ using this method are not complete, given the number of coating experiments conducted across samples and different sizes and coating *wt* %. We consider this valuable information to add to the manuscript and thus prefer to leave this section in the paper.

**101 L645: The fatty acid example is irrelevant to the hypothesis of sulfuric acid coating process. What these two have in common?**

R:    Here, the oleic acid coating process on soot particle surfaces (Garland et al., 2008) is used as an analogy to the case of $H_2SO_4$ coating material state on soot particle surfaces. We did not use it directly to draw any conclusion.

**102 L652: I'm not sure I understand the physics of this process, could you elaborate further? See comment to Fig.12.**

R:    We addressed this in L123-124 in original manuscript (L127-128 in revised manuscript) as '*Schnitzler et al. (2017) also demonstrated that liquid material coating can change soot-aggregate structure by the tension induced by coating condensation on the particle surfaces. It is therefore possible that, if the particle structure can be changed, the pore volume and pore size distribution of the soot-aggregate are also alterable.*' We explained this further when replying the comment on Fig. 12 in below (see comment Nr. 134).

**103 L654: please avoid drama in science, 'supported by the compaction of thickly coated...'.**

R:    We agreed and changed (See L665 in revised manuscript).

**104 L658: remove a dramatic**

R:    Agreed and removed (See L669 in revised manuscript).

**105 L663: robust support from measurements in this study – I'd highly recommend to tone down the statement and properly address the major comments.**

R: We agree and changed the tone of the statement (See L674 in revised manuscript).

'*In brief, our three-stage hypothesis can be supported from measurements in this study and is also in agreement with previous findings in the literature.*'

Original statement:

'*In brief, our three-stage hypothesis has robust support from measurements in this study and is also in good agreement with previous findings in the literature.*'

**106 L674: the sentence here is fairly general and doesn't require citation, unless you want to refer the readers e.g. 'see examples in Kanji et al 2017'?**

R: It was adjusted as below (See L685 in revised manuscript):

'*…the processes soot particles undergo with respect to ageing and cloud processing can be complex (see examples in Kanji et al., 2017)*'

**107 L678: morphological properties and size?**

R: Yes, agreed. Now the statement is changed as below (See L689 in revised manuscript):

'*…ice nucleation ability depending on their size and morphological properties.*'

**108 L688-L697: wouldn't it also depend on the dynamic expansion pulse into the low pressure environment at high altitude and induced supersaturation?**

R: Thanks for this suggestion. Yes, this is true but the dynamic expansion pulse into the low pressure system will affect contrail formation. However, in this paragraph we refer and discuss ice nucleation onto soot particles contributing to cirrus clouds and not contrail formation.

**109 L699: similar to our – change to similar to the FW200 soot sample used here.**

R: Agreed and adjusted (See L716 in revised manuscript).

**110 L705: it is likely that ice crystals can be induced by them – reword.**

R: Thanks for this comment. It was adjusted as following (See L722 in revised manuscript):

'*...they likely nucleate ice homogeneously*'

**111 L712: loses change to loose**

R: Agreed and changed (See L729 revised manuscript).

**112 L719: carbon poor?**

R: It was changed to 'organic poor' (See L745 in revised manuscript).

**113 L774: because a 200 nm pure $H_2SO_4$ particle has a mass of 7.7 fg - what about the bottom edge of the PSD for pure $H_2SO_4$?**

R: The size distribution of pure $H_2SO_4$ was shown in Fig. A1 and indicated by the red line (down to ~ 10 nm), with a size mode of ~ 35 nm.

**114 L805: to be reliably detected in the 5 μm channel - Korhonen et al 2021 selected 6 micron as their bottom threshold. Would lamina flow peripheral regions have slightly different RH conditions?**

Korhonen, K., Kristensen, T. B., Falk, J., Malmborg, V. B., Eriksson, A., Gren, L., Novakovic, M., Shamun, S., Karjalainen, P., Markkula, L., Pagels, J., Svenningsson, B., Tunér, M., Komppula, M., Laaksonen, A., and Virtanen, A.: Particle emissions from a modern heavy-duty diesel engine as ice-nuclei in immersion freezing mode: an experimental study on fossil and renewable fuels, Atmos. Chem. Phys. Discuss. [preprint], https://doi.org/10.5194/acp-2021-111, in review, 2021.

R:    As introduced in Sect. 2.2.2 (L237 in original manuscript, now L241 in revised manuscript), the aerosol particle size was detected in six size channels (0.3, 1.0, 2.0, 3.0, 4.0 and 5.0 μm) of the optical particle counter (OPC). By comparing the activated fraction (AF) curves derived from these different OPC channels for the same type of soot particles, we judged the phase of particles coming out of the ice nucleation chamber by using the methodology presented in Appendix B and referring to (Lohmann et al., 2016; Lacher et al., 2017; Mahrt et al., 2018). Within the particle residence time (~ 10_s) in the ice nucleation chamber used in this study, the largest 5.0 μm OPC channel is enough to differentiate the particle phase (between ice and water, ice can grow to larger than 5 μm but water droplets do not) and shows the most apparent results when comparing with other small channels. Our relative humidity (RH) uncertainty was calculated according to the uncertainty of chamber wall temperature following the same methodology applied by (Lacher et al., 2017; Mahrt et al., 2018) and the error bar was attached for each data point in AF curves for soot particles.

**115 L867: believed - In my opinion EDX is not a suitable technique to analyze organics, it is more suited for the analysis of inorganic materials, rather than materials that contain for the most part all the same organic elements bonded differently. Especially in the case of a commercial FW200.**

R:    We agree that EDX is better to be used to characterize inorganic materials. In L867 (in original manuscript, now L898-900 in revised manuscript), we concluded that the footprint of material residue indicates small $H_2SO_4$ droplets generated upon the impaction of thickly coated soot-aggregate onto the grid by sputtering the acid droplet around, by comparing to the image of uncoated soot-aggregate. Here, $H_2SO_4$ is inorganic material. Moreover, organic material in FW200 soot particles is not in liquid phase and should not form droplet residues. Therefore, it is safe to state '*... some small residues, surrounding single soot-aggregates in Fig. C2f, are believed to be small $H_2SO_4$ droplets generated during the impaction/drying of these soot-aggregates.*'.

**116 L868: impaction/drying - you also mentioned earlier the electron beam evaporated the acid.**

R:    Yes, herein drying means the effect of evaporation by electron beam heating. The statement in L868 in the original manuscript is now changed into (L899-900 in revised manuscript):

'…*are believed to be the footprint of small $H_2SO_4$ droplets generated during the impaction and evaporation of these coated soot particles.*'

**117 L872: particle concentration is high - why this was not addressed during the experiments? This is a known issue, which is controlled by grid size selection, collection time, and flow settings.**

R:    Yes, we identified from the experiments that the sampling time, flow rate, sample particle size and

number concentration determine the grid sample quality. But the exact effect of particle concentration on the grid quality was not clear enough at the beginning. For the first grid collection (uncoated mCASTblack soot), we used a high particle concentration (> 3000 cm$^{-3}$) and a short sampling time ~ 30 s at a 1 L min$^{-1}$ flow rate. After the first grid sample (uncoated mCASTblack) was analyzed, we adjusted the sampling protocol for following grid samples was changed to a low particle concentration (~ 200-300 cm$^{-3}$) but a longer sampling time (~ 600-900 s) still at a 1 L min$^{-1}$ flow rate.

**118 L874: a high probability - I believe the collection efficiency is something that can be calculated. See previous comment. Was the ZEMI instrument calibrated for proper collection in previous studies?**

R:    The collection efficiency of an impactor varies from sample particle properties (see response to comment 117). Direct calculation depending on the flow settings, particle size and collection time is not reliable. For example, Virtanen et al. (2010) reported that the particle phase can affect impactor collection efficiency and there is a bouncing effect from solid phase particles. In this study, H$_2$SO$_4$ coated soot particles are in mixing state and different from uncoated soot particles in solid phase. Thus, the collection efficiency and collection protocol for bare and coated particles are different. As such for ZEMI we use settings known from previous experiments (Mahrt et al., 2018; Mahrt et al., 2020b) and iterate until desired deposition onto the grids is achieved.

**119 L877: 3000 cm-3 - what is the highest number of particles the instrument can size without coincidence bias that will cause mis-sizing?**

R:    Thanks for your question. Here, we understood 'the instrument' refers to an optical particle counter (OPC, MetOne, GT-526S). According to the instrument handbook, the particle number concentration range for the OPC is from 0 to 106 cm$^{-3}$ with a counting accuracy of ±10 % for PSL calibration particles in the size range from 0.3 to 10.0 μm. Here, we take the advantage of the OPC, which is generally used to detect micron size particles, to test whether micron size soot particles exist in our DMA (Differential Mobility Analyzer) size selected 200 and 400 nm soot samples. If we could not detect the presence of micron size particles by the OPC, then we are confident that the micron size soot-aggregates, which we have observed in SEM and TEM images for uncoated soot samples, are accumulated on the microscopy girds upon sample collection but do not represent the soot sample in aerosol phase for ice nucleation experiment. We cannot directly use the OPC to measure such a particle concentration. Hence, we first generated such a high concentration soot sample as it was for microscopy grid sample collection and then diluted it before the OPC inlet for OPC test. After dilution, the soot particle concentration for the OPC test is lower than the coincidence limit of the instrument (106 cm$^{-3}$). Now, we clarified this point in L911 (in revised manuscript) as below.

'*The measurement aerosol flow, also with a particle number concentration ~ 3,000 cm$^{-3}$ measured by a low flow mode CPC 3776 at a 0.3 L min$^{-1}$ flow rate, was first sampled upstream of ZEMI. Then the sample concentration was diluted to lower than the coincidence limit of the OPC (~ 106 cm$^{-3}$). Each OPC channel logged the number of particles larger than the channel threshold value in a 5 s interval. After measuring for 5 minutes, the percentage of 200 and 400 nm mCASTblack bare soot particles in different OPC counting channels could be calculated.*'

**120 L881: 'suggesting' - see comments L338, L877. A better way would be to analyze the volatile components while heating the collected particles, or analyzing the exhaust of a thermal**

**denuder.**

R:     By heating collected bulk soot particles, e.g. thermogravimetric analyzer, the vapor and volatile content in soot samples can be evaluated. By conducting mass spectrum measurement, e.g. single particle soot photometer, the chemical composition of a single particle or particle population can be analyzed. However, we could not find a relation between the volatile components (or mass spectrum results) and the size of single soot-aggregate collected on the grid.

**121 L908: primary particle edges are more ambiguous in Fig. C4f for thickly coated soot particles - this is not obvious, a higher magnification is shown in C4f. what differences one would expect to see for organic volatile content in ultra-high vacuum for HRTEM and high energy beams? Figures like this of lower significance and their description can be moved to a supplement.**

R:     As replied to comment 95, the evaporation of volatiles or coated $H_2SO_4$ can occur but this would not cause pronounced change to soot-aggregate morphology. Moreover, it will occur to both coated and uncoated soot particles if it is the case. In L908 (in original manuscript, now L936 in revised manuscript), we compared the HRTEM (high resolution transmission electron microscopy) images of uncoated and coated soot-aggregates using the same microscope (TFS F30). As we do not have a supplementary document, we would like to keep these figures as where they are now.

**122 L921: see comment in L867. I think figures C4-C11 can be moved to a supplement. A more informative figure/table would be of a statistical analysis for these measurements, which is the standard practice in electron microscopy for extrapolation of properties to a broader sample area.**

R:     Please see our response to comment 115 (for L867 in original manuscript) and 121. And we would like to present these figure as they are now as there is not any supplementary document. Also, as we stated in our response to major comment 2, more than 40 images were randomly taken from each soot sample grid area and we presented more microscopic images (8 imagers per sample). These images show further evidence on soot coating mixing states but it is not good enough to do statistical analysis.

**123 L921: the areas of interest.**

R:     Agreed and accepted (See L954 revised manuscript).

**124 L976: is this a thesis? please indicate.**

R:     Yes, it has been indicated now in revised manuscript in L1010.

**125 L1068 and others: The DOI links don't work. All those should be changed to the right format e.g. https://doi.org/10.5194/acp-17-15199-2017.**

R:     Accepted and changed.

**126 L1994: 2020b should be listed after 2020a?**

R:     Accepted and changed (See L1150 in revised manuscript).

**127 Fig. 1: Exhaust filter, flow splitter – correct labels,**
    **1.5 L/min does that belong to the CPMA on the right?**

R:     Thanks for your comment. Accepted and corrected. And the '1.5 L $min^{-1}$' was moved close to the

CPMA.

**128 Fig. 2: remove the turquoise box in the CPC, irrelevant.**

R:    Agreed and removed

**129 Table 1: first column coating T – change to H₂SO₄ evaporation T.**

R:    It was changed.

**130 Fig. 3: The different soot types should be indicated on the plots too.**
         **caption: size – change to diameter measured by SMPS.**
         **Effective – change to calculated effective.**

R:    They were changed.

**131 Table 2: first column, please elaborate what is that temperature. Also, is this table really**
         **necessary or the numbers can be mentioned in the main text? What's the significance of**
         **mentioning 30C results, which suggest, if I understood correctly, that there might be no coating?**

R:    Firstly, the first column in Table 2 is the same as Table 1 and now is changed to '*H₂SO₄ evaporation*
*T (°C)*'.

         Second, as addressed in the caption, this table presents the coating *wt* % for 200 nm soot particles
collected for TEM (transmission electron microscopy) analysis, which has similar coating *wt* % but not
the same as for the soot particles used for SEM (scanning electron microscopy) analysis. The temperature
of 30 °C should be mentioned because the coating does occur at this temperature. Not only because a
slight mass increase was detected (coating *wt* % = 2.9) but also because the ice nucleation activity of
thinly coated soot particles (see Fig. 4, coating *wt* % = 2.7) has been significantly depressed.

**132 Fig. 10: I suspect multiple measurements were done to make this figure rather than one single**
         **measurement for each nanoparticle type and inference for the whole grid. Please include some**
         **statistical analysis of sampling in different locations on same particle and/or on different**
         **particles on the grid in each category and add stdev values to elemental percentage that you**
         **present. With the currently presented values, the difference significance between all 3 is not**
         **obvious. Why did you choose to use a precision of 1 decimal for carbon and oxygen but 2**
         **decimals for sulfur?**

R:    Our 200 nm soot particle sample grid was firstly used for TEM (transmission electron microscopy)
images collection (microscopes TFS F30 and Hitachi HT7700) and then analyzed for element maps by
using the EDX (energy dispersive X-ray, TFS Talos F200X) technique. In Figs. 10 and 11, the soot-
aggregate was firstly detected by STEM (scanning transmission electron microscopy) and then element
composition of the areas of interest was analyzed by the software Esprit 1.9 (Bruker). The detected soot-
aggregates and the areas of interest (AOI) were selected arbitrarily. As we already provided more TEM
images in response to the second major comment, we did not think it is still necessary to provide more
statistical analysis for EDX results. Moreover, it does not contribute to a further understanding by
calculating the STDEV values to elemental percentage from some EDX AOIs because it is already
demonstrated that the distribution of H₂SO₄ coating is not uniform. As can be seen for the case in Figs.
10 and 11, the detected S percentage in the AOI is not constant with the H₂SO₄ coating *wt* % for the
global soot-aggregate coating. Presenting EDX results without statistical calculation is also commonly

used by the other studies in the literature (Belosi and Santachiara, 2019; Rabha and Saikia, 2020; Jahl et al., 2021; Bora et al., 2021).

We used one decimal for C and O percentage and two decimals for S percentage because the amount of S is too small and one decimal is not enough to present the S content change by $H_2SO_4$ coating. In addition, one may not expect strictly precise values from EDX results as we addressed that the distribution of $H_2SO_4$ coating is not uniform (see L405, L410, L476, L593 and L623 in revised manuscript).

**133 Fig. 11: see comment for Fig. 10.**

R:    Same response as above.

**134 Fig. 12: interestingly, you illustrate here the sulfuric acid as homogeneously nucleated droplets rather than vapour condensing in monolayers? Or perhaps I got it wrong?**
**I'm not sure I understand this conceptual collapse in Fig. 12d. what forces are causing it? I would expect the particle to collapse upon drying due to surface tension exerted on the aggregate rather than during the coating process.**
**Also, in all references it's sulfuric while you use sulphuric, consider changing throughout the text for consistency.**

R:    We believe the reviewer misunderstood. We did not mention that the $H_2SO_4$ coating in the form of condensed vapor monolayers. In fact, we suggested a non-uniform and heterogeneous distribution of $H_2SO_4$ coating over soot particles (see L405, L410, L476, L593 and L623 in revised manuscript). In addition, we did not mention or indicate the red dot in Fig. 12 as homogeneously nucleated droplets but mentioned it as $H_2SO_4$ molecules (L629 in original manuscript, now L637 in revised manuscript).

Secondly, some studies observed compacted soot-aggregates internally mixed with coating materials (Bhandari et al., 2019; Zhang et al., 2008). Some researchers suggested that soot compaction occurs during the condensation or addition of coating materials (Hallett et al., 1989; Tritscher et al., 2011; Schnitzler et al., 2017) because the coating material exerts capillary forces between coating monomers leading to soot-aggregate restructure and a more compact morphology.

Thirdly, we are using commonwealth spelling other than American spelling so that 'sulphuric' is used other than 'sulfuric'. However, the final spelling will be decided during ACP typesetting and language editing.

**135 Fig. B8: After this figure, or in the main text, I'd recommend to add a composite figure that will include all the results of onset saturation ratio with respect to ice freezing where 1 % of the aerosol particles are activated as a function of temperature.**

R:    Thanks for this suggestion. The onset saturation ratio for soot samples with respect to ice activated fraction (AF) 1 % as 1.0 μm ice crystals can be read from the AF curves at different temperatures in figures attached in Appendix B. As we investigated four soot samples (200 and 400 nm both for mCASTblack and FW200) and each soot sample has at least eight $H_2SO_4$ coating masses, the ice onset plot will be very busy and with a low readability.

**136 Fig. C3: what are the implications of these charts on the ice nucleation results and analysis? A discussion is missing in the main text.**

R:    As addressed in L875-885 (in original manuscript, now L907-918 in revised manuscript), the

particle optical size measurement for 200 and 400 nm bare mCASTblack soot particles was conducted to exclude the existence of super micron size particles in high concentration 200 nm mCASTblack samples for TEM measurement. The results demonstrated the absence of micron size soot-aggregates and that visualized micron size soot-aggregates in TEM (transmission electron microscopy) images may be because of soot-aggregate agglomeration upon girds sampling as the particle number concentration is as high as 3000 cm$^{-3}$.

---

## Author Response (AR2)

Response to acp-2021-645 reviews for RC3

We are grateful to reviewer 3 for reviewing our manuscript acp-2021-645. Please find our responses in below and corrections in the revised manuscript. Your comments are reproduced **in bold** and our responses are given directly afterward in normal font. *The original text in previous reviewer's RC document and our response document is reproduced in red italic* and *the text changed in the revised manuscript is in blue italic*.

- **Minor:**

1   **In the last paragraph of the Summary, it might be helpful to provide a couple of sentences to discuss the implications of the findings of this work to the model community. Should the aerosol models keep track of the amount of $H_2SO_4$ coated on soot particles for more accurate INP calculations?**

R:   Thank you. We added relevant statements in the last paragraph in the Summary as below (in the revised manuscript L767-772):

'*...Moreover, this study contributes to the climate impact evaluation of soot emissions. Despite the various coating thicknesses in the atmosphere, a thin $H_2SO_4$ coating ($\leq$ monolayer coverage) onto soot particles can lead to suppressed ice nucleation. As such, aerosol-climate model may only need to differentiate coated and uncoated soot particles when evaluating the $H_2SO_4$ coating effects on soot ice nucleation abilities as monolayer of sulphuric acid coating can be easily achieved in the atmosphere (Pósfai et al., 1999; Adachi et al., 2011).*'

2   **In response to Comment #5 of the first reviewer (RC1), the authors mentioned that they have submitted a manuscript to ACP where they present ice nucleation onto soot particles down to 60 nm (Gao et al., 2021). I think that the authors should cite Gao et al.'s paper and summarize its key relevant findings here.**

R:   Thank you for this suggestion. Yes, we agree that the investigation of size threshold for soot ice nucleation is important. A new statement was added at the end of the Sect. 5 as below (L767-770 in revised manuscript):

'*...Moreover, measuring the lower size limit to identify the threshold of soot ice formation will be important for future laboratory studies, given the strong particle size dependence of (soot) ice nucleation and that the Aitken mode dominates size distribution from aviation soot emissions and also other high temperature combustion sources.*''

Now, the revised manuscript in L762-767 is as below:

'… *The findings in this study have implications on the ice nucleation of smaller size soot particles (< 100 nm). The understanding of the lower size threshold for soot ice formation is important, given the strong particle size dependence of (soot) ice nucleation and that the Aitken mode dominates size distribution from aviation soot emissions and other high temperature combustion sources. A recent study reported that small size (< 100 nm) soot particles, even as active as uncoated FW200 in this study, nucleate ice homogeneously for T < HNT (Gao et al., 2021), implying that $H_2SO_4$ coated soot particles of a size smaller than 100 nm require homogeneous freezing conditions to form ice in the cirrus cloud regime.*'

Adachi, K., Freney, E. J., and Buseck, P. R.: Shapes of internally mixed hygroscopic aerosol particles after deliquescence, and their effect on light scattering, Geophys. Res. Lett., 38,

http://10.1029/2011gl047540, 2011.

Gao, K., Friebel, F., Zhou, C.-W., and Kanji, Z. A.: Enhanced soot particle ice nucleation ability induced by aggregate compaction and densification, Atmos. Chem. Phys. Discuss., *in press,* http://10.5194/acp-2021-883, 2021.

Pósfai, M., Anderson, J. R., Buseck, P. R., and Sievering, H.: Soot and sulfate aerosol particles in the remote marine troposphere, J. Geophys. Res. Atmos., 104, 21685-21693, https://10.1029/1999jd900208, 1999.